# Unique benthic foraminiferal communities (stained) in diverse environments of sub-Antarctic fjords, South Georgia

Wojciech Majewski[1], Witold Szczuciński[2], Andrew J. Gooday[3,4]

[1] Institute of Paleobiology, Polish Academy of Sciences, Twarda 51/55, 00-818 Warszawa, Poland
[2] Geohazards Research Unit, Institute of Geology, Adam Mickiewicz University, Poznań, Bogumiła Krygowskiego 12, 61-680 Poznań, Poland
[3] National Oceanography Centre, European Way, Southampton SO14 3ZH, UK
[4] Life Sciences Department, Natural History Museum, Cromwell Road, London SW7 5BD, UK
*Correspondence to*: Wojciech Majewski (wmaj@twarda.pan.pl)

**Abstract.** Sub-Antarctic fjords are among the environments most affected by the recent climate change. In our dynamically changing world, it is essential to monitor changes in these vulnerable settings. Here, we present a baseline study of "living"

(rose Bengal stained) benthic foraminifera from fjords of South Georgia, including fjords with and without tidewater glaciers. Their distribution is analyzed in the light of new fjord water and sediment property data, including grain size and sorting, total organic carbon, total sulfur, and $\delta^{13}C$ of bulk organic matter. Four well-defined foraminiferal assemblages are recognized. *Miliammina earlandi* dominates in the most restricted, near-shore and glacier-proximal habitats, *Cassidulinoides* aff. *parkerianus* in mid-fjord areas, and *Globocassidulina* aff. *rossensis* and an assemblage dominated by *Ammobaculites rostratus*,

*Reophax subfusiformis*, and *Astrononion echolsi* in the outer parts of fjords. *Miliammina earlandi* can tolerate strong glacial influence, including high sedimentation rates in fjord heads and sediment anoxia, as inferred from sediment color and total organic carbon/sulfur ratios. This versatile species thrives both in the food-poor inner reaches of fjords that receive mainly refractory petrogenic organic matter from glacial meltwater, and in shallow-water coves where it benefits from an abundant supply of fresh, terrestrial and marine organic matter. A smooth-walled variant of *C.* aff. *parkerianus*, apparently endemic to

South Georgia, is the calcareous rotaliid best adapted to inner fjord conditions characterized by moderate glacial influence and sedimentation rates and showing no preference for particular sedimentary redox conditions. The outer parts of fjords with clear, well-oxygenated bottom water, are inhabited by *G.* aff. *rossensis*. *Ammobaculites rostratus*, *R. subfusiformis*, and *A. echolsi* dominate in the deepest-water settings with water salinities $\geq 33.9$ PSU and temperatures 0.2–1.4 °C, characteristic for Winter Water and Upper Circumpolar Deep Water. The inner- and mid-fjord foraminiferal assemblages seem specific to South

Georgia, although with continued warming and deglaciation they may become more widespread in the Southern Ocean.

## 1 Introduction

South Georgia (SG) is uniquely positioned between the northern and southern streams of the Antarctic Circumpolar Current (ACC) (Orsi et al., 1995). Since 1925, it has experienced significant warming of the surrounding shallow oceanic waters (Whitehouse et al., 2008) and widespread glacier retreat (Gordon et al., 2008; Cook et al., 2010), changes that coincide with

the climatic reorganization of the Southern Ocean caused by the southward shift of the ACC (Gille, 2014) and intensification and southward migration of the Southern Westerly Wind belt (Perren et al., 2020). These major environmental changes will likely continue into the future, strongly affecting marine and terrestrial ecosystems (Constable et al., 2014). They will be particularly evident in fjords (Bianchi et al., 2020) and especially in the highly sensitive sub-Antarctic island of SG, where biological invasions due to increasing vessel traffic are also likely to occur (Frenot et al., 2005). Now is therefore a timely

moment to widen our understanding of fjord biota and their links with the local environmental conditions.

This study focuses on foraminifera, a major component of benthic assemblages in marine ecosystems. They are abundant, highly diverse, and many species are preserved in the fossil record. The short life cycle of foraminifera means that they are highly responsive to ecological changes (Jorissen et al., 2009) and therefore particularly valuable for reconstructing paleoenvironments and for monitoring the current state of marine environments (Murray, 1991). Moreover, elemental and

isotopic composition of tests of calcareous foraminiferal may be calibrated to reflect composition of ambient sea water, which makes them useful for reconstructing past environmental conditions, including water temperatures and salinities (de Nooijer et al., 2014). All these factors make foraminifera an important subject of study.

The primary goal of this research is to document the distribution of different foraminiferal communities inhabiting fjords of SG. Based on new hydrological (water salinity and temperature) and sedimentological data, including grain size and sorting,

total organic carbon (TOC), total sulfur (TS), and $\delta^{13}C$ of bulk organic matter, we aim to link these communities with their typical habitats. This will provide a present-day baseline for interpreting fossil records and monitoring future faunal change. Finally, we compare the composition of foraminiferal assemblages inhabiting fjords of SG with those in similar habitats north and south of the ACC. These results may provide insights into likely faunal and ecosystem changes in the Southern Ocean linked to anticipated climate warming.

**1.1 Study area**

South Georgia is approximately 170 km long and up to 40 km wide, making it one of the largest sub-Antarctic islands, as well as one of the most remote. It is located within the influence of the ACC (Fig. 1). The southern ACC front wraps anticyclonically around SG before continuing westwards to the north of the island (Thrope et al., 2002). The island is surrounded by a wide shelf composed of continental crust with a long geological history (Curtis et al., 2010) and water-

depths rarely exceeding ~300 m (Graham et al., 2008). The closest large land areas are the Falklands and Southern Patagonia, ~1400 km and ~1700 km, respectively, to the west, and the Antarctic Peninsula ~1500 km to the southwest. The smaller archipelagos of the South Sandwich Islands and the South Orkney Islands, lie about 700 km southeast and about 850 km southwest of SG, respectively (Fig. 1).

SG is under the influence of a maritime climate with mean annual temperatures of ~2 °C at Grytviken and annual

precipitation of ~1400 mm (Smith, 1960). Weather conditions can be quite variable from year to year, depending on the behavior of the ACC (Cook et al. 2010). SG is strongly affected by Southern Westerly Winds that make its SW shores quite

exposed and NE more sheltered. Due to structural upwelling on the leeward side of the archipelago, the close proximity of oceanic fronts, and local lithogenic sources, waters are rich in nutrients and productive (Davenport, 1995). Coastal water is very cold throughout the year, and in winter and spring it approaches freezing point. In Cumberland Bay, late summer fjord

water temperatures decrease from ~2.5 °C to ~1.5 °C and salinity increases from 33–34 PSU to >34.2 PSU near the sea floor at ~250 m water depth (Römer et al., 2014; Geprägs et al., 2016). The water column is stratified, with lower salinity Local Water at the surface (<33.6 PSU), underlain by a water mass resembling Antarctic Surface Waters, and usually a deepest water mass approaching the density of Winter Water and Upper Circumpolar Deep Water (Meredith et al., 2003).

SG is a heavily glaciated island, with ice caps and snowfields covering over 50% of the surface. Many fjords are terminated

by tidewater glaciers. Out of 103 coastal glaciers, 97% have retreated since the 1950s (Cook et al., 2010). The areas of this study include most of the Cumberland East Bay and West Bay. These are fed at their heads by the Nordenskjöld and Neumayer glaciers, which rank among the largest on the island but show strikingly different retreat rates. While the Nordenskjöld Glacier is almost stable (Gordon et al., 2008), the retreat rate of the Neumayer Glacier accelerated to nearly 400 m/year during 2005–2008 (Cook et al., 2010). The other main sampling area was Stromness Bay. Its catchment is

mainly ice-free, and only small cirque glaciers are present. All these bays are located in the center of the NE coast of SG. Additional material was collected in Fortuna Bay and Antarctic Bay, located to the north-west (Fig. 1).

Detailed multibeam swath bathymetry of the studied fjords indicated diverse geomorphology with underwater glacial moraine ridges and distinct basins overprinting large-scale geomorphological features, including troughs that continue out to the shelf edge (Hodgson et al., 2014). The age of these submarine features, both larger- and smaller-scale, remains

controversial because marine geological data are sparse (Barnes et al., 2016). The extent of the LGM glaciation is also debated, although an extensive LGM glaciation overriding at least much of the continental shelf (Graham et al., 2008, 2017; Barlow et al. 2016; White et al., 2018) seems more likely than the restriction of LGM glaciers to the fjords (Bentley et al., 2007; Hodgson et al., 2014). The millennial scale sediment accumulation rates are known only from a few sediment cores recovered from the outer parts of Cumberland Bay and vary from 0.1 to 0.2 cm/yr (Berg et al., 2019, 2020; Graham et al.,

2017).

**1.2 Previous research**

Terrestrial ecosystems of sub-Antarctic islands, including SG, have attracted considerable scientific attention, focusing on the impacts of climate change as well as the dispersal of invasive species (Bergstrom and Chown, 1999). However, because of its remoteness, marine ecosystems of SG remain largely understudied (Barnes et al., 2005), despite their important

contribution to global and regional biodiversity (Hogg et al., 2011).

Nevertheless, thanks to financing from the whaling industry, SG was the location of one of the classic foraminiferal studies in the Southern Ocean (Earland, 1933, with preliminary papers by Heron-Allen and Earland, 1930, 1932). This work was based mainly on surface sediments collected during expeditions of the R.R.S. *Discovery* (1925) and R.R.S. *William Scoresby*

(1926–1930), mainly at shelf sites but including also some shallower stations in Stromness Bay and Cumberland East Bay and West Bay. These publications established 49 new taxa, making SG one of the taxonomically best-known areas for foraminifera in the Southern Ocean. Apart from a recent paper based on samples from the outer shelf and upper slope (238–354 m water depth) north of SG (Dejardin et al., 2018), and an earlier study that included several sites at similar upper bathyal depths to the south of SG (Echols, 1971), nothing further has been published on foraminiferal distribution and taxonomy in this region. However, a number of foraminiferal studies have focused on surrounding shelf areas, including the Falkland Islands and their adjacent shelf (Heron-Allen and Earland, 1932), fjords and channels of Patagonia (Violanti et al., 2000; Hromic et al., 2006), the South Shetland Islands (Finger and Lipps, 1981; Gray et al., 2003; Majewski, 2005, 2010; Majewski et al., 2007; Rodriguez et al., 2010), and the Antarctic Peninsula (Ishman and Domack, 1994; Ishman and Szymcek, 2003; Murray and Pudsey, 2004; Majewski et al., 2016). Research was also conducted in deep-water regions, highlighting foraminiferal gradients across the Drake Passage (Herb, 1971) and the frontal system of the ACC (Mackensen et al., 1993). Together, these works provide a firm regional background for this study.

## 2. Material and methods

### 2.1 Fieldwork

The fieldwork was conducted in fjords located in the central part of the northern coast of SG (Fig. 1 and Table 1) from the *SRV Saoirse* in November and December of 2019. Hydrologic parameters, namely water density, temperature, and salinity (conductivity), were measured using a CTD48M Memory probe (Sea & Sun Technology GmbH) at 20 stations (Table 1). Bottom-water measurements for particular stations were based either on direct CTD measurements or extrapolated from the nearest CTD profile at a similar water depth, since the water properties indicated a uniform water column stratification throughout each particular fjord (Fig. 2). The salinity is expressed in PSU (practical salinity unit). The classification of local water masses followed mainly a scheme proposed by Orsi et al. (1995), Meredith et al. (2003), Carter et al. (2008), and Geprägs et al. (2016).

### 2.1 Sampling

All sediment samples were collected using a Van Veen grab sampler with a sample area of 1000 cm², manufactured by KC Denmark (model 12.211 matching Norwegian ISO standards). The grab was equipped with inspection windows and rubber plates that opened during descent. After the grab was recovered and secured on board, the inspection windows were opened and excess seawater carefully removed. In the case of over penetration, or if the sediment surface was disturbed (e.g., due to failure in sealing the grab during the ascent), the opertion was repeated until a good sample was recovered.

For quantitative foraminiferal analyses, the upper 2 cm of sediment was sampled from an area of 63.5 cm$^2$, defined by a ring 9 cm in diameter pressed to the sediment surface through one of the inspection windows. Samples for grain size and

geochemical analyses were taken from the remaining surface sediment if available. Additional material was also taken for studies of delicate, gromids (Gooday et al., 2022) and monothalamous foraminifera (Holzmann et al., 2022). At most stations, two replicate samples were obtained from two separate deployments of the grab. At stations SG-20 and SG-26, the boat's position could not be maintained due to difficult conditions, resulting in the recovery of replicates from significantly different water depths. In these two cases, both samples were analyzed. Replicates from the remaining stations were archived. In total, 29 samples were fully processed for stained foraminifera: seven from Cumberland East Bay, eight from Cumberland West Bay, three from outer Cumberland Bay, eight from Stromness Bay, two from Antarctic Bay and one from Fortuna Bay (Fig. 1 and Table 1). They were obtained at water depths down to 250 m, i.e. including the deepest parts of the fjords. To study the depth distribution of "live" (stained) foraminifera in the sediment profile, grab samples taken at stations SG-21 and SG-27 were sub-sampled with tubes of 7-cm diameter, providing cores 6 cm long. These were sliced into 1 cm thick intervals directly after recovery.

**2.3 Sediment types and grain size analysis**

All surface sediment samples were described onboard in terms of sediment type, color, presence of ice-rafted clasts, and macro-biota. Subsamples were taken from 25 grab samples for volumetric grain-size analysis using laser beam diffraction. Prior to the analysis, grains >2 mm in size and organic matter were removed by passing the sample through a sieve and the remaining sediment treated with sodium hexametaphosphate and ultrasound to avoid grain aggregation. The analyses were performed in a Mastersizer 2000 Particle Size Analyzer and the resulting grain-size statistics were calculated with GRADISTAT (Blott and Pye, 2001) using the logarithmic method of moments.

**2.4 Geochemistry**

Total carbon, TOC and TS analyses were performed using an Eltra CS-500 IR-analyzer at the Faculty of Earth Science, University of Silesia, following standard procedure (Racka et al., 2010). The TOC/TS ratio was used as an approximate indicator of sediment redox, where ratios <1.5 indicate anoxic, 1.5–5 periodically anoxic, and >5 oxic conditions (Berner, 1983). The stable carbon isotopes of bulk organic matter in the sediments were analyzed using a Thermo Electron DeltaV Advantage IRMS with ConFlo II and a Carlo Erba NA1500 Elemental Analyzer at the University of Florida. The data were related to the USGS40 standard (n=28, standard deviation SD=0.081). All carbon isotopic results are expressed in standard delta notation relative to Vienna Peedee Belemnite (VPDB).

**2.5 Micropaleontology**

Directly after collection, samples were gently washed on a 63-μm-mesh sieve with cold seawater and stained with rose Bengal (2g/l) and 70% ethanol diluted in sea water. Samples were left to stain for at least two days, washed, and left to dry for easier transport. The 63–125 μm and >125 μm grain-size fractions were dry-picked. Wherever possible, at least 300

stained individuals were picked from each fraction. If samples yielded <300 stained individuals, specimens from replicates

were also picked in the same way as the regular samples. Consequently, for stations SG-12, SG-13, SG-14, SG-16, and SG-28, specimens from both replicates were further analyzed. Samples rich in foraminifera were divided using a dry microsplitter and all stained tests were picked from the splits. Diversity indices, i.e., Shannon diversity index and dominance were calculated using PAST4.03 software (Hammer et al., 2001).

Transparent calcareous foraminifera were classified as "living" if at least the final chamber was occupied by brightly red- or

165 violet-stained cytoplasm (Silva et al., 1996) and opaque agglutinated tests if the cement was intensely stained, especially when wet, or if stained material filled chambers that had been broken open. Porcellaneous foraminifera (miliolids) were regarded as "living" if material exposed in the aperture was stained or if the test acquired a distinct coloration after re-wetting (Schönfeld et al., 2012).

All foraminiferal specimens were arranged by taxon on micropaleontological slides and counted. Photographic

documentation of specimens typical for each species (Figs. C1–C6) was performed with a Phillips XL20 Scanning Electron Microscope. The classification scheme used here is that of Loeblich and Tappan (1987) and the WoRMS database (Hayward et al., 2021). For species of Cassidulinidae, we adopted the latest species-level taxonomy of Majewski et al. (2021), which is based on molecular data. Specimens are housed at the Institute of Paleobiology, PAS (Warszawa) under the catalogue number ZPAL F.65.

**2.6 Statistics**

As a first step, we selected species that were most characteristic of different environmental settings. The frequencies of foraminiferal specimens collected from the >125 μm fraction and the entire assemblage (>63 μm; i.e., the >125 plus 63–125 μm fractions), were analyzed individually with Q-mode orthogonal rotated (Varimax) principal component (PC) analysis, following Malmgren and Haq (1982) and Mackensen et al. (1990), and using a commercially distributed statistics package

(SYSTAT 12). Fragile monothalamous foraminifera, such as *Cribrothalammina*, *Pelosina*, and *Vanhoefenella*, which are not preserved in sub-fossil samples, as well as species never surpassing 1% of the entire fauna and those present only at a single station, were excluded from the statistical analyses, as was sample SG-28 for the >125 μm dataset, which yielded only 28 specimens in that size fraction. This procedure left 36 and 43 taxa in the >125 μm and >63 μm datasets, respectively.

In the second step, a P-mode PC analysis with the number of factors set at two was conducted and PC loadings were plotted

to identify species with similar distributions. Species within distinct groupings were assumed to have similar environmental requirements, thus belonging to distinct foraminiferal assemblages (Caulle et al., 2014). Combined abundance, expressed as a percentage of the entire assemblage, was then used for further analysis. Canonical correspondence analysis (CCA) was conducted using PAST4.03 software (Hammer et al., 2001) in order to investigate the relationship between cumulative percentages of foraminiferal assemblages (FAs), as well as environmental parameters and sediment properties including

water depth, distance to major sediment source, distance to fjord mouth, bottom-water salinity and temperature, TOC, TS,

$\delta^{13}C$, mean grain size and sorting.

## 3    Results

### 3.1  Fjord hydrology and sediments

#### 3.1.1 Water temperature and salinity

The hydrology of the fjords was surveyed during the spring-summer onset of 2019 and revealed a similar pattern of water

mass distribution in all the studied fjords (Fig. 2). The surface Local Water layer was characterized by relatively low salinity

(28.4 to 33.6 PSU) and variable temperatures (0.5 to 7.6 °C). This water mass ranged in thickness from ~30 m in the inner

part of Cumberland Bay (SG-24) and 26 m in the inner Stromness Bay (SG-18) to 2.4–3.2 m at the outer fjord/shelf stations

(e.g. SG-01, SG-03, and SG-21). The lowest salinity (<32 PSU) was found in the surface water of the inner Cumberland

West Bay (SG-23, SG-24, and SG-25), which was influenced by meltwater from the tidewater Neumayer Glacier, and in

bays supplied with freshwater snowmelt by streams and small rivers (e.g. stations SG-17 and SG-19 in Stromness, and

shallow coves in Cumberland Bay: SG-08, SG-12 and SG-22). The lowest temperatures for Local Water (<1 °C)  were noted

in the inner parts of Cumberland Bay affected by tidewater glaciers, and the highest (>5 °C) in the inner part of Stromness

Bay.

The major fjord water mass was found below Local Water, down to c. 190–200 m water depth (mwd) (Fig. 2). It was

characterized by temperatures that decreased downwards from approximately 2.5 to 0.2 °C, and salinities of between 33.6

and 34 PSU. This water mass was generally similar to Antarctic Surface Waters and to Winter Water in the deepest parts.

The deepest water mass (>200 mwd) was encountered in two parts of Cumberland Bay (Fig. 1), near the fjord mouth and in

the innermost western arm of the fjord, which is bounded by a shallow sill, <30 mwd according to *Admiralty Chart 3588*

(Fig. 1), and supplied with meltwater by the rapidly retreating Neumayer Glacier. In the outer part of Cumberland Bay (SG-

11 and SG-21), the salinity of this deepest water mass increased with depth reaching almost 34.4 PSU, while its temperature

slightly increased downward from 0.2 to 1.3 °C, thus corresponding to Upper Circumpolar Deep Water. However, in the

innermost part of the fjord (SG-22), the water mass below 200 m was colder <0.2 °C and with maximum salinities reaching

only 33.86 PSU.

The temperature and salinity of the near-bottom water, which directly influences the benthic biota, ranged from 0.1 to 2.3 °C

and 33.5 to 34.35 PSU and varied with water depth (Fig 2 and Table A). The highest temperatures (>1.5 °C) were found at

stations shallower than 60 m, located mainly in small bays far away from glacier fronts in Cumberland Bay (SG-22) and

Stromness Bay (SG-28, SG-29, and SG-30) but also at SG-13 located close to the front of the Nordenskjöld Glacier.

Moderate near-bottom temperatures of 0.75 to 1.5 °C were also found at shallow stations (<60 mwd), as well as at the

deepest stations of outer Cumberland Bay (SG-11 and SG-21, both >240 m deep). The lowest near-bottom water

temperatures (0.1–0.6 °C) were encountered at water depths of 90–250 m. Near-bottom waters with a salinity of <33.8 PSU

were found at water depths <45 m, and those with values between 33.8 and 34 PSU at depths between 37 and 210 m. In the deepest stations (SG-07, SG-11, and SG-21), water salinity was slightly >34 PSU.

### 3.1.2 Sedimentology and grain size

Sediments in Antarctic Bay (SG-01 and SG-03) were composed of light grey, poorly sorted glacimarine mud (medium to coarse silt). Sediments in Fortuna Bay (SG-04) were also composed of gray, poorly sorted sandy mud inhabited by Ophiuroidea. In Stromness Bay, sediments mainly comprised poorly-sorted olive-brown to light brown sandy mud. The mean grain size displayed a coarsening offshore trend, from medium silt in inner fjords to very coarse silt/fine sand at the fjord mouth (Fig. 3). Sediment in the central part of Stromness Bay (SG-27) was covered with an algal mat. There was some

kelp and abundant plant detritus at station SG-28, located next to Grass Island at the entrance to the Stromness Harbour, the middle branch of Stromness Bay. Here, the sediments below the thin oxidized layer were dark (blackish). At station SG-29 (46 mwd), a near-shore location located at the fjord mouth and exposed to ocean swell, sediment was composed of dark, moderately sorted, fine sand.

Samples retrieved from the small coves of Cumberland Bay, i.e., Jason Harbour, King Edward Cove, Sandbugten, and

Maiviken, contained olive-gray, poorly sorted muds. In Jason Harbour (stations SG-08 to SG-10) they displayed a subtle, offshore fining trend. Sediments from the restricted, shallow-water settings of Jason Harbour (SG-08) and King Edward Cove (SG-12) were characterized by a circa 1 cm thick light layer with dark (blackish) sediment below. At other stations, the color difference between surface and underlying sediment was much less striking. Fragmented kelp, algal mats, as well as biogenic sediment granules covered the surfaces of sediments retrieved at stations SG-08, SG-09, and SG-12. Sediment

collected from deeper water (SG-10) contained more ice-rafted debris (IRD) with gravels up to 6 cm in diameter as well as tubes of Polychaeta.

Sediments in the inner part of Cumberland West Bay (SG-23 and SG-24) were light-gray, very loose, and very poorly sorted sandy muds with IRD gravel. In the outer part (SG-32, SG-11, SG-20, and SG-21), they were composed of IRD-rich, poorly sorted sandy muds and muds, colonized by macroscopic benthic organisms (e.g. sponges, Polychaeta). At station SG-31,

located west of the shallow sill in the middle part of the fjord, gray and well-sorted coarse sand predominated. Cumberland East Bay was characterized by gray (SG-13 and SG-16 in the inner part) to olive-gray (SG-06 and SG-07 in the outer part of the bay), poorly sorted, mud to sandy mud (SG-13 next to the glacier). These sediments displayed an offshore grain-size fining trend (Fig. 3). In front of the tidewater glacier (SG-13), the sediment was loose, rich in IRD and inhabited by Ophiuroidea, while in the central part of the fjord polychaeta tubes were present more frequently than at other locations.

The majority of the analyzed sediment samples were classified as poorly sorted, fine to coarse silt. The sorting was generally correlated with mean grain size, the finer sediments being generally better sorted (Fig. 3). The two outliers were moderately to well sorted sand samples from the swell-affected, shallow-water station SG-29 in Stromness Bay and from SG-31 located on the slope of a sill/moraine in the central part of Cumberland West Bay, a station apparently swept by currents. When

related to the distance along the fjord, samples collected between 7 and 15 km from the fjord mouth (all from Cumberland

Bay) included the finest sediments and were slightly better sorted than those from other stations, except for the two outliers (Fig. 3b–c). Subtle trends of coarsening towards both the mouth and head of the fjord (Fig. 3b) may reflect a reduced influence of meltwater discharge with distance from the tidewater glaciers, combined with an increasing winnowing effect of the oceanic currents and swell, and/or increasing relative contribution of IRD in bulk sediments towards the fjord mouth.

### 3.1.3 TOC and TS

The total carbon concentrations in the analyzed sediment samples were almost the same as TOC, reflecting the very low total inorganic carbon values (<0.04%). The TOC values varied between 0.23 and 1.1%, except for sample SG-28, located next to Grass Island in Stromness Bay, with a TOC content of 2.92% (Fig. 4). The lowest value (<0.5%) was recorded at all stations located in the inner part of Cumberland Bay, >13 km away from the fjord mouth (Fig. 4A). TOC was elevated (~1%) in the shallow-water settings of King Edward Cove (SG-12), Jason Harbor (SG-08), and stations inside Stromness Bay (SG-17,

SG-18, and SG-27). Sediments from stations near the fjord mouths and in the middle of Cumberland Bay yielded intermediate TOC contents.

TS varied between 0.07 and 0.38% (Fig. 4). The highest values (>0.2%) were limited to the four shallowest stations (SG-08, SG-12, SG-13, and SG-28), all <40 mwd. Intermediate values between 0.15% and 0.2% were only recorded in the innermost part of Cumberland West Bay (SG-24, SG-31, and SG-32), while the remaining samples yielded the lowest values (<0.15%)

(Fig. 4b).

The variations in TOC and TS were not consistent and their ratio was used as a rough indicator of redox conditions (Fig. 5). The highest ratios, typical for mainly oxic sediments, were found in Stromness Bay sediments (all except of SG-29) and in the outer parts of the Cumberland and Antarctic Bay. The lowest values, typical for anoxic sediments, were found in the inner parts of the two main branches of Cumberland Bay, next to terminating tidewater glaciers (Fig. 5a). Two general trends

in TOC/TS were observed. Ratios increasing from the fjord mouth towards the fjord head in Stromness Bay and decreased from the fjord mouth towards tidewater glaciers occupying fjord heads in Cumberland Bay (Fig. 5a).

### 3.1.4 $\delta^{13}$C of bulk organic matter

The $\delta^{13}$C in bulk organic matter ranged from –26.06 to –22.42‰ (Fig. 5), and correlated with TOC. The most negative $\delta^{13}$C values were in samples poorest in TOC, close to the tidewater glacier fronts in the innermost parts of Cumberland Bay. The

280 least negative values were found in samples richest in TOC in the restricted, shallow-water settings of Jason Harbour (SG-08) and King Edward Cove (SG-12), and in Stromness Bay (SG-17, SG-18, SG-27, and SG-28). Two major trends were observed in relation to the distance from the fjord mouth (Fig. 5). In fjords with tidewater glaciers, $\delta^{13}$C became more negative towards the glaciers at the fjord heads. However, in the case of the coves and Stromness Bay, which are not directly

affected by glaciers, values increased in the opposite direction. Station SG-15, located in a semi-restricted cove of

285 Sandbugten at a modest distance of ~12 km from the Nordenskjöld Glacier, showed $\delta^{13}C$ values between these two trends.

## 3.2 Foraminiferal data

### 3.2.1 General indices

In the 29 surface samples, a total of 15191 "living" (i.e., stained) benthic foraminiferal specimens were isolated, including 8136 specimens in the >125 μm fraction (Tables B1–B3). They represented >55 species (Figs. C1–C6). The assemblages

show significant faunal variability, with numbers of taxa in a single sample (>125-μm fraction) ranging from 5 to 32 (average 16.6) and the Shannon diversity index from 0.84 to 2.61 (average 1.70) (n=29), with standard deviations (SDs) of 6.92 and 0.50, respectively. The Shannon diversity index is the highest in outer fjords at depths of ~100 m or more, and the lowest near the heads of fjords with terminating tidewater glaciers (Fig. 6). Dominance reaches 0.12–0.64 (average 0.30). The numbers of stained specimens per sample (63.5 cm$^2$ in area) range from 14 to 1979 (average 551, SD=501). The highest

abundances are found between 150 and 100 mwd in outer parts of fjords and in the deepest parts of Stromness Bay. The lowest abundances are in the shallow parts of Stromness Bay and at locations near fjords heads (Fig. 6). For the >63 μm fraction, the values of all indices, except dominance, are higher (Fig. 6). The Shannon diversity index ranged from 1.18 to 2.85 (average 1.99).

The two short cores taken at stations SG-21 and SG-27 yielded a total of 2810 stained specimens, including 1062 from the

300 >125-μm fraction, representing 47 species. The richest assemblages (>63 μm), in terms of abundances and number of species per sample, were in the upper layers (0–1 cm), totaling 952 and 1724 individuals in cores SG-21 and SG-27, respectively (Fig. 7). The lowest numbers of specimens (36 and 34, respectively) were in the bottom layers (5–6 cm). In the SG-21 sample, the decline in abundances down-core was not uniform, due to a sharp increase in *Stainforthia fusiformis* between 2 and 3 cm below sediment surface.

### 3.2.2 Identification of foraminiferal assemblages

Based on the Q-mode PC analyses, four PC models were selected as best reflecting actual assemblages for both the >125 μm and >63 μm datasets. They explain 83.1 and 82.0% of the total variance in the two datasets, respectively. The calculated PCs are defined by foraminiferal species with large score values (Tables D1 and D2, Fig. 8). They are hereafter referred to as foraminiferal assemblages (FAs) using the names of the dominant taxa, namely the calcareous rotaliids *Globocassidulina* aff.

*rossensis* (Fig. C6: 12–13). and *Cassidulinoides* aff. *parkerianus* (Fig. C6: 6–11), and the agglutinated *Ammobaculites rostratus* (Fig. C2: 1–2), and *Miliammina earlandii* (Fig. D1: 17). The species with the highest PC scores are the same in both datasets and are therefore not affected by the size-fraction bias.

The P-mode PC analysis was also run on the two datasets separately. The four faunal groups identified in the Q-mode PC analysis could be better distinguished in the PC loadings plot for the >125 μm fraction (Fig. 8b). They seem to include different numbers of species, ranging from 3 in the case of the *G.* aff. *rossensis* FA to 16 in the case of the *A. rostratus* FA. Importantly, all FAs combine agglutinated and calcareous species. The plot for the >63 μm fraction is less resolved, showing only two groups of taxa (Fig. 8a), indicating that assemblages may be better resolved in the coarser fraction (Jennings and Helgadottir, 1994; Schönfeld et al., 2012; Caulle et al., 2014). Consequently, only the >125 μm results are discussed further.

### 3.2.3 Spatial distribution patterns of the FAs and their nominative species

Trends in the relative combined abundances of the four FAs as well as in the nominative species alone along the fjord axes are shown in Fig. 9. The *M. earlandi* FA dominated near the fronts of tidewater glaciers and at the most restricted, shallow-water stations. The nominative species, *M. earlandi*, exceeded 35% of the entire assemblage in samples from King Edward Cove near Grytviken and from Jason Harbor (Fig. 6), both small coves with water-depths of ~20 m. *Miliammina earlandi* accounts also for >30% of the entire assemblage at four stations located closest to the fjord heads and tidewater glaciers of Cumberland West Bay and East Bay with water depths of 190 and 28 m, respectively (Fig. 6). In Cumberland West Bay, this species is commonly accompanied by the agglutinated species *Psammosphaera fusca* and *Hippocrepinella hirudinea*, and in Cumberland East Bay by the calcareous species *Gordiospira fragilis* and *Pyrgo patagonica*, all of which contribute to the *M. earlandi* FA (Fig. 8b).

The *C.* aff. *parkerianus* FA dominates in the largest number of samples (Fig. 9). It is common in transitional locations between stations dominated by the three remaining FAs. The nominative *C.* aff. *parkerianus* is the most widespread and abundant species in our dataset. It is represented by two forms (Majewski et al., 2021), a dominant, smooth-walled conical morphotype, assigned by Heron-Allen and Earland (1929) and Earland (1933) to *Ehrenbergina crassa*, and a subordinate porous form resembling *C. parkerianus* sensu Brady (1881), the presence of which is marked on Fig. 9 by red circles. The dominant morphotype reaches the highest percentages, usually well over 50%, far inside Cumberland East Bay and West Bay at water-depths >150 m, as well as at stations SG-08 to SG-10 between 114 and 23 mwd in the middle Cumberland West Bay (Fig. 9). At stations in outer Stromness (SG-26A, SG-26B and SG-27) and outer Cumberland Bay (SG-20A and SG-20B), between 100 and ~150 mwd, the subordinate porous morphotype of *C.* aff. *parkerianus* is also present.

The *G.* aff. *rossensis* FA dominates near fjords mouths, where foraminifera are relatively abundant and rather rich in calcareous forms (Figs. 6 and 9). *Globocassidulina* aff. *rossensis* strongly dominates its FA, reaching up to 51% in the sample from station SG-26A in outer Stromness Bay and exceeding 40% SG-20A in outer Cumberland Bay. It contributes the highest proportion between ~50 and ~150 mwd in the outer reaches of fjords but also at two stations, SG-15 and SG-06 (51 and 121 mwd, respectively), in central Cumberland East Bay (Fig. 5). *Globocassidulina* aff. *rossensis* is also present at water-depths >150 m, well inside Cumberland West Bay at stations dominated by *M. earlandi* and *C.* aff. *parkerianus*, albeit in low and definitely subordinate numbers.

The *A. rostratus* FA shows a strong presence in the middle and outer Cumberland Bay and throughout Stromness Bay, although not at stations <100 mwd. It is especially important in the deepest fjord settings, but not proximal to glacier fronts (Fig. 9). This FA dominates at stations with the highest species diversity (Fig. 6) and comprises the largest number of species, including agglutinated and calcareous forms (Fig. 5a). Its nominative species, the agglutinated *A. rostratus,* is less dominant than those of other FAs and according to the Q-mode PC analysis (Table D1), *Reophax subfusiformis* and

*Astrononion echolsi* and to a lesser degree *Cribrostomoides jeffreysii* and *Pullenia subcarinata,* are also important for defining this FA.

### 3.2.4 Relation between FAs and environmental and sediment properties

CCA was performed in order to explore the relationship between location (water depth, distance to major sediment source, distance to fjord mouth), environmental parameters (water salinity and temperature), and sediment properties (mean grain

size and sorting, TOC, TS, $\delta^{13}$C of bulk organic matter), all listed in Table A, and the cumulative percentages of the four FAs (Table D1 and Fig. 8b). CCA axis 1 explains 73.71% of the variance and axis 2 explains 19.05% (together 92.76%). Most of the variables plot along axis 1, with the *M. earlandi* FA, positive distance to open sea, TS and negative $\delta^{13}$C strongly on the positive side, and most of the remaining parameters and the three remaining FAs on the negative side (Fig. 10). The *A. rostratus* and *C.* aff. *parkerianus* FAs, as well as water-depth, bottom-water salinity and temperature, showed significant

variability along axis 2. The *A. rostratus* FA appears to be correlated with elevated salinity, the *G.* aff. *rossensis* FA with elevated TOC, distance from major sediment source, and less negative $\delta^{13}$C. No clear correlation can be noted for the *C.* aff. *parkerianus* FA, which also shows the weakest relation with axis 1 of the CCA plot.

## 4 Discussion

### 4.1 Data quality

Members of the Foraminiferal BIo-MOnitoring initiative (FOBIMO) have proposed several recommendations for monitoring soft-bottom environments using benthic foraminifera (Schönfeld et al., 2012). We aimed to follow their recommendations, such as collecting replicates, rose Bengal staining, and analyzing the >125 µm as well as the 63−125 µm fractions. In some cases, however, we had to adapt the protocol to our field conditions.

Firstly, because sampling time was limited, staining had to be reduced from the recommended 14 days to only a few days. In

order to ease penetration of the stain, we gently washed the samples on a 63-µm sieve with cold sea-water, as is routinely done for studying fragile monothalamid foraminifera, before adding rose Bengal in ethanol. We believe it was a reasonable precaution, especially since 24–48 hours of staining can already provide satisfactory results (Bernhard et al. 2006). Secondly, we sampled the upper 0−2 cm instead of the 0−1 cm interval as recommended by FOBIMO. Due to a frequent presence of a semi-liquid, flocculent top layer, rich in suspended organic fragments, it was difficult to recognize the sediment-water

interface precisely (Fossile et al., 2020). Our motivation was that, for studying robust calcareous and agglutinated species, it is better to sample a thicker rather than a thinner upper sediment layer, especially considering the sharp decline of "living" individuals with depth in similar fjord settings (Majewski, 2013; Fossile et al., 2020). The final and potentially the most critical modification was our use of a Van Veen grab sampler, a device not recommended by Schönfeld et al. (2012). However, our model, equipped with rubber plates that opened during descent reducing the bow wave, proved to be highly

efficient and reliable during former sampling for soft testate foraminifera (Majewski et al., 2007; Gschwend et al., 2016). A Kajak sampler was also available during our sampling campaign but it was inefficient.

To provide additional confidence in our procedures, we collected and analyzed two short cores from two of the Van Veen grab samples (SG-21 and SG-27; Fig. 7). In both sediment profiles, we observed abundant and diverse stained assemblages (>125-µm fraction) that were concentrated mainly in the surface layer (Fig. 7). Although the lower distribution limits

appeared significantly shallower than on the shelf north of SG (Dejardin et al., 2018), they were comparable to those from similar fjord settings in the South Shetlands (Majewski, 2010) and Spitsbergen (Fossile et al., 2020). Moreover, the 63−125 µm fraction of the SG-21 sub-core revealed a strong subsurface peak in the abundance of *Stainforthia fusiformis* (Fig. 7). The clear distribution pattern of this minute, predominantly infaunal species agrees with its known occupancy of infaunal microhabitats, as reported, for example, by Alve (1994), and confirms that the Van Veen grab samples were relatively

undisturbed. The Shannon diversity index calculated for >63 µm data was in the range 1.18 to 2.85, encompassing the values of 2.29 and 2.36 recorded more offshore by Dejardin et al. (2018). Nevertheless, it is important to note that our surface sampling targeted mainly epibenthic species and that the following discussion is based on the >125 µm data, thereby underestimating the presence of minute, infaunal species, such as *S. fusiformis*.

A final point is that two of the four nominative species for the FAs, *G.* aff. *rossensis* and *C.* aff. *parkerianus,* are

phylogenetically closely related (Majewski et al., 2021) and their immature individuals can be difficult to distinguish (Nomura, 1984). However, the color of the cytoplasm when stained was clearly different, more violet in the case of *G.* aff. *rossensis* and more pinkish in *C.* aff. *parkerianus*, making it easy to discriminate between "living" individuals.

## 4.2 Environmental zones characterized by FAs

We identified four FAs by PC analysis (Fig. 8b). They are very distinctive in terms of species composition and distribution.

Figure 9 emphasizes the gradation between FAs dominating proximal to fjord head stations and those showing a stronger presence near the open sea or at greater water depths. According to the CCA (Fig. 10), these four FAs are related to different environmental conditions and sediment properties. In the following sections, we discuss the environmental setting of each.

### 4.2.1 Inner parts of Cumberland Bay and shallow-water coves: strong glacial influence and sediment anoxia

Restricted areas near tidewater glacier fronts at fjord heads (SG-13, SG-16, SG-24, SG-31) and shallow-water coves (SG-08,

SG-12) are dominated by the *M. earlandi* FA (Fig. 9). In these locations, foraminiferal assemblages are impoverished,

showing generally low numbers of stained specimens, low to moderate species diversities, and, except at SG-08, a clear dominance of agglutinated forms (Fig. 6). The CCA (Fig. 10) indicates that this FA correlates with (1) large distance from the open sea and, although less clearly, proximity to major sediment sources, (2) strongly negative $\delta^{13}C$ values, and (3) high TS.

The first two correlations are somehow ambiguous as the two types of settings characteristic for the *M. earlandi* FA, i.e., near the fjord heads and in the coves (Figs. 1 and 6), although both quite restricted, have rather different characteristics. The environments near tidewater glaciers fronts are characterized by variable water depth, relatively coarse glacially-derived sediments that display a fining trend with distance from the sediment source (Fig. 3), and a thick layer of brackish, turbid surface water (Fig. 2). In these glacial fjords, sediment TOC values are low (<0.5%, Fig. 4a) and organic-matter $\delta^{13}C$ values

strongly negative (Fig. 5b), likely due to the mainly petrogenic source of organic carbon (Berg et al., 2021). Given the climate (high precipitation), mountainous relief, and the susceptibility of rock types to erosion (mudstones and sandstones), it can be expected that sedimentation rates in front of tidewater glaciers are high, in the order of several cm/year, as in the case of similar subpolar fjord systems (e.g. Majewski et al., 2012; Boldt et al., 2013; Streuff et al., 2015). Indeed, preliminary $^{210}Pb$ dating of a sediment core from station SG-16 (c. 2.5 km away from the front of the Nordenskjöld Glacier)

revealed a sediment accumulation rate of ~1.4 cm/yr (Szczuciński, unpublished). In similar depositional environments, the sediment accumulation rate usually exhibits a rapid exponential decrease within the first several km from the glacier front (e.g. Syvitski, 1989; Jaeger and Nittrouer, 1999; Szczuciński and Zajączkowski, 2012). Close to the tidewater glacier fronts, the rates may be in the order of several tens of cm per year. The sediments in glacial fjords are also loose and not compacted, which is also typical for high sedimentation rate conditions. In contrast, sediments in the shallow-water settings of King

Edward Cove and Jason Harbor are among the finest in terms of mean grain size (Fig. 3) and have a thin surficial oxidized lighter mud layer underlain by dark (blackish), probably anoxic sediments. Their sediment accumulation rates were found to be still relatively high, in order of 0.4–0.7 cm/year, as revealed by $^{210}Pb$ dating (Szczuciński, unpublished). These sediments also yielded high TOC concentrations (>1%, Fig. 4a) and $\delta^{13}C$ ratios that were the least negative in this study (Fig. 5b). These differences seem to reflect different organic-matter sources (Caulle et al., 2014; Jernas et al., 2018), with petrogenic

OC dominating near the tidewater glacier fronts and fresher organic matter of mixed origin (both terrigenous and marine) dominating in the coves (Berg et al., 2021).

These fairly substantial differences raise the question of whether *M. earlandi* might comprise several cryptic species, as suggested by molecular data for a related species, *Miliammina fusca* (Pawłowski and Holzmann, unpublished). All stations dominated by the *M. earlandi* FA, however, do have one factor in common, namely elevated TS, roughly ~0.2% (Fig. 4b).

The environmental implications of TS alone are difficult to interpret unequivocally but the proportion TOC/TS has been used to evaluate the redox conditions in sediments (Berner, 1983). Thus, although we do not have direct measurements of bottom-water oxygenation or oxygen penetration depths (Caulle et al., 2014), the low TOC/TS ratios suggest oxygen deficiency at all stations dominated by *M. earlandi* FA (Fig. 5a). It is also in accordance with macroscopic sediment

observations in shallow coves sediment samples. This interpretation is consistent with the occurrence of other representatives

of *Miliammina* in dysoxic settings (Hayward and Hollis, 1994; Tyszka, 1997; Habura et al., 2006).

Earlier studies have shown that *Miliammina earlandi* is not confined to near-shore habitats in the Southern Ocean. This species has a rather complicated taxonomic history, the name having been established by Loeblich and Tappan (1955) to replace the earlier name *M. oblonga,* which was applied incorrectly by Heron-Allen and Earland (1930) and Earland (1933, 1934) (see also p. 53 in Loeblich and Tappan, 1987). The lectotype designated by Loeblich and Tappan (1955) is from 200

445 m depth, near the shelf edge well to the south of SG. According to Earland (1933), *M. oblonga* (= *M. earlandi*) is 'universally distributed' and 'probably the commonest and most widely distributed rhizopod of the South Georgia area'. The sites where it is common range from 100 to 200 m, consistent with our new data, but it is also found down to 1752 m. In his subsequent *Discovery* report on the foraminifera of the Falklands area, Earland (1934) concludes that *M. oblonga* (= *M. earlandi*) is 'generally distributed in all areas within the Antarctic convergence and sometimes very common'. The stations he mentioned

are at depths of between 244 and 1838 m. *Miliammina earlandi* also occurs in Patagonian fjords, although it is not common (Hromic et al., 2006). Assuming that these records refer to a single species, *M. earlandi* is clearly widely distributed both geographically and bathymetrically in the Southern Ocean and adjacent areas, suggesting that it can tolerate a wide range of conditions.

Adaptability to different habitats seems to be a characteristic of *Miliammina* species more generally. Together with other

members of the Rzehakinidae, they inhabit a wide spectrum of both open- and marginal-marine environments (Habura et al., 2006). They are common in salt marshes, mangrove swamps, oligohaline estuaries (Sen Gupta, 1999), persist longer than other foraminifera in marine basins isolated from the sea (Lloyd and Evans, 2002), and dominated in a volcanic caldera less than a decade after significant eruptions (Finger and Lipps, 1981). *Miliammina* species have been also reported from mesohaline habitats (Hayward and Hollis, 1994; Lloyd and Evans, 2002; Habura et al., 2006). In Antarctic waters, *M.*

*arenacea* is associated with corrosive High Salinity Shelf Water (Kennett, 1968; Milam and Anderson, 1981; Ishman and Sperling, 2002) and thrives in areas affected by Antarctic polynya (Capotondi et al., 2018). In SG, highly saline bottom waters were not detected during the present study (Fig. 2) or previous SG surveys (Römer et al., 2014; Geprägs et al., 2016). Instead, the *M. earlandi* FA tends to be associated with rather low salinity (Fig. 10) and its distribution is not restricted to deeper stations (Fig. 9).

This adaptability could confer advantages for *M. earlandi* over other foraminifera at our stations in SG fjords. This may be the last species with a robust test to withstand elevated sediment accumulation rates in fjord heads. It seems to survive anoxia, be able to exploit different food sources, and at the shallow-water stations (SG-08, SG-12, and SG-13), which are at the lower limit of Local Water influence (Fig. 2), it might experience changes in water salinity and temperature.

It is also worth noting that the *M. earlandi* FA includes different auxiliary species in different areas. In Cumberland West

Bay, these are mainly the agglutinated *Psammosphaera fusca* and *Hippocrepinella hirudinea*, and in Cumberland East the calcareous *Gordiospira fragilis* and *Pyrgo patagonica* (Fig. 8b). This difference may be related to different glacial regimes in these two branches of Cumberland Bay (Gordon et al., 2008).

### 4.2.2 Mid-fjord settings: transitional conditions

Stations bordering those dominated by the *M. earlandi* FA in the inner reaches of Cumberland Bay (SG-14 and SG-32) show low to intermediate foraminiferal abundance and diversity (Fig. 6) and a dominance of the *C.* aff. *parkerianus* FA. This is the most widespread FA in our dataset (Fig. 9) and also the most enigmatic. It shows the weakest relation with the main axis 1 on the CCA plot and does not reveal a straightforward correlation with any of the environmental parameters measured in our study (Fig. 10). The *C.* aff. *parkerianus* FA possibly shows some association with limited water depths and water properties corresponding with Antarctic Surface Waters. However, it is less important near fjord mouths, where this water mass predominates (Figs. 1 and 2), therefore an explicit correlation of this FA with Antarctic Surface Waters is unlikely. The subordinate, porous morphotype of *C.* aff. *parkerianus* (Majewski et al., 2021) is rare or absent far inside Cumberland Bay and in the central part of its western branch (Fig. 9), suggesting a preference towards more open marine conditions. *Cassidulinoides* aff. *parkerianus* is exceptionally well represented throughout Cumberland Bay but not in the innermost part, where the *M. earlandii* FA dominates, nor at the deepest stations dominated by the *A. rostratus* FA or around fjord mouths, where the *G.* aff. *rossensis* FA predominates. Its strongest presence is in transitional locations between those characterised by these three FAs, so it seems to flourish where conditions are not optimal for other taxa. The abundance of the smoothly-walled conical morphotype (Majewski et al., 2021) at stations SG-14 and SG-32, adjacent to the most glacier-proximal stations dominated by the *M. earlandi* FA, suggests that *C.* aff. *parkerianus* FA and its associated species (Fig. 8b) are well adapted to inner/middle fjord conditions characterized by moderate sedimentation rates. This would be unique among *Cassidulinoides* species, which are generally rare or absent in similar settings in the Arctic (Alve et al., 2016), Antarctic (Majewski, 2005) and Patagonia (Hromic et al. 2006). The strong presence of this FA in Jason Harbour, within Cumberland West Bay (SG-08, SG-09, SG-10), suggests an association with the finest-grained sediments (Fig. 3b). The TOC/TS ratios at stations dominated by the *C.* aff. *parkerianus* FA cover a wide range (1.8–6.9), consistent with hypoxic to fully oxic conditions (Berner, 1983), and no strong preference for any particular redox conditions.

### 4.2.3 Outer-fjord: weak glacial influence, low accumulation rate, relation to bathymetry and water masses

The outer-fjord areas are usually characterized by the lowest water turbidity and sedimentation rates (Syvitski, 1989), which in the outer Cumberland Bay are in the order of 0.1 to 0.2 cm/yr (Berg et al., 2019, 2020; Graham et al., 2017). Our data show variable foraminiferal abundances, the highest between 100 and 150 mwd and some of the lowest around 250 mwd, with high Shannon diversity at the deepest stations (Fig. 6). The *G.* aff. *rossensis* and *A. rostratus* FAs dominating in outer-fjords, are located on the side of axis 1 opposite to the *M. earlandi* FA in the CCA plot (Fig. 10). They correlate with (1) decreased distance to the open sea and increased distance from a major sediment source, (2) less negative $\delta^{13}$C, and (3) high TOC and low TS, thus high TOC/TS ratios indicating oxic conditions (Berner, 1983). The *A. rostratus* FA seems to be associated with greater water depths and particularly with elevated salinity (Fig. 10).

The *A. rostratus* FA is the most species-rich of the four FAs (Fig. 8b). It is abundant throughout Stromness Bay at >100 mwd and at the deepest stations in the middle and outer Cumberland Bay (Fig. 9). Its nominative species was described in SG from water-depth considerably >200 m (Heron-Allen and Earland, 1929) and seems endemic to this area (Dejardin et al., 2018). It is less dominant than the nominative species of other FAs and *R. subfusiformis* and *A. echolsi* are almost equally important for defining this FA (Table D1). According to our CCA, This FA is strongly correlated with elevated salinity (Fig. 10), pointing to a possible association with Winter Water and Upper Circumpolar Deep Water. This would be consistent with observations from the Northern Hemisphere, where *R. subfusiformis* was recorded from locations with high and stable salinities (Höglund, 1947; Thiede et al., 1981; Jernas et al., 2018) and seemed to be more tolerant of a lower quality or quantity of food (Jernas et al., 2018). Moreover, the calcareous species *A. echolsi* and *Bulimina aculeata,* also important for this FA (Fig. 8b), are typical for open marine assemblages in West Antarctica and show an affinity with highly saline Circumpolar Deep Water (Ishman and Domack, 1994, Majewski et al., 2016) that are the least cold (Mackensen, 2001) and one of the oldest (Catalá et al., 2015) water masses in that area.

By contrast with the apparent association of *A. rostratus* FA with Winter Water and Upper Circumpolar Deep Water, the distribution of the *G.* aff. *rossensis* FA (Fig. 9) may coincide with the upper and warmer Antarctic Surface Waters, which dominates down to 100–150 mwd in Cumberland Bay and throughout the outer Stromness Bay (Fig. 2). Correlation with high TOC/TS ratios indicates oxic conditions (Berner, 1983). However, it is important to note that the *G.* aff. *rossensis* FA is abundant only in the central part of Cumberland East Bay and not in Cumberland West Bay (Fig. 9). The front of the Nordenskjöld Glacier is grounded at a relatively shallow water depth of 20–30 m (Hodgson et al., 2014), and its position is relatively stable (Gordon et al., 2008). It is therefore likely delivering less turbid meltwater to Cumberland East Bay than the rapidly retreating Neumayer Glacier and several land-terminating glaciers in the Cumberland West Bay. As a result, the central part of the eastern branch of Cumberland Bay is less glacially affected, promoting a FA dominated by *G.* aff. *rossensis* rather than by *C.* aff. *parkerianus*. Thus, the *G.* aff. *rossensis* FA appears to flourish in the presence of well-oxygenated, clearer water and possibly lower sediment accumulation rates.

The presence of *G.* aff. *rossensis* near fjord mouths at locations distant to glacial and fluvial sediment sources (Fig. 9) is consistent with the distribution of this species in Chilean fjords, where it is an important component of the oceanic and intermediate biofacies but is insignificant in the inner-fjord biofacies (Hromic et al., 2006). Its Antarctic sister species *Globocassidulina biora* (Finger and Lipps, 1981; Majda et al., 2018), on the other hand, seems to be well adopted to habitats proximal to glacier fronts (Majewski, 2005) and beneath ice-shelves (Majewski et al., 2019). However, West Antarctic glaciers deliver significantly less turbid water than rapidly retreating glaciers in SG (Gordon et al., 2008; Cook et al., 2010), which may be one reason for the different distribution patterns near glacier fronts of these two phylogenetically related species.

**4.4. Regional comparisons**

Foraminifera have been widely studied in fjords, especially in the Arctic (Alve et al. 2016) where there is evidence for strong faunal differences between outer- and inner-fjord locations (e.g. Hald and Korsun, 1997; Korsun and Hald, 1998; Zajączkowski et al., 2010; Fossile et al., 2020) and links with different water-masses (Jennings and Helgadottir, 1994; Jennings et al., 2004). The taxonomic composition of foraminiferal assemblages in Arctic fjords, however, is very different

from that in the Southern Hemisphere. For example, in Svalbard and Novaya Zemlya, *Cribroelphidium/Elphidium excavatum* f. *clavata* thrives close to glacier fronts (Hald and Korsun, 1997; Korsun and Hald, 1998), while in SG we encountered only a few empty tests assigned to this genus, and it is also very rare in Antarctica (Majewski and Tatur, 2009). *Cassidulinoides*, so widespread in SG, seems absent in Arctic fjords (Alve et al., 2016), as are species important for other FAs, with the exception of the cosmopolitan *R. subfusiformis* and *Trifarina earlandi/angulosa* (Alve et al., 2016). These

strong taxonomic differences between the Northern and Southern Hemispheres make direct faunal comparisons impossible. Previous studies of foraminiferal distributions in fjords located in the same sector of the Southern Hemisphere as SG are limited to Patagonia (Hromic et al., 2006; Korsun et al., unpublished) and Admiralty Bay in South Shetlands (Majewski, 2005, 2010; Majewski et al., 2007). These have revealed clear taxonomic disparities between foraminifera inhabiting different sides of the Drake Passage (Majda et al., 2018; Majewski et al., 2021).

The taxonomic succession observed in the fjords of SG corresponds only in part with observations from Patagonia. Hromic et al. (2006) distinguished three benthic foraminiferal biofacies in fjords and channels of southern Chile. The oceanic sandy biofacies, characterized by elevated organic-matter concentrations and strongly influenced by Pacific water, showed high species richness and high abundances of *G. rossensis* and *Trifarina angulosa*. The intermediate fine-grained biofacies with low organic-matter content and influenced by mixed Pacific and fresh water, was dominated by two calcareous species,

*Cassidulina laevigata* and *G. rossensis,* but also included an increased number of agglutinated taxa. The inner-fjord silty biofacies, influenced by cold, low saline water, was characterized by low species diversity and dominated by *Bulimina notovata* and *C. laevigata* (calcareous), together with *Alveophragmium orbiculatum*, *Labrospira kosterensis*, *Recurvoides scitulum*, and *Labrospira jeffreysii* (agglutinated), the last two corresponding to *Veleroninoides scitulus* and *Cribrostomoides jeffreysii,* members of the *A. rostratus* FA in SG.

It appears, therefore, that the dataset of Hromic et al. (2006) only includes representatives of the two outer-fjord associations from SG, the *G.* aff. *rossensis* and *A. rostratus* FAs. Although *M. earlandi,* the nominative taxon of the inner- fjord FA, as well as *C. parkerianus* were encountered in Patagonia, they were not significant species in defining foraminiferal biofacies. The dataset of Hromic et al. (2006), however, did not include the most southerly glaciated fjords in Chilean Patagonia. These were analyzed by Korsun et al. (unpublished), who sampled sea-floor sediments in the Beagle Channel and its tributary

fjords, including glaciomarine muds deposited in direct proximity to glacial fronts (Gschwend et al., 2016). They noted *C. parkerianus* as an important species, but it seems morphologically and genetically different from *C.* aff. *parkerianus* in SG (Majewski et al., 2021). The innermost fjord samples in the Beagle Channel region contained only a few foraminifera ($\sim$1/10cm$^3$), which alternated randomly between stations. There was no distinct glacier-proximal assemblage and *Miliammina*, which is so prominent in the most restricted settings in SG, was absent (Korsun et al., unpublished).

In conclusion, it appears that the faunal change is more pronounced along the axes of sub-Antarctic SG fjords, where the *M. earlandi* and *C.* aff. *parkerianus* FAs are well established, than elsewhere around the Scotia Sea. Moreover, there is a disparity between FAs dominant in inner fjord settings, which are found only in SG, and the more biogeographically widespread assemblages inhabiting outer-fjords and shelf sites (Earland, 1933). This is consistent with the contrast between shallow-water SG macrofaunal communities, which show clear Antarctic characteristics, and the more geographically

widespread macrofauna in surrounding deep waters which do not (Barnes et al., 2006), further emphasizing the exceptional character of the SG biota (Hogg et al., 2011).

## 5. Concluding remarks

As already indicated, benthic foraminifera can serve as valuable proxies for marine environmental conditions recorded in the geological record. It is therefore somewhat surprisingly that no attempts have been ,made to use them in order to reconstruct

coastal environments around sub-Antarctic islands during past climatic oscillations, such as those associated with Quaternary glacial/interglacial cycles. Our results demonstrate that these microfossils have considerable potential in this regard. They seem especially suitable for studying paleoenvironmental changes in the most restricted settings proximal to tidewater glacial fronts in shallow-water settings that are strongly affected by processes taking place on land. These may be rich in organic matter, which is believed to exert a strong control on foraminiferal assemblage composition and diversity.

The sensitivity of foraminifera to environmental changes linked to current and likely future climatic changes are of more immediate interest. Since the 1920s, there has been a 0.9–2.3 °C warming in the top 100 m of the water column around SG (Whitehouse et al. 2008), as well as air temperature increases since the 1950s coinciding with dynamic glacial retreats (Gordon et al., 2008; Cook et al., 2010). The warming around SG and in the Southern Ocean (Meredith and King, 2005) is also associated with the southward migration of the ACC (Gille, 2014) and shifts in the position and intensity of the

Southern Westerly Wind belt (Lamy et al., 2010; Perren et al., 2020). These changes, and their environmental consequences, raise questions regarding the stability of present FA distributions in SG fjords and the possible wider-scale reorganization of biogeographic patterns in the Atlantic sector of the Southern Ocean. There is already a broad overlap between foraminiferal species occurring in the fjords of SG and those located to the north and south of the ACC. Among the 60 species recognized in our study, 27 are shared with Chilean fjords (Hromic et al., 2006; Korsun et al., unpublished) and 31 with Admiralty Bay

in South Shetlands (Majewski, 2005, 2010, Majewski et al., 2007). However, as discussed above, differences currently exist both at species and assemblage levels between foraminiferal faunas in these regions.

Although the data of Earland (1933) cannot be quantitatively compared with the present study due to different methodologies, they do show that all major taxa defining the FAs were present in SG during the 1920s (see also Dejardin et al., 2018). Hence, there seems to be no evidence of recent species invasions. Elsewhere, however, the biogeographic barrier

of the Drake Passage and the ACC (Orsi et al., 1995) have not prevented pulses of faunal migration in and out of Antarctica, especially during past warm periods (Clarke et al., 1992; Majewski et al., 2021). The current warming will likely have major

ecological consequences south of the ACC, especially around the Antarctic Peninsula, exposing this region to alien species (Fraser et al., 2018; Convey and Peck, 2019; Avila et al., 2020). This process has already begun (Fillinger et al., 2013; Griffiths et al., 2013) and may be exacerbated by human activity (Frenot et al., 2005; Hughes et al., 2020). With a warming

climate and accelerating glacial retreats, conditions in South Shetlands and the Antarctic Peninsula will become increasingly temperate, making them more suitable for the development of foraminiferal assemblages currently typical for SG. Warming is also likely to promote natural faunal dispersal between these areas and SG, with intensified ship traffic (McCarthy et al., 2022) possibly providing an additional vector for the rapid introduction of species. Given the potential for major ecological and biogeographic readjustments in this crucial region of the globe, it is clearly important to continue monitoring for

evidence of increasing faunal connectivity across the ACC, as well as between both sides of the Drake Passage and the unique fjord environments of SG.

Appendix A. Environmental parameters and sediment properties discussed in this study and used for the CCA analysis (Fig. 10).

Appendix B1 to B3. Foraminiferal census data and diversity indices for the 63–125 μm and >125 μm fractions as well as the entire assemblage (>63 μm; i.e., the >125 plus 63–125 μm fractions).

Appendix C1 to C6. Scanning electron micrographs of all benthic foraminiferal species encountered in this study.

Appendix D1 to D2. Results of the Q-mode principal component (PC) analysis for the >125 μm and >63 μm datasets.

*Author contribution.* WM and WS designed the research and conducted most of laboratory work. WM, WS and AG did fieldwork. WM drafted the manuscript and prepared the figures. AG and WS substantially revised this work.

*Competing interests.* The authors declare that they have no conflict of interest.

*Acknowledgements.* We kindly thank Keri Pashuk, Greg Landreth, the crew of RSV *Saorise* crew, as well as Maria Holzmann, Jan Pawlowski, and Piotr Rozwalak for their help during the cruise to South Georgia in November-December 2019. Paul Brickle (SAERI), Alistair Crame (BAS), and Robert Bialik (IBB PAS) are thanked for help in organizing the fieldwork. Karolina Leszczyńska kindly helped with grain size analyses. The study was funded by Polish National Science Centre grant No. 2018/31/B/ST10/02886. We also thank the reviewers for their constructive comments that helped to

improve this paper.

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

Table 1. List of stations investigated for this study. Sedimentary indices include grain size, total carbon, total organic carbon, total sulfur, and $\delta^{13}C$ of bulk organic matter indicated in Table A.

| Sample ID | Latitude (54°S) | Longitude (36°W) | Water depth (m) | Date of 2019 | CTD | Van Veen grab | Sedimentary indices |
|---|---|---|---|---|---|---|---|
| SG-01 | 06.200' | 58.800' | 111 | 26 Nov | v | v | v |
| SG-03 | 04.250' | 56.812' | 250 | 27 Nov | v | v | v |
| SG-04 | 07.658' | 47.598' | 135 | 27 Nov | v | v | v |
| SG-06 | 17.299' | 28.247' | 121 | 27 Nov | v | v | v |
| SG-07 | 16.052' | 26.227' | 250 | 29 Nov | v | v | v |
| SG-08 | 12.038' | 34.267' | 23 | 29 Nov | v | v | v |
| SG-09 | 12.044' | 34.253' | 60 | 29 Nov | | v | v |
| SG-10 | 12.759' | 33.677' | 114 | 29 Nov | v | v | v |
| SG-11 | 13.593' | 33.525' | 240 | 29 Nov | v | v | v |
| SG-12 | 16.980' | 29.977' | 21 | 30 Nov | v | v | v |
| SG-13 | 21.936' | 22.486' | 28 | 30 Nov | v | v | v |
| SG-14 | 20.851' | 22.749' | 155 | 30 Nov | v | v | v |
| SG-15 | Sandbugten | | 51 | 30 Nov | | v | v |
| SG-16 | 21.181' | 22.948' | 136 | 1 Dec | | v | v |
| SG-17 | 09.533' | 41.583' | 91.4 | 2 Dec | v | v | v |

| SG-18 | 10.783' | 39.979' | 102 | 2 Dec | v | v | v |
|---|---|---|---|---|---|---|---|
| SG-19 | 08.418' | 40.862' | 48 | 2 Dec | v | | |
| SG-20A | 13.536' | 24.792' | 144 | 3 Dec | | v | v |
| SG-20B | 13.586' | 25.234' | 106 | 4 Dec | | v | |
| SG-21 | 12.900' | 26.963' | 252 | 4 Dec | v | v | v |
| SG-22 | 14.744' | 30.395' | 45 | 4 Dec | v | v | |
| SG-23 | 14.321' | 47.206' | 202 | 4 Dec | v | | |
| SG-24 | 14.653' | 42.150' | 190 | 4 Dec | v | v | v |
| SG-25 | 15.181' | 39.446' | 170 | 4 Dec | v | | |
| SG-26A | 08.542' | 36.592' | 155 | 5 Dec | v | v | v |
| SG-26B | 08.467' | 36.550' | 123 | 5 Dec | | v | |
| SG-27 | 09.372' | 38.426' | 136 | 5 Dec | | v | v |
| SG-28 | 09.861' | 40.175' | 37 | 5 Dec | | v | v |
| SG-29 | 10.148' | 37.022' | 46 | 5 Dec | | v | v |
| SG-30 | 08.337' | 37.935' | 60 | 5 Dec | | v | |
| SG-31 | 14.702' | 41.702' | 188 | 6 Dec | | v | v |
| SG-32 | 15.041' | 38.716' | 194 | 6 Dec | | v | v |

Fig. 1. Study area. Upper map (after https://freevectormaps.com) shows the position of South Georgia in the southwestern Atlantic as well as southern Antarctic Circumpolar Current front (SACCF), the polar front (PF), and the subantarctic front (SAF) after Orsi et al. (1995). Lower map (after *South Georgia and The Shackleton Crossing map* 1 : 200 000, published by British Antarctic Survey in 2017 and *Admiralty Chart 3588, Approaches to Stromness and Cumberland Bays*, 1 : 50 000, second edition, 23rd Jan 2003) shows locations of sampling stations and those CTD stations which data profiles are shown on Fig. 2.

Fig. 2. Selected salinity and temperature profiles from Antarctic, Fortuna, Cumberland, and Stromness bays in South Georgia. Locations of the CTD profiles are indicated on Fig. 1 and in Table 1.

Fig. 3. Sediment mean grain-size vs. sorting expressed in phi scale (a) and the same parameters in relation with distance from fjord mouth (b and c). Sediment categories are after Blott and Pye (2001). Note trend line to coarser and less sorted sediments in (a), two coarsening trends towards fjord mouth and glacier front in (b), and presence of generally better sorted sediments in samples collected between 7 and 925  15 km away from fjord mouth in (c).

Fig. 4. Total organic carbon (TOC) and total sulfur (TS) in relation with distance from fjord mouth.

Fig. 5. Total organic carbon to total sulfur ratio (TOC/TS) (a), carbon stable isotopes of bulk organic matter in the sediments (b) in relation to distance from fjord mouth, and relation between TOC and the $\delta^{13}C$ values (c). Note different trends/mixing lines in (a): towards

increasing TOC/TS ratios in Stromness Bay and towards lower ratios in Cumberland Bay, and in (b): towards less negative $\delta^{13}C$ coves for

Stromness Bay and coves and towards more negative values for the main basins of Cumberland Bay affected by tidewater glaciers. Dashed TOC/TS lines in (a) are after Berner (1983). They demarcate TOC/TS ratios suggested for anoxic (<1.5), periodically anoxic (1.5–5), and oxic conditions (ratios >5). Trends in (b) may represent the progressive mixing of different types of bulk organic matter, namely (1) material typical for open-marine conditions ($\delta^{13}C$ ~–24‰), (2) likely petrogenic organic carbon supplied by glaciers ($\delta^{13}C$ ~–26‰), and (3) organic matter derived from fresh terrestrial and marine sources ($\delta^{13}C$ ~–23‰).

Fig. 6. General indices of benthic foraminiferal assemblages plotted against bathymetry and distance to open sea along fjord axes. Different colors represent locations in various fjords. Color lines show simplified bathymetric profiles for Stromness Bay, Cumberland East Bay, and Cumberland West Bay. Colored circles are for the >125 μm fraction, white circles in two upper graphs are for the >63 μm fraction. Schematic profiles of the Nordenskjöld and Neumayer glaciers marked in bright and dark grey, respectively.

Fig. 7. Abundances in numbers per sample of stained foraminifera in two short cores taken at stations SG-21 and SG-27. Note different
scales for specific and total abundances.

Fig. 8. Plots of the factor loadings for the P-mode PC analysis, showing distribution of different foraminifera species for the >63 μm (a) and >125 μm (b) datasets. Important species for each foraminiferal assemblage (FA), encircled, defined by the Q-mode PC analysis (Table D1) are in bold, the dominant species are underlined. The PC model was based on relative abundances of the more important species, i.e. >1% of total assemblage in at least one sample. Important species for each FA are in bold, dominant species are underlined. Note much
better resolution of the four FAs for the coarser fraction.

Fig. 9. Cumulative percentages of the four species groups (FAs) defined by the PC analysis in relation to the entire >125 μm assemblages plotted against bathymetry and distance along fjord axis to its mouth. Smaller circles indicate percentages of the nominative species and accumulative percentage of *A. rostratus*, *R. subfusiformis* and *A. echolsi* for the *A. rostratus* FA. All graphs at the same scale. Different colors represent locations in various fjords. Color lines show simplified bathymetric profiles for Stromness Bay, Cumberland East Bay,
and Cumberland West Bay. Schematic profiles of the Nordenskjöld and Neumayer glaciers marked in bright and dark grey, respectively.

Fig. 10. CCA plot of the faunal assemblages (FAs) defined by PCA (in green) with various parameters (in black), including: location (water depth, distance from major sediment source and distance to open sea), environmental parameters (salinity and temperature), and sediment properties (total organic carbon (TOC) and its isotopic composition, total sulfur (TS), mean grain size and sorting).

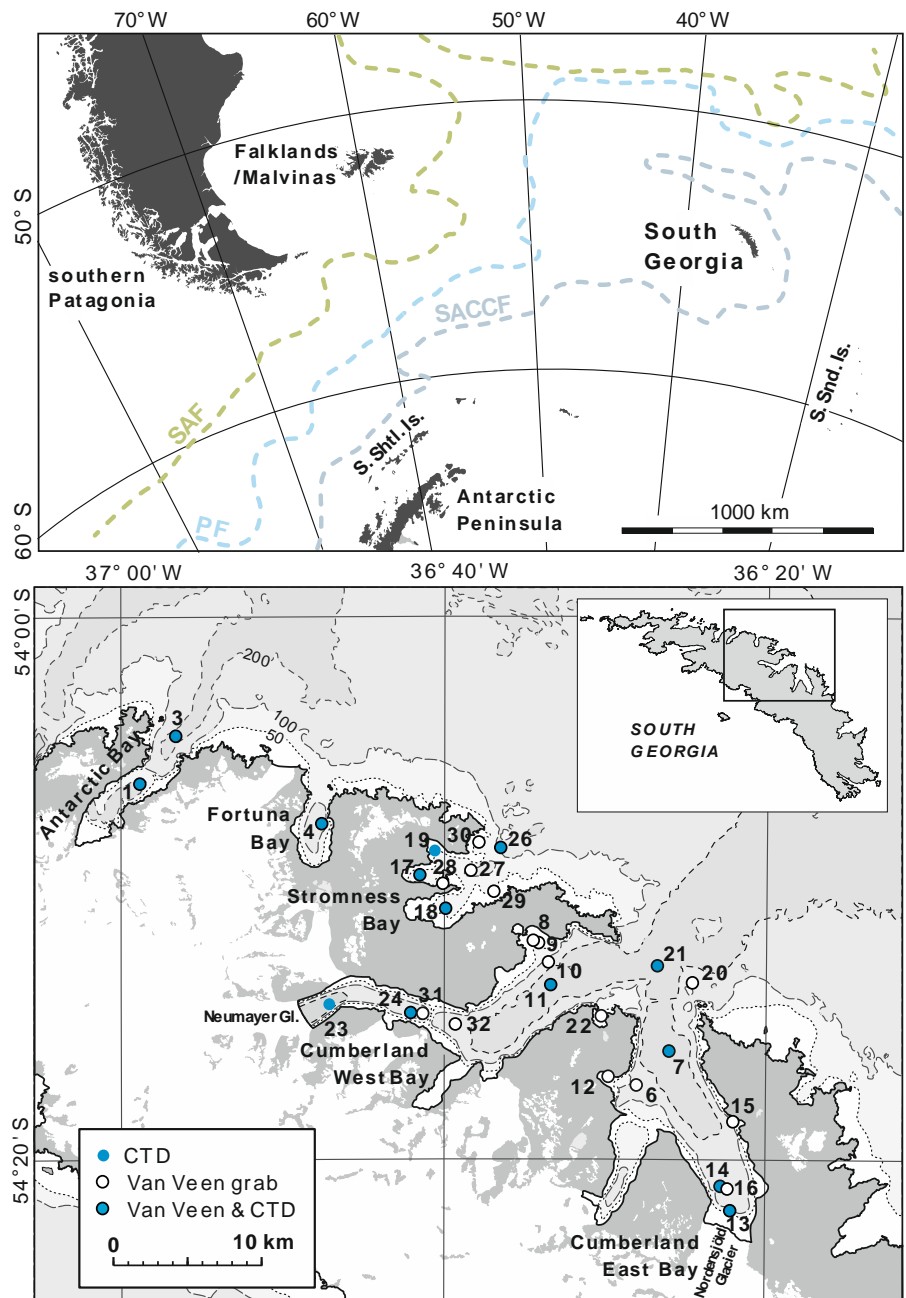

**Fig. 1. Study area.** Upper map (after https://freevectormaps.com) shows the position of South Georgia in the southwestern Atlantic as well as southern Antarctic Circumpolar Current front (SACCF), the polar front (PF), and the subantarctic front (SAF) after Orsi et al. (1995). Lower map (after *South Georgia and The Shackleton Crossing map* 1 : 200 000, published by British Antarctic Survey in 2017 and *Admiralty Chart 3588, Approaches to Stromness and Cumberland Bays*, 1 : 50 000, second edition, 23$^{rd}$ Jan 2003) shows locations of sampling stations and those CTD stations which data profiles are shown on Fig. 2.

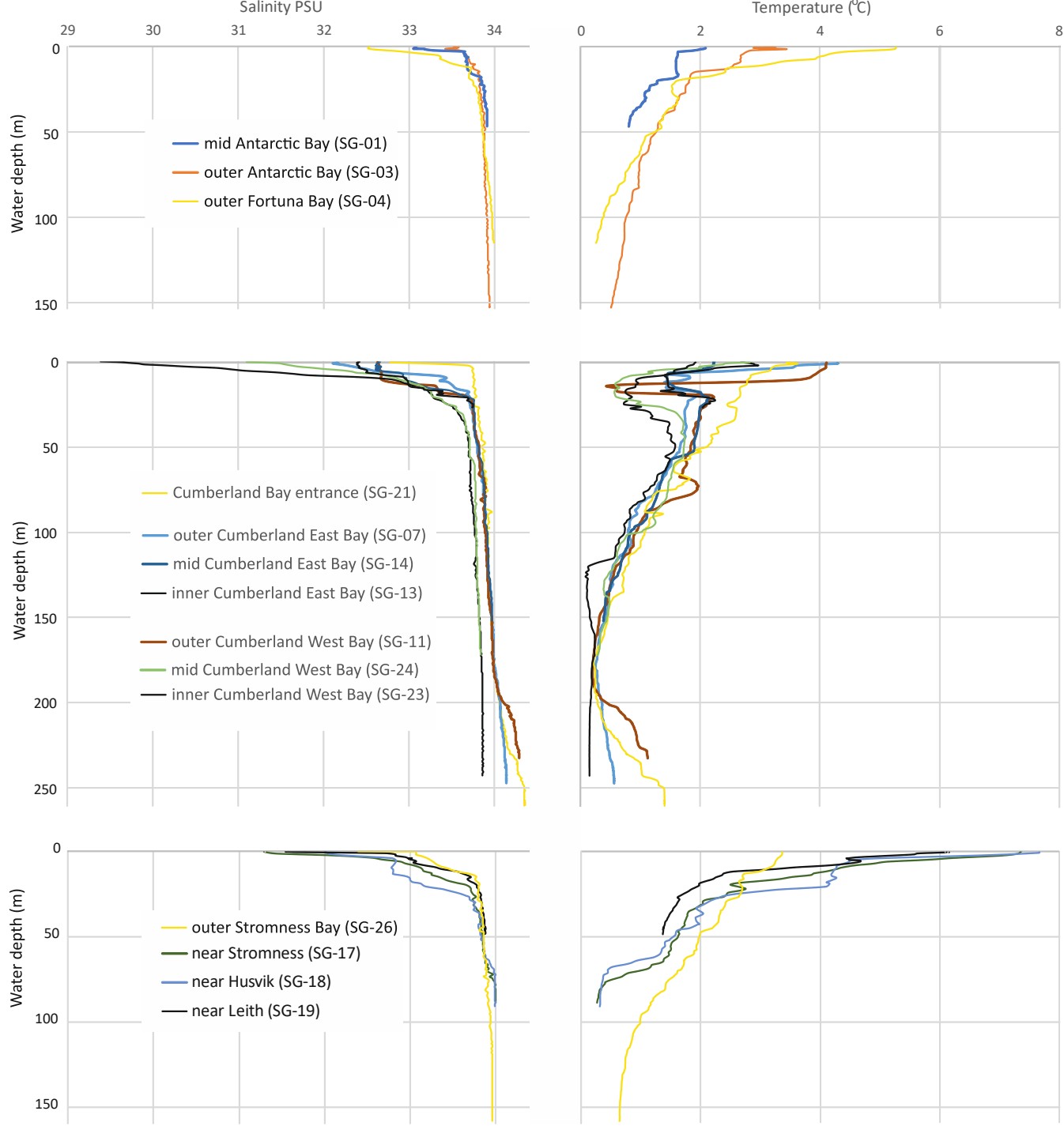

**Fig. 2. Selected salinity and temperature profiles from Antarctic, Fortuna, Cumberland, and Stromness bays in South Georgia. Locations of the CTD profiles are indicated on Fig. 1 and in Table 1.**

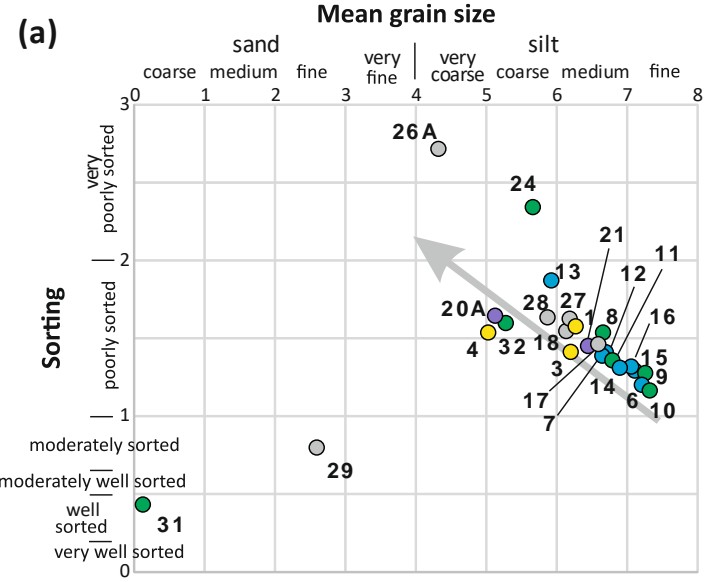

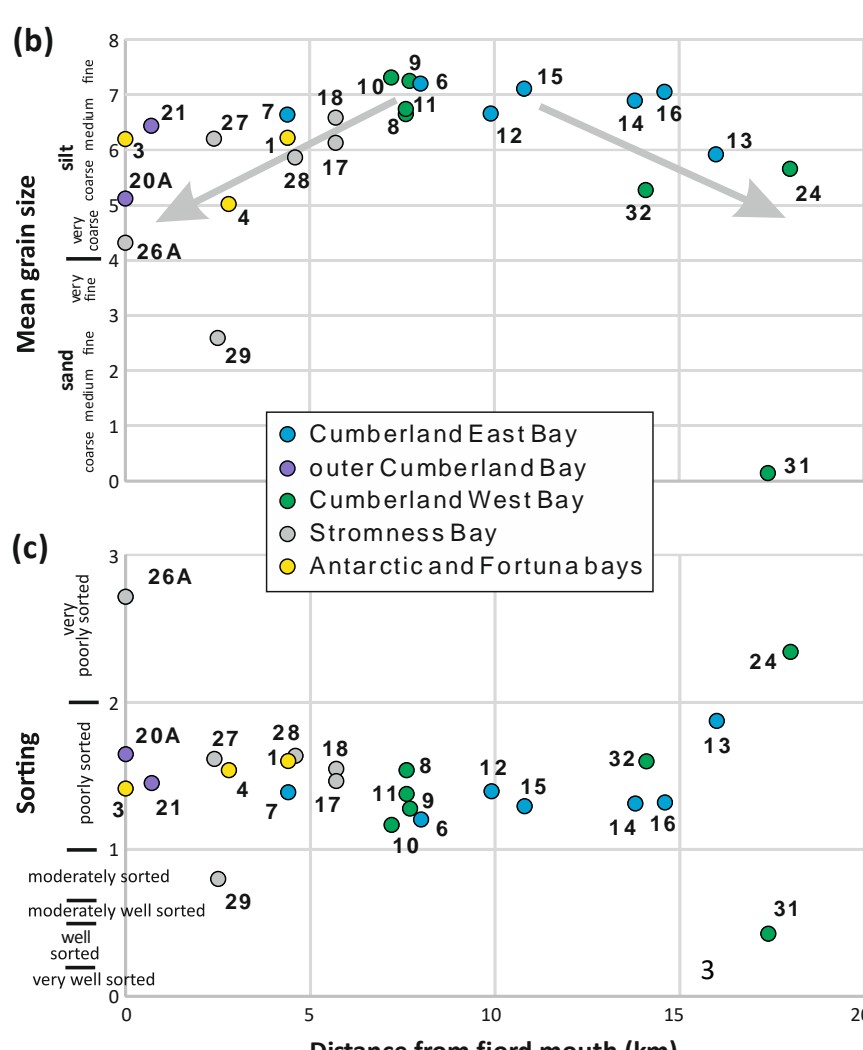

**Fig. 3. Sediment mean grain-size vs. sorting expressed in phi scale (a) and the same parameters in relation with distance from fjord mouth (b and c). Sediment categories are after Blott and Pye (2001). Note trend line to coarser and less sorted sediments in (a), two coarsening trends towards fjord mouth and glacier front in (b), and presence of generally better sorted sediments in samples collected between 7 and 15 km away from fjord mouth in (c).**

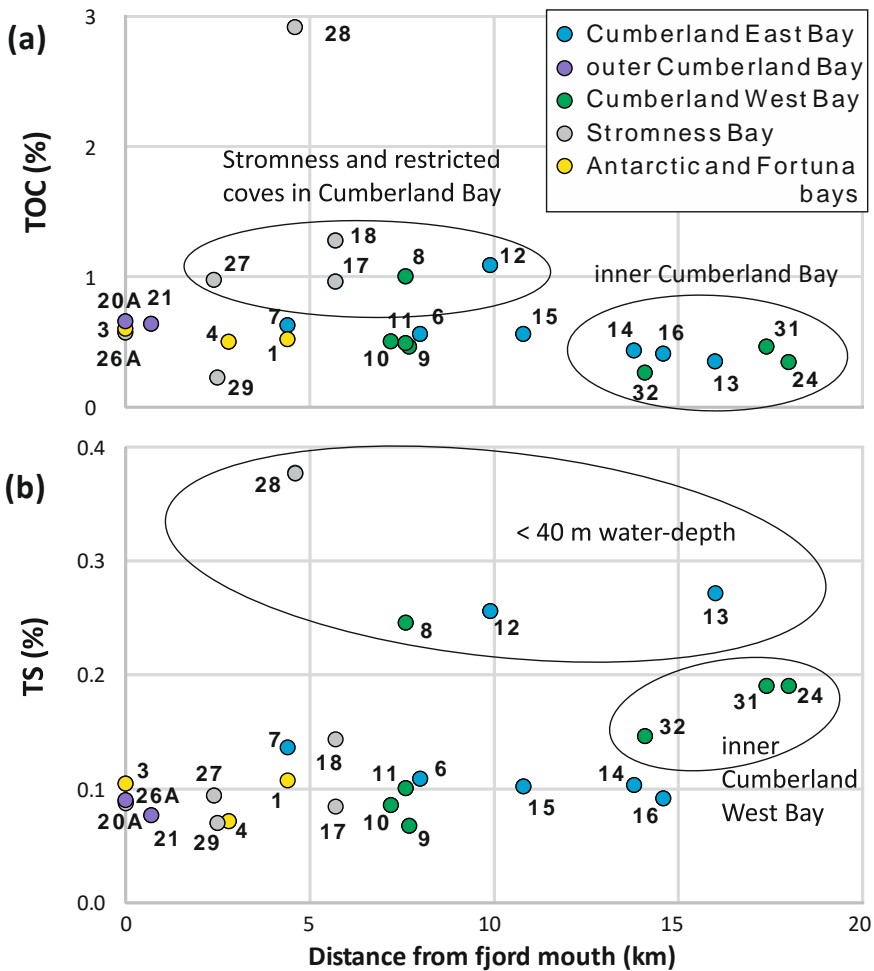

**Fig. 4. Total organic carbon (TOC) and total sulfur (TS) in relation with distance from fjord mouth.**

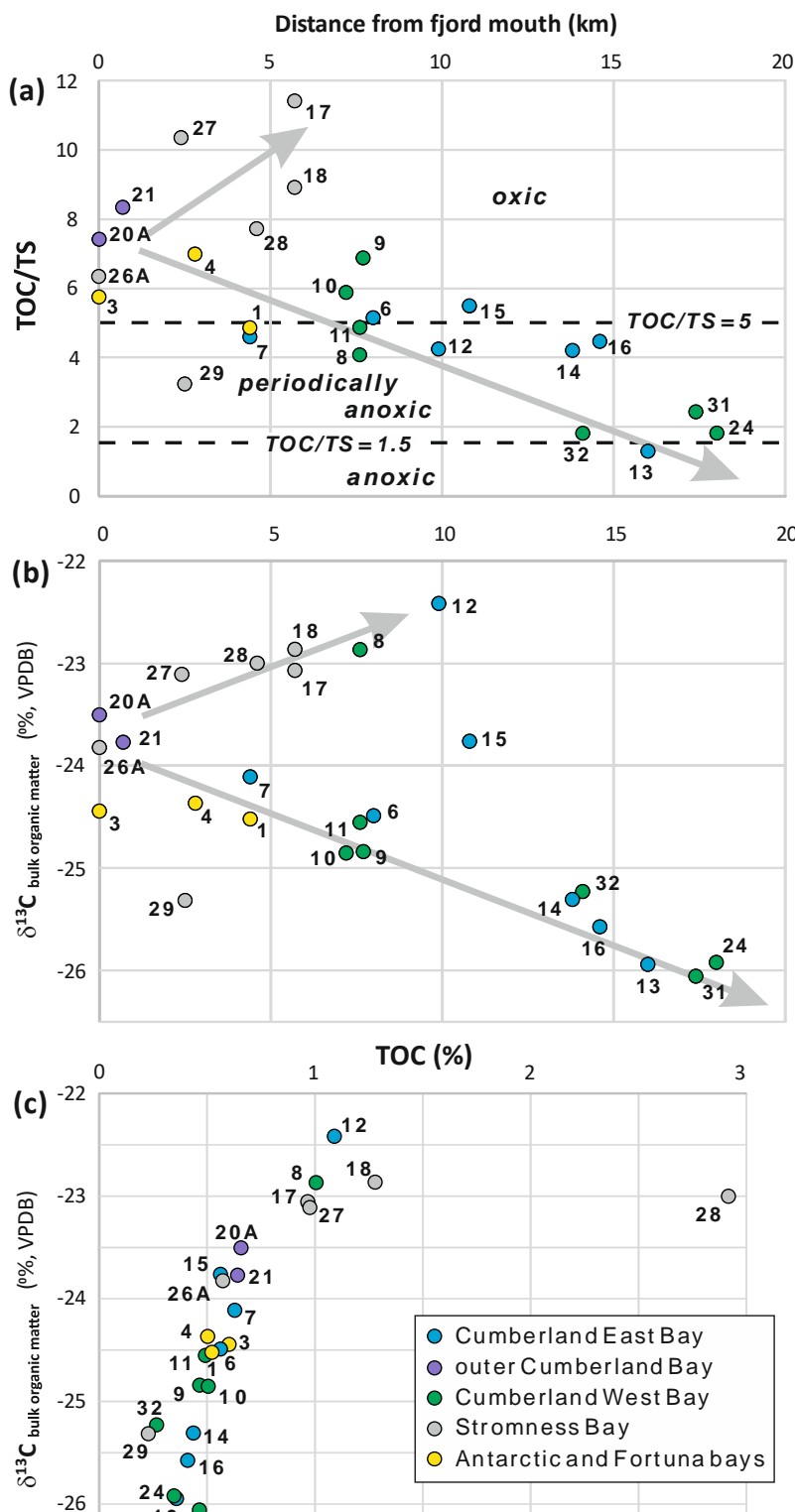

Fig. 5. Total organic carbon to total sulfur ratio (TOC/TS) (a), carbon stable isotopes of bulk organic matter in the sediments (b) in relation to distance from fjord mouth, and relation between TOC and the $\delta^{13}C$ values (c). Note different trends/mixing lines in (a): towards increasing TOC/TS ratios in Stromness Bay and towards lower ratios in Cumberland Bay, and in (b): towards less negative $\delta^{13}C$ coves for Stromness Bay and coves and towards more negative values for the main basins of Cumberland Bay affected by tidewater glaciers. Dashed TOC/TS lines in (a) are after Berner (1983). They demarcate TOC/TS ratios suggested for anoxic (<1.5), periodically anoxic (1.5–5), and oxic conditions (ratios >5). Trends in (b) may represent the progressive mixing of different types of bulk organic matter, namely (1) material typical for open-marine conditions ($\delta^{13}C$ ~–24‰), (2) likely petrogenic organic carbon supplied by glaciers ($\delta^{13}C$ ~–26‰), and (3) organic matter derived from fresh terrestrial and marine sources ($\delta^{13}C$ ~–23‰).

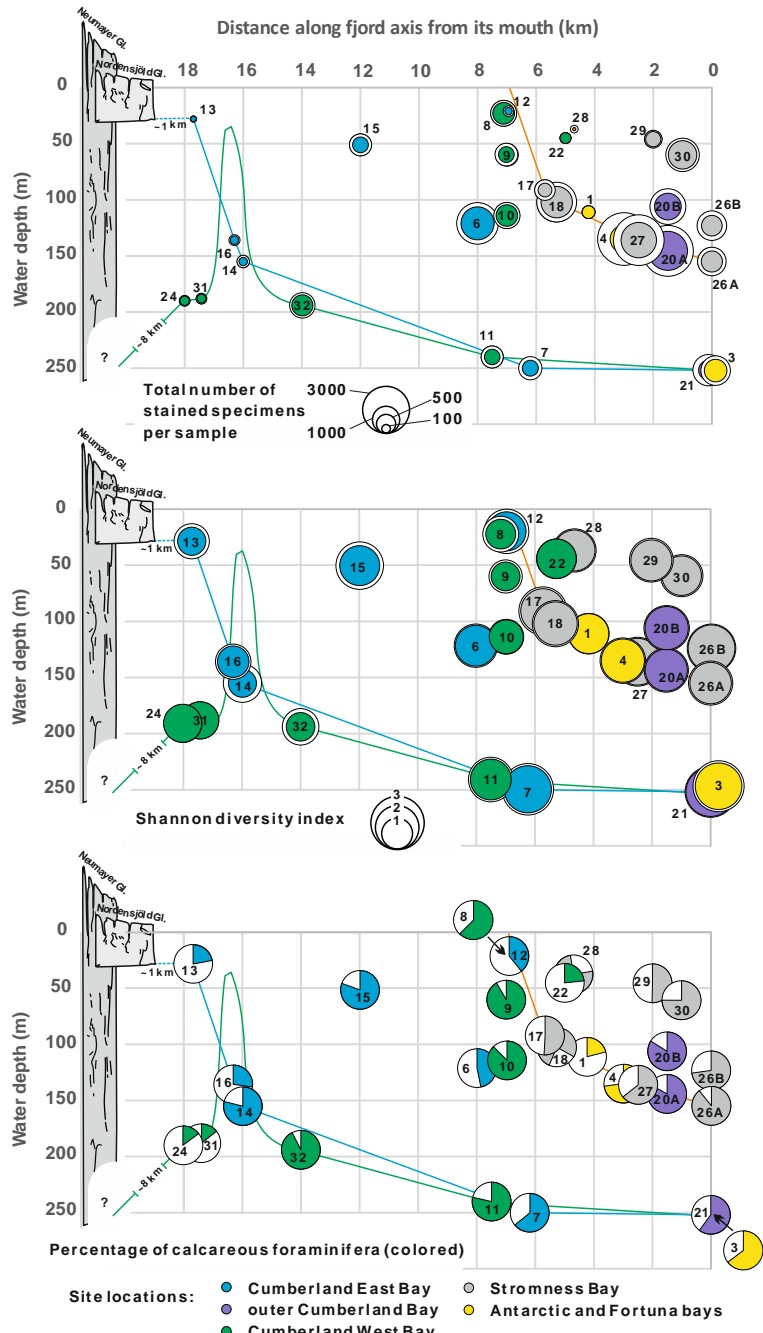

Fig. 6. General indices of benthic foraminiferal assemblages plotted against bathymetry and distance to open sea along fjord axes. Different colors represent locations in various fjords. Color lines show simplified bathymetric profiles for Stromness Bay, Cumberland East Bay, and Cumberland West Bay. Colored circles are for the >125 μm fraction, white circles in two upper graphs are for the >63 μm fraction. Schematic profiles of the Nordenskjöld and Neumayer glaciers marked in bright and dark grey, respectively.

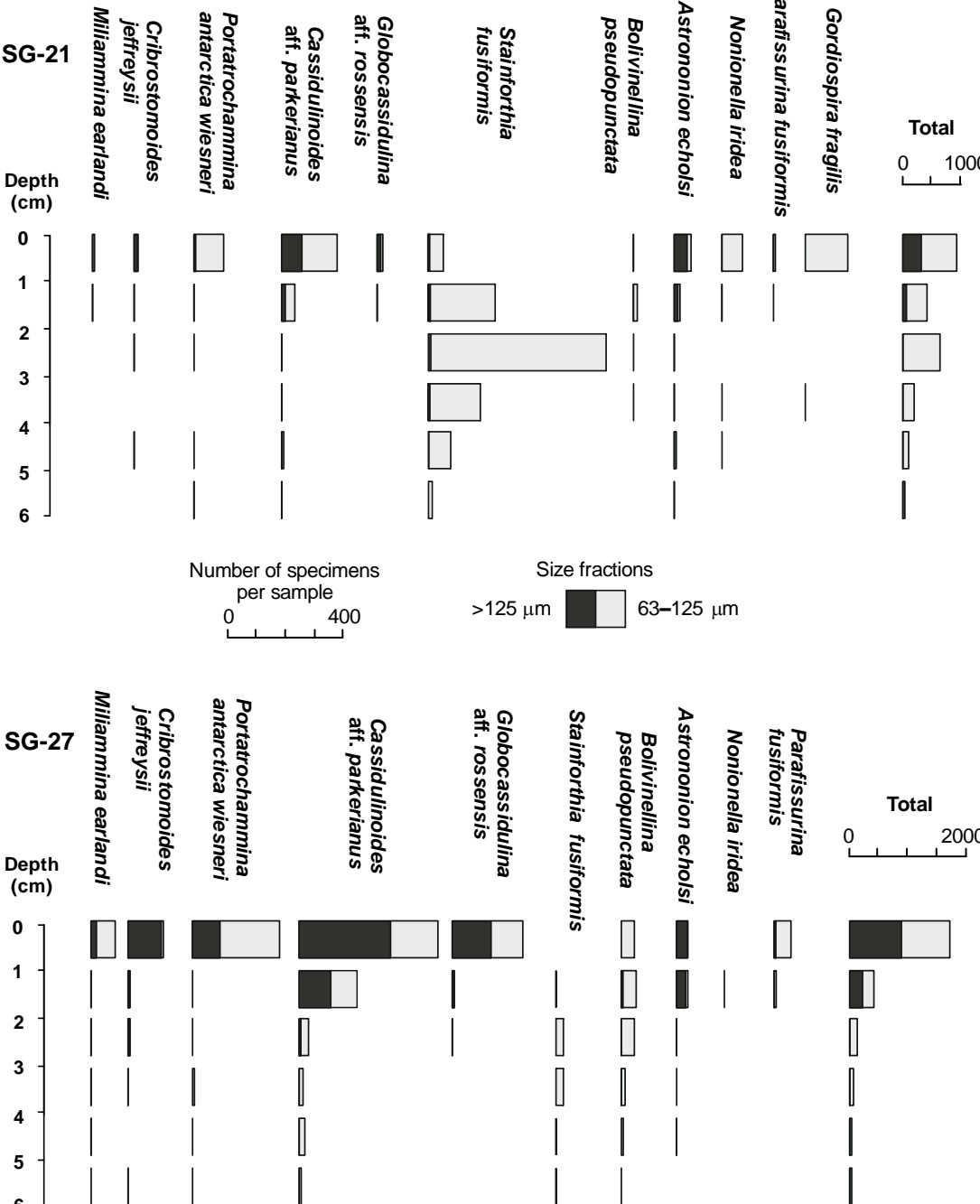

Fig. 7. Abundances in numbers per sample of stained foraminifera in two short cores taken at stations SG-21 and SG-27. Note different

35   scales for specific and total abundances.

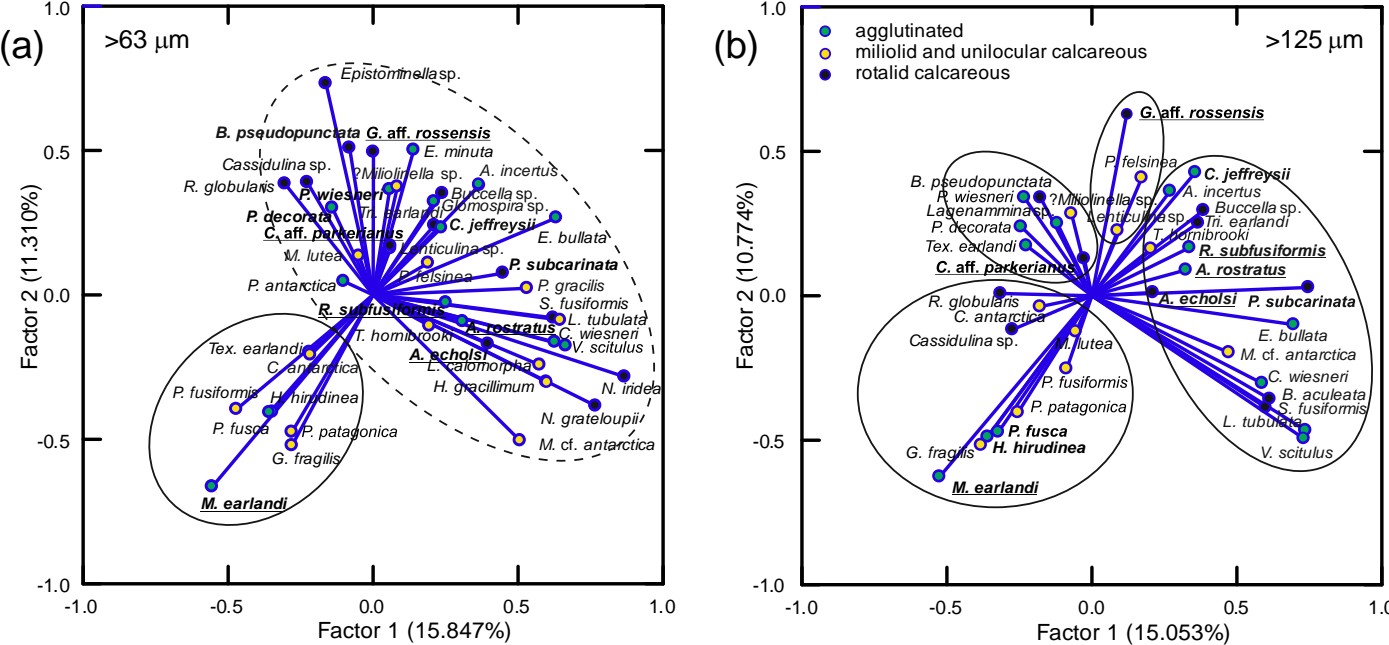

**Fig. 8. Plots of the factor loadings for the P-mode PC analysis, showing distribution of different foraminifera species for the >63 μm (a) and >125 μm (b) datasets. Important species for each foraminiferal assemblage (FA), encircled, defined by the Q-mode PC analysis (Table D1) are in bold, the dominant species are underlined. The PC model was based on relative abundances of the more important species, i.e. >1% of total assemblage in at least one sample. Important species for each FA are in bold, dominant species are underlined. Note much better resolution of the four FAs for the coarser fraction.**

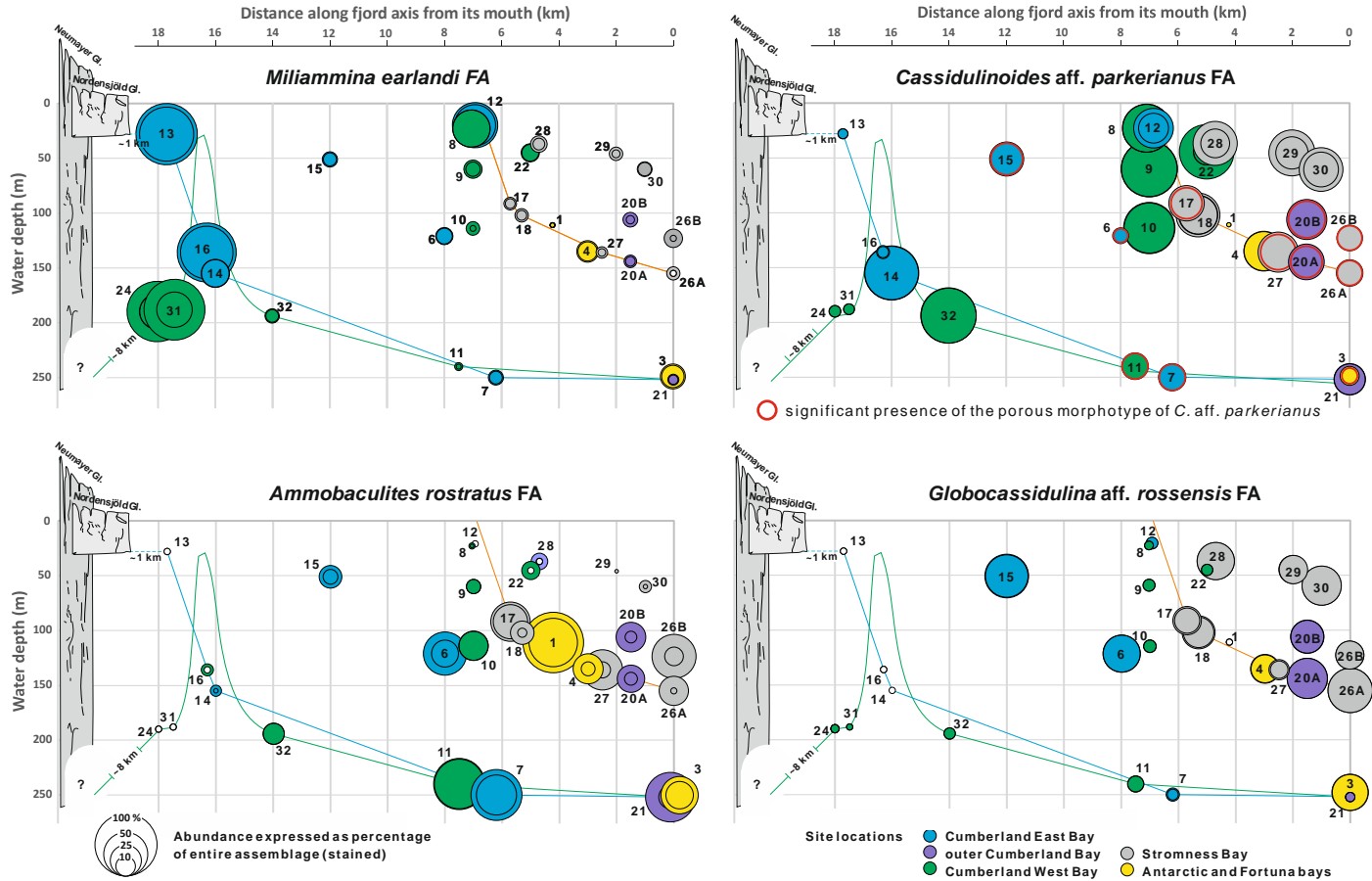

**Fig. 9.** Cumulative percentages of the four species groups (FAs) defined by the PC analysis in relation to the entire >125 μm assemblages plotted against bathymetry and distance along fjord axis to its mouth. Smaller circles indicate percentages of the nominative species and accumulative percentage of *A. rostratus*, *R. subfusiformis* and *A. echolsi* for the *A. rostratus* FA. All graphs at the same scale. Different colors represent locations in various fjords. Color lines show simplified bathymetric profiles for Stromness Bay, Cumberland East Bay, and Cumberland West Bay. Schematic profiles of the Nordenskjöld and Neumayer glaciers marked in bright and dark grey, respectively.

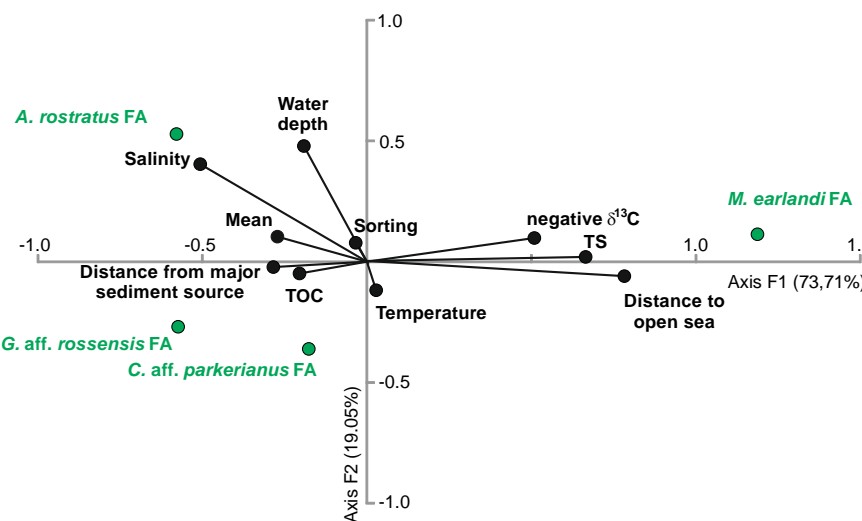

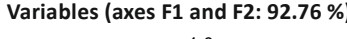

**Fig. 10. CCA plot of the faunal assemblages (FAs) defined by PCA (in green) with various parameters (in black), including: location (water depth, distance from major sediment source and distance to open sea), environmental parameters (salinity and temperature), and sediment properties (total organic carbon (TOC) and its isotopic composition, total sulfur (TS), mean grain size and sorting).**

Table A. Environmental parameters and sediment properties discussed in this study and used for the CCA analysis (Fig. 10).

| Station | Water depth (m) | Distance from major sediment source (km) | Distance from fjord's mouth (km) | Mean grain size (phi) | Sorting (phi) | Temp. (°C) | Salinity (PSU) | TS (%) | TOC (%) | $\delta^{13}C_{sed}$ (‰) |
|---|---|---|---|---|---|---|---|---|---|---|
| SG-01 | 111 | 4.6 | 4.4 | 6.22 | 1.60 | 0.85 | 33.9 | 0.11 | 0.52 | -24.52 |
| SG-03 | 250 | 10.0 | 0.0 | 6.20 | 1.41 | 0.5 | 34.35 | 0.10 | 0.60 | -24.45 |
| SG-04 | 135 | 2.1 | 2.8 | 5.03 | 1.54 | 0.2* | 33.9* | 0.07 | 0.50 | -24.37 |
| SG-06 | 121 | 11.9 | 8.0 | 7.21 | 1.20 | 0.6* | 33.9* | 0.11 | 0.56 | -24.49 |
| SG-07 | 250 | 12.6 | 4.4 | 6.64 | 1.39 | 0.5* | 34.1* | 0.14 | 0.63 | -24.11 |
| SG-08 | 23 | 20.3 | 7.6 | 6.66 | 1.54 | 1.0* | 33.5* | 0.25 | 1.00 | -22.87 |
| SG-09 | 60 | 20.3 | 7.7 | 7.25 | 1.28 | 1.6 | 33.8 | 0.07 | 0.47 | -24.84 |
| SG-10 | 114 | 19.1 | 7.2 | 7.32 | 1.16 | 0.6* | 33.9* | 0.09 | 0.50 | -24.85 |
| SG-11 | 240 | 18.4 | 7.6 | 6.74 | 1.37 | 1.1* | 34.3* | 0.10 | 0.49 | -24.55 |
| SG-12 | 21 | 0.6 | 9.9 | 6.66 | 1.39 | 1.3* | 33.7* | 0.26 | 1.09 | -22.42 |
| SG-13 | 28 | 1.2 | 16.0 | 5.92 | 1.87 | 2.0* | 33.7* | 0.27 | 0.35 | -25.94 |
| SG-14 | 155 | 3.2 | 13.8 | 6.89 | 1.31 | 0.4* | 34.0* | 0.10 | 0.44 | -25.31 |
| SG-15 | 51 | 7.5 | 10.8 | 7.11 | 1.29 | 1.6 | 33.8 | 0.10 | 0.56 | -23.76 |
| SG-16 | 136 | 2.6 | 14.6 | 7.06 | 1.32 | 0.4 | 33.9 | 0.09 | 0.41 | -25.57 |
| SG-17 | 92 | 1.1 | 5.7 | 6.59 | 1.46 | 0.3* | 34.0* | 0.08 | 0.96 | -23.07 |
| SG-18 | 102 | 2.8 | 5.7 | 6.13 | 1.55 | 0.3* | 34.0* | 0.14 | 1.28 | -22.87 |
| SG-20A | 144 | 17.8 | 0.0 | 5.12 | 1.64 | 0.5 | 33.9 | 0.09 | 0.66 | -23.51 |
| SG-20B | 106 | 17.8 | 0.0 | | | 1.0* | 33.9* | *no sediment available* | | |
| SG-21 | 252 | 18.7 | 0.7 | 6.44 | 1.45 | 1.4* | 34.3* | 0.08 | 0.64 | -23.77 |
| SG-22 | 45 | 0.8 | 6.5 | | | 2.0* | 33.8* | *no sediment available* | | |
| SG-24 | 190 | 7.9 | 18.0 | 5.66 | 2.34 | 0.3* | 33.8* | 0.19 | 0.35 | -25.92 |
| SG-26A | 155 | 6.9 | 0.0 | 4.32 | 2.72 | 0.6* | 34.0* | 0.09 | 0.57 | -23.82 |
| SG-26B | 123 | 6.9 | 0.0 | | | 0.7 | 34.0 | *no sediment available* | | |
| SG-27 | 136 | 4.6 | 2.4 | 6.21 | 1.61 | 0.7 | 34.0 | 0.09 | 0.98 | -23.11 |
| SG-28 | 37 | 2.7 | 4.6 | 5.87 | 1.63 | 2.3 | 33.8 | 0.38 | 2.92 | -23.00 |
| SG-29 | 46 | 6.2 | 2.5 | 2.59 | 0.80 | 2.1 | 33.8 | 0.07 | 0.23 | -25.32 |
| SG-30 | 60 | 5.9 | 0.8 | | | 1.9 | 33.9 | *no sediment available* | | |
| SG-31 | 188 | 8.5 | 17.4 | 0.14 | 0.42 | 0.3 | 33.8 | 0.19 | 0.46 | -26.06 |
| SG-32 | 194 | 11.8 | 14.1 | 5.28 | 1.60 | 0.3 | 34.0 | 0.15 | 0.27 | -25.23 |

* direct measurement

Table B1. Foraminiferal census data for the 63–125 µm fraction.

| Station | *L. tubulata* | encrusting monothal. | *H. hirudinea* | *P. decorata* | *M. earlandi* | *C. jeffreysii* | *C. wiesneri* | *A. rostratus* | *A. incertus* | *Glomospira* sp. | *P. wiesneri* | *E. bullata* | *Tex. earlandi* | *R. subfusiformis* | *A. glomeratum* | *E. minuta* | *P. antarctica* | other agglutinated | *C.* aff. *parkerianus* | *G.* aff. *rossensis* | *Cassidulina* sp. | *Trif. earlandi* | *B. aculeata* | *S. fusiformis* | *B. pseudopunctata* | *A. echolsi* | *P. subcarinata* | *N. grateloupii* | *N. iridea* | *Epistominella* sp. | *Cibicides* sp. | *R. globularis* | other rotalid | *P. felsinea* | *H. gracillimum* | *P. gracilis* | *P. fusiformis* | other Lagenidae | *G. fragilis* | *C. antarctica* | *L. calomorpha* | *Lenticulina* sp. | *T. hornibrooki* | *M.* cf. *antarctica* | *M. lutea* | *?Miliolinella* sp. | other miliolids | *N. pachyderma* | fraction of sample picked | total picked forams | total foraminifera per sample |
|---|---|---|---|---|---|---|---|---|---|---|---|---|---|---|---|---|---|---|---|---|---|---|---|---|---|---|---|---|---|---|---|---|---|---|---|---|---|---|---|---|---|---|---|---|---|---|---|---|---|---|---|---|
| SG-01 | | | | | | | | | | | 1 | | | | | | | | | 1 | | | | | | | | | | | | 1 | | | | | | | | | | | | | | | | | 1 | 3 | 3,0 |
| SG-03 | | | 5 | | 36 | 2 | | | 2 | | 10 | | 9 | 15 | | | 5 | 2 | 57 | 30 | 8 | | 1 | 23 | 31 | 16 | 4 | 15 | 7 | 1 | | | 4 | 11 | 1 | 4 | 17 | 4 | 35 | 2 | | | | | 1 | | | | 0,75 | 358 | 477,3 |
| SG-04 | | | | | 73 | | | 1 | | | 5 | | 3 | 1 | | | | | 45 | 45 | | | | | 17 | 3 | 3 | | 2 | 2 | | 1 | 21 | | | | 15 | 2 | | | | | 84 | | 4 | | | | 0,125 | 327 | 2616,0 |
| SG-06 | | | | | 38 | 7 | | | | | 28 | | | 10 | | | | | 59 | 123 | 3 | | | | | 1 | | | | 8 | | | | 4 | | | 22 | 8 | 3 | | | | 1 | | | | | | 0,375 | 315 | 840,0 |
| SG-07 | | | | | 8 | 8 | | | 3 | | 6 | 1 | | 6 | | | 5 | | 75 | 4 | 4 | | | 26 | 37 | 19 | 3 | 16 | 44 | | | | | | 11 | 11 | 8 | 16 | 4 | | | | 25 | 29 | 5 | | | | 1 | 374 | 374,0 |
| SG-08 | | | | | 95 | | | | | | 8 | | | | | | 1 | | 65 | 25 | 14 | | | 13 | | | | | | 8 | 1 | 1 | 17 | | | | 9 | 3 | 1 | | | | 1 | | | | | | 1 | 262 | 262,0 |
| SG-09 | | | | | 21 | | | | | | 17 | | | 1 | | | | | 230 | 5 | 9 | | | | 3 | 3 | | | 1 | 7 | 1 | | | | | | 16 | | 1 | | | | | 33 | | | | | 1 | 348 | 348,0 |
| SG-10 | | | | | 8 | | | | | | 22 | | | 3 | | | | | 215 | 20 | 1 | | | | 4 | 4 | | 2 | | | 1 | 1 | | 1 | | | | | 2 | | | | | 5 | 8 | | 1 | | 1 | 298 | 298,0 |
| SG-11 | | | | | 2 | 7 | | | 1 | | 8 | 1 | | 11 | | | 3 | | 110 | 21 | | 1 | 1 | 3 | 8 | 80 | | 6 | 25 | 3 | | 1 | 3 | 8 | 6 | 2 | | 2 | | | 4 | 1 | | | | | 1 | | 1 | 319 | 319,0 |
| SG-12 | | | | | 27 | | | | | | 9 | | | | | | 42 | 1 | 2 | 1 | 32 | | | 2 | 7 | | | | | 5 | | 2 | | | | | 9 | | 1 | | | | | 2 | | | 4 | | 2 | 146 | 73,0 |
| SG-13 | | | | | 9 | | | | | | | | 1 | | | | | | 0 | | | | | | | | 2 | | 1 | | | 1 | | | | | 7 | | | 1 | | | | 2 | | | | | 2 | 24 | 12,0 |
| SG-14 | | | | | 42 | | | | | | 8 | | 39 | 1 | | | | | 74 | 17 | 6 | 1 | | 1 | 3 | | 2 | | | 5 | | | | | | | 11 | 2 | | | | | | | | | | | 2 | 212 | 106,0 |
| SG-15 | | | | | 18 | | | 3 | | | 41 | | 8 | 4 | | 1 | 4 | 1 | 41 | 37 | 15 | | | | 25 | 25 | 2 | | 10 | 2 | | | 35 | | | | 9 | | | 1 | | 19 | 1 | | 13 | | | | 1 | 315 | 315,0 |
| SG-16 | | | | | 71 | | | | | | 1 | | 18 | | | | | 1 | 4 | | 3 | | | | 2 | | | | 1 | | | | | | | | 33 | | | | | | | | | | | | 2 | 134 | 67,0 |
| SG-17 | | | 1 | | 11 | | | | | | 33 | | 11 | 28 | | 3 | 22 | | 56 | 38 | 13 | | | 53 | 3 | 2 | 1 | | 1 | 2 | | | | 9 | 1 | 1 | 13 | | | | | | | | 2 | | | | 1 | 304 | 304,0 |
| SG-18 | | | | | 19 | | | | 2 | | 30 | | 3 | | | 3 | 9 | 1 | 124 | 20 | 8 | | | 22 | 38 | | | | | 11 | | | | | 10 | | 16 | 3 | 4 | | | | | 9 | 9 | | | | 0,4375 | 341 | 779,4 |
| SG-20A | | | | | 8 | 2 | | | 1 | | 31 | 1 | 8 | | | | | 2 | 105 | 100 | 12 | 1 | | 6 | 25 | 3 | 3 | 5 | 7 | 14 | | | | | 1 | 1 | 2 | 1 | | | | | | 6 | | | | | 0,25 | 345 | 1380,0 |
| SG-20B | | | | | 5 | | | | 3 | | 89 | | 6 | 1 | | | | 4 | 115 | 28 | 6 | | | 9 | 17 | 1 | 2 | | 5 | 17 | 1 | | | 4 | | | 9 | 2 | | | | | | 1 | | | | | 0,4375 | 325 | 742,9 |
| SG-21 | | | | | | 6 | 3 | 1 | 3 | | 27 | 4 | 3 | 10 | 1 | 4 | 3 | 5 | 93 | 3 | 2 | | 1 | 66 | 2 | 20 | 4 | 19 | 39 | | | | 2 | | 1 | | 2 | 2 | | | | | 2 | 1 | 3 | 2 | 2 | 2 | 0,4375 | 338 | 772,6 |
| SG-22 | | | | | | | | | | | 2 | | | | | | | | 1 | | | | | | 1 | | | | 1 | | | | | | | | | | | | | | | | | | | | 1 | 5 | 5,0 |
| SG-24 | | | | | 12 | | | | | | | | | | | | | | 1 | | | | | | | | | | | | | | | | | | 4 | | | | | | | | | | | | 0,5 | 17 | 34,0 |
| SG-26A | 1 | | | | 3 | 1 | 1 | | | 1 | 55 | | | 1 | | | 6 | 2 | 139 | 72 | 11 | 1 | | 3 | 5 | | 1 | 13 | 12 | | | | | | | | 18 | 3 | | | | | 1 | 2 | | | | | 0,75 | 352 | 469,3 |
| SG-26B | | | | | 1 | 2 | | | | | 93 | | 1 | | | 1 | 2 | 4 | 62 | 29 | 13 | | | 13 | 2 | 52 | 6 | 9 | 11 | | | | | 1 | 1 | | 1 | 9 | 2 | | | | | | | | | 0,5313 | 315 | 592,9 |
| SG-27 | | | | | 15 | 6 | | | 8 | 12 | 102 | 3 | 3 | | | | 3 | 7 | 101 | 26 | | | | 1 | 14 | 13 | 8 | | 3 | 16 | | | | 5 | | 7 | 22 | 2 | | | | | 2 | 9 | 1 | 3 | 10 | | 0,2344 | 402 | 1715,2 |
| SG-28 | | | | | 2 | | | | | | | | | | | | 3 | 1 | 1 | 2 | 21 | | | 5 | 60 | | | | 14 | 5 | | | | | | | 10 | | | | | | | | 1 | | | | 2 | 125 | 62,5 |
| SG-29 | | 1 | | 1 | 1 | | | | | | 4 | | | | | | 1 | 2 | 10 | 32 | 3 | | | 19 | | | | 2 | 26 | | | | | | | | | | | | | | | | | | | | 1 | 102 | 102,0 |
| SG-30 | | 1 | | 5 | 13 | 5 | | | 2 | | 28 | | | | | | 1 | 5 | 87 | 96 | 6 | | | 25 | | | | | 43 | | 4 | | 1 | | | | 9 | 3 | 1 | | | | 1 | | 1 | | | | 0,8125 | 337 | 414,8 |
| SG-31 | | | | | 36 | | | | | | | | 1 | | | | | | 0 | 3 | | | | | | | | | | | | | | | | | 10 | | | | | | | | | | | | 1 | 50 | 50,0 |
| SG-32 | | | 1 | | 23 | | | | | | 3 | | 12 | 21 | | | | 1 | 89 | 12 | 3 | | | 9 | 1 | 40 | | | | | | 3 | 4 | 2 | | 2 | 10 | 1 | | 2 | | 21 | | | 1 | | | 1 | 1 | 262 | 262,0 |

Cores

| Station | *L. tubulata* | encrusting monothal. | *H. hirudinea* | *P. decorata* | *M. earlandi* | *C. jeffreysii* | *C. wiesneri* | *A. rostratus* | *A. incertus* | *Glomospira* sp. | *P. wiesneri* | *E. bullata* | *Tex. earlandi* | *R. subfusiformis* | *A. glomeratum* | *E. minuta* | *P. antarctica* | other agglutinated | *C.* aff. *parkerianus* | *G.* aff. *rossensis* | *Cassidulina* sp. | *Trif. earlandi* | *B. aculeata* | *S. fusiformis* | *B. pseudopunctata* | *A. echolsi* | *P. subcarinata* | *N. grateloupii* | *N. iridea* | *Epistominella* sp. | *Cibicides* sp. | *R. globularis* | other rotalid | *P. felsinea* | *H. gracillimum* | *P. gracilis* | *P. fusiformis* | other Lagenidae | *G. fragilis* | *C. antarctica* | *L. calomorpha* | *Lenticulina* sp. | *T. hornibrooki* | *M.* cf. *antarctica* | *M. lutea* | *?Miliolinella* sp. | other miliolids | *N. pachyderma* | fraction of sample picked | total picked forams | total foraminifera per sample |
|---|---|---|---|---|---|---|---|---|---|---|---|---|---|---|---|---|---|---|---|---|---|---|---|---|---|---|---|---|---|---|---|---|---|---|---|---|---|---|---|---|---|---|---|---|---|---|---|---|---|---|---|---|
| SG-21, 0-1 cm | 4 | | | | 1 | 1 | 1 | | | | 47 | 5 | | 1 | 3 | 2 | 1 | 3 | 63 | 3 | | | | 25 | 1 | 6 | | 12 | 35 | 2 | | 3 | 3 | | | | | 3 | 76 | 1 | | | 2 | | | | | | 0,5 | 304 | 608 |
| SG-21, 1-2 cm | | | | | 1 | | | | | | 1 | 1 | | | | | | 1 | 21 | 1 | | | | 115 | 7 | 5 | 1 | | | 1 | | | | 4 | 7 | | | 2 | | | | 2 | | | | | | | 0,5 | 170 | 340 |
| SG-21, 2-3 cm | | | | | 1 | | | | | | | | | | | | | | 1 | | | | | 305 | 5 | 3 | 1 | | | | | | | | 3 | | | | | | | | 1 | | | | | | 0,5 | 315 | 630 |
| SG-21, 3-4 cm | | | | | | | | | | | | | | | | | | | 3 | | | | | 180 | 2 | 1 | | 1 | | | | | | | 1 | 1 | | | 2 | | | | | | | | | | 1 | 191 | 191 |
| SG-21, 4-5 cm | | | | | | | | | | | 1 | | | | | | | | 5 | | | | | 78 | | 2 | 2 | | 1 | | | | | | | | | | | | | | | | | | | | 1 | 89 | 89 |
| SG-21, 5-6 cm | | | | | | | | | | | 1 | | | | | | | | 1 | | | 1 | 1 | 22 | | 1 | | | | | | | | 1 | | | | | | | | | | | | | | | 1 | 28 | 28 |
| SG-27, 0-1 cm | | | | | 15 | 1 | | | 4 | 2 | 52 | 5 | 1 | | | | 1 | 7 | 43 | 27 | 2 | | | 12 | | | | 1 | | | | | | 6 | | | 2 | 13 | 1 | | | | | | 3 | 6 | 1 | | 0,25 | 205 | 820 |
| SG-27, 1-2 cm | | | | | 2 | | | | | | 5 | 1 | 4 | | | | 2 | | 92 | 2 | | | | 3 | 43 | 1 | 2 | | | | | | | | | 2 | 1 | 4 | | | | | 1 | 23 | | 3 | | | 1 | 191 | 191 |
| SG-27, 2-3 cm | | | | | 3 | | | | | | 4 | | 2 | | | | 1 | | 31 | | | | | 26 | 49 | | | | | | | | | | | | | | | | | | 1 | | | | | | 1 | 117 | 117 |
| SG-27, 3-4 cm | | | | | 1 | | | | | | 11 | | 1 | | | 1 | 1 | | 14 | | | | | 29 | 13 | 1 | | | | | | | | | 1 | | 1 | | | | | | | | | | | | 1 | 72 | 72 |
| SG-27, 4-5 cm | | | | | | | | | | | 3 | | | | | | | | 18 | | | | | 12 | 9 | | | | | 2 | | | | | | | 1 | | | | | | | | | | | | 1 | 45 | 45 |
| SG-27, 5-6 cm | | | | | 1 | | | | | | 1 | | | | | | 1 | | 7 | | | | | 2 | 6 | | | | | 2 | | | | | | | 1 | | | | | | | | | | | | 1 | 21 | 21 |

Table B2. Foraminiferal census data and diversity indices for the >125 μm fraction.

| Station | P. fusca | Lagenammina sp. | L. tubulata | encrusting monothal. | H. hirudinea | P. decorata | V. gaussi | M. earlandi | C. jeffreysii | C. wiesneri | V. scitulus | A. rostratus | A. incertus | Glomospira sp. | P. wiesneri | E. bullata | C. alba | Pelosina sp. | Tex. earlandi | R. subfusiformis | A. glomeratum | A. antarctica | other agglutinated | C. aff. parkerianus | G. aff. rossensis | Cassidulina sp. | Trif. earlandi | B. aculeata | S. fusiformis | B. pseudopunctata | Buccella sp. | A. echolsi | P. subcarinata | N. grateloupii | N. iridea | Epistominella sp. | Cibicides sp. | R. globularis | other Rotaliida | P. felsinea | H. gracillimum | P. gracilis | P. fusiformis | other Lagenidae | G. fragilis | C. antarctica | L. calomorpha | Lenticulina sp. | T. hornibrooki | M. cf. antarctica | M. lutea | ?Miliolinella sp. | P. patagonica | other Miliolida | N. pachyderma | fraction of sample picked | total picked forams | total foraminifera per sample | % calcareous forms |
|---|---|---|---|---|---|---|---|---|---|---|---|---|---|---|---|---|---|---|---|---|---|---|---|---|---|---|---|---|---|---|---|---|---|---|---|---|---|---|---|---|---|---|---|---|---|---|---|---|---|---|---|---|---|---|---|---|---|---|---|
| SG-01 | | | | | | | 2 | 1 | 37 | | | 63 | | | | | | | | 58 | | | 5 | 1 | | 5 | | | | | | 6 | 30 | | | | 1 | | | | | | | | | | | | | | | | | | | 1 | 210 | 210,0 | 21 |
| SG-03 | 7 | | | 4 | | | | 53 | 18 | 2 | | 3 | 2 | | 9 | | | 1 | 1 | 38 | | | 7 | 26 | 136 | 4 | | | 3 | | | 39 | 36 | 1 | | | | | | 6 | 3 | | | 1 | 3 | | | | 4 | 2 | | | 2 | | 0,625 | 411 | 657,6 | 64,7 |
| SG-04 | 3 | | | 9 | | | | 47 | 39 | 10 | | 3 | | | 6 | | | 5 | 1 | 13 | | | | 197 | 100 | | | | | | | 12 | 11 | | | | | | | 9 | | 1 | | 3 | | | | | 29 | | | | 2 | | 0,5 | 500 | 1000,0 | 72,8 |
| SG-06 | | 1 | | 1 | | | | 23 | 98 | | | 3 | 1 | | 4 | | | | | 54 | | | 3 | 18 | 126 | | 1 | 3 | | | | 9 | 1 | | | | | | 1 | | | 2 | | | | | 1 | 1 | 1 | | | | 1 | | 0,25 | 353 | 1412,0 | 46,7 |
| SG-07 | | | | | | | | 14 | 13 | | 1 | 74 | 1 | | 2 | | | | | 11 | | | 2 | 58 | 10 | | | | 3 | | | 51 | 18 | 1 | 1 | | 1 | | | 5 | | 3 | | 2 | 1 | | 2 | | 30 | 22 | 1 | | | 4 | | 1 | 329 | 329,0 | 64,1 |
| SG-08 | | | | | | | | 108 | 1 | | | | | | 2 | | | | | | | 2 | 1 | 179 | 6 | | | | 1 | | | | | | | | 1 | | | | | | | | | 1 | | 1 | 2 | | | | | | 0,5 | 304 | 608,0 | 62,5 |
| SG-09 | | | | | | | | 17 | | | | | | | 6 | | | | | | | 3 | 1 | 257 | 12 | | | | | | | 14 | | | | | | | 2 | | | | | | | 1 | 1 | | | | 8 | | | | 1 | 322 | 322,0 | 91,6 |
| SG-10 | | | | 8 | | | | 3 | 2 | | | | | | 9 | | | | | 19 | | | | 218 | 13 | | | | | | | 53 | | | | | | | | 1 | | | | 5 | | | | | 1 | | | | | 1 | 0,5625 | 333 | 592,0 | 87,7 |
| SG-11 | 2 | | | 1 | | | 1 | 2 | 4 | | | 15 | 3 | | 4 | | | | 4 | 26 | | | 4 | 52 | 20 | | | | 5 | | | 155 | | | | | 1 | | | | | 2 | 1 | 2 | | 1 | 1 | | 1 | | | | 1 | | 1 | 308 | 308,0 | 78,6 |
| SG-12 | | | | | | | | 62 | | | | | | | 24 | | | | | | | | 1 | 33 | 5 | 6 | | | | | 1 | | | | | | | | 1 | | | | | | 2 | 4 | | | | | 1 | | | 3 | | 2 | 143 | 71,5 | 39,2 |
| SG-13 | 2 | | | | | | | 58 | | | | | | | | | | | | | | | | 2 | | | | | | | | | | | | | | | | | | | | | 8 | | | | | | | | 7 | | | 2 | 77 | 38,5 | 22,1 |
| SG-14 | | | | | | | | 36 | | | | | | | 1 | | | | 2 | | | | | 137 | | | | | 1 | | | 1 | | | | | | | | | | | | | | | | | 4 | | | 1 | 1 | | 2 | 185 | 92,5 | 78,9 |
| SG-15 | | | | 4 | | | | 14 | 3 | | | 13 | | | 12 | | | | | 15 | | | 1 | 83 | 148 | 1 | | | 2 | | | 3 | 1 | | | | | | 1 | 5 | | | | | | | 2 | 4 | 4 | | | | 1 | | 1 | 317 | 317,0 | 80,4 |
| SG-16 | | | | | | | | 130 | | | | 1 | | | | | | | 2 | | | | | 6 | | 1 | | | | | | | 4 | | | | | | 3 | | | | | | | | | | 2 | 4 | 1 | | 29 | | | 2 | 183 | 91,5 | 27,3 |
| SG-17 | | | | 2 | | | | 6 | 14 | | | 11 | 2 | | 17 | | | | 4 | 1 | 63 | | | 61 | 40 | 1 | | 1 | | | | 4 | | | | | | | | 11 | 1 | 1 | | | | | | | 1 | 1 | 1 | | | 1 | 1 | 1 | 244 | 244,0 | 50,8 |
| SG-18 | 1 | | | | | | | 8 | 19 | | | 5 | 3 | | 27 | | 1 | | | 6 | | | 4 | 123 | 76 | 1 | | | 2 | 6 | | 2 | | | | | | | | 14 | 2 | 1 | | 5 | | | | 2 | 8 | 2 | | | | | 0,25 | 318 | 1272,0 | 76,7 |
| SG-20A | 4 | | | | | | | 9 | 17 | | | 6 | | | 21 | 1 | | | 2 | 1 | | | 2 | 95 | 153 | 1 | | | 5 | 1 | | 16 | 17 | 1 | | | | | | 3 | | 1 | 1 | 1 | | 1 | | | 13 | | | | | | 0,1875 | 371 | 1978,7 | 83 |
| SG-20B | 2 | | | 1 | | | | 7 | 22 | | | | 1 | | 26 | 1 | | | 2 | | | | 1 | 137 | 110 | 23 | | | | 10 | | 13 | 14 | | | | | | 2 | 3 | 1 | | | 9 | 3 | | | | 4 | 1 | | | | | 0,4275 | 393 | 919,3 | 84 |
| SG-21 | 1 | 12 | | | | | | 8 | 7 | 7 | | 61 | 12 | 2 | 1 | | | | 10 | 19 | | | 1 | 88 | 8 | | 3 | 7 | 3 | | 3 | 18 | 44 | 2 | 1 | 2 | | | 4 | 1 | | | | 2 | | 1 | | | 2 | 14 | 1 | | 1 | 7 | 2 | 0,6875 | 353 | 513,5 | 60,1 |
| SG-22 | 2 | | | | | | 9 | 11 | 13 | | | | | | | | | | | 72 | 4 | 4 | 0 | 27 | 5 | | | | | | 3 | | | | | | | | | | | | | | | | | | | | | | | | | 1 | 150 | 150,0 | 23,3 |
| SG-24 | 35 | | | | 25 | | | 37 | | | | | | | | | | | | | | | 0 | 4 | 2 | | | | | | | | | | | | | | 1 | | | | | 5 | 5 | | | | | | | | | | | 1 | 114 | 114,0 | 14,9 |
| SG-26A | 1 | 1 | | 1 | | | | 3 | 11 | 1 | | | | | 4 | 3 | | | 1 | 1 | | | 6 | 50 | 157 | 15 | | | 1 | 12 | 2 | 21 | | | | | | | 2 | | | 1 | 8 | 1 | | 3 | | | 1 | | | | 1 | | 0,5313 | 308 | 579,8 | 89,3 |
| SG-26B | 1 | | | 1 | | | | 61 | | | 1 | 7 | 2 | | 4 | 4 | | | | 7 | | | 2 | 52 | 67 | 12 | | | 1 | | | 11 | 16 | 42 | | | | | | | | | | 26 | 1 | | 2 | 1 | 1 | | | | | 9 | | 0,6094 | 331 | 543,2 | 72,8 |
| SG-27 | 1 | | | | | | | 6 | 67 | | | 3 | 16 | 1 | 35 | 13 | | | | 2 | | | | 129 | 32 | 5 | | | | 9 | | 22 | 20 | | | | | | | 11 | | 3 | | 5 | | 1 | 2 | | 9 | 3 | 1 | | 3 | 4 | | 0,25 | 403 | 1612,0 | 64,3 |
| SG-28 | | | | | | | | 1 | 1 | | | 1 | | | | 3 | 1 | | | | | | | 6 | 10 | | | | | | 4 | | | | | | | | 1 | | | | | | | | | | | | | | | | | 2 | 28 | 14,0 | 75 |
| SG-29 | 5 | 1 | 57 | | | 56 | | 5 | | | | | | | 24 | | | | | | | | 1 | 81 | 62 | | | | | | 2 | | | | 1 | | | 4 | | | | | | | | | | | | | | | | 1 | | 1 | 300 | 300,0 | 50,3 |
| SG-30 | 2 | 5 | | 1 | | 38 | | 18 | 12 | | | 1 | | | 18 | | | | | | | 1 | 1 | 122 | 154 | | | | | | | | | | | | | | 2 | | | | | | | | | | 1 | 9 | | | | 5 | | 0,375 | 390 | 1040,0 | 75,1 |
| SG-31 | 21 | | 32 | | | | | 32 | | | | | | | | | | | | | | | | 3 | 1 | | | | | | | | | | | | | | | | | | | 8 | 2 | | | | | | | | | | | 1 | 99 | 99,0 | 14,1 |
| SG-32 | 1 | | | | | | | 15 | | | | | | | 1 | | | | 2 | 1 | 8 | | | 282 | 12 | | | | | | | 33 | | | | | | | | | | | | 3 | | | | | | | | | | | | 0,625 | 359 | 574,4 | 92,2 |
| **Core** | | | | | | | | | | | | | | | | | | | | | | | | | | | | | | | | | | | | | | | | | | | | | | | | | | | | | | | | | | | | |
| SG-21, 0-1 cm | | 2 | | | | | | 3 | 8 | 4 | 4 | 3 | 2 | | 4 | 7 | 1 | | | 20 | | | 1 | 36 | 7 | 1 | | 3 | 3 | 2 | | 25 | 13 | 2 | 1 | | | | 3 | 1 | 2 | 1 | 1 | | 1 | | | | 5 | 3 | 2 | | | 1 | | 0,5 | 172 | 344,0 | 65,7 |
| SG-21, 1-2 cm | | 1 | | | | | | 2 | 6 | 6 | | 3 | | | 3 | 1 | | | | | | | | 14 | 1 | 1 | | | 4 | 1 | | 14 | 8 | 4 | 1 | | | | 4 | 2 | | | | 3 | | | | 1 | | | | | | | | 1 | 80 | 80,0 | 72,5 |
| SG-21, 2-3 cm | | | | | | | | | | 1 | 1 | | | | | | | | | | | | | 1 | | | | | 4 | | | 2 | 4 | | | | | | | 1 | | | | | | | | | | | | | | | | 1 | 14 | 14,0 | 85,7 |
| SG-21, 3-4 cm | | | | | | | | | | | | | | | | | | | | | | | | 3 | | | | | 3 | | | 2 | 1 | | | | | | | | | | | | | | | | | | | | | | | 1 | 9 | 9,0 | 100 |
| SG-21, 4-5 cm | | | | | | | | | | 1 | 1 | | | | | | | | | | | | 1 | 3 | | | | | | | | 5 | 4 | | | | | | | | | | | | | | | | | | | | | | | 1 | 15 | 15,0 | 80 |
| SG-21, 5-6 cm | | | | | | | | | | | | | | | 2 | | | | | | | | | 2 | | | | | | | | 2 | 1 | | | | | | | | | | | | | | | | | | | | | 1 | | 1 | 8 | 8,0 | 75 |
| SG-27, 0-1 cm | | 1 | | | | | 1 | 12 | 60 | | | 13 | | | 48 | 9 | | | | 1 | | | | 157 | 69 | 6 | | | | 3 | | 22 | 7 | | | | | | | 9 | | 4 | | 5 | | | | | 1 | | 1 | | 20 | 3 | | 0,5 | 452 | 904,0 | 67,9 |
| SG-27, 1-2 cm | | | | | | | | 3 | 11 | | 2 | 5 | | | 1 | 3 | | | | 1 | | | 1 | 111 | 8 | 4 | | | | 8 | | 38 | 14 | | | | | | | 11 | | | | 3 | | | | | 13 | | | | | | | 1 | 236 | 236,0 | 89 |
| SG-27, 2-3 cm | | | | | | | | 2 | 7 | | 1 | 1 | | | | | | | | | | | | 10 | 3 | | | | 1 | | | 2 | | | | | | | | 1 | | | | | | | | 1 | 1 | | | | | | 1 | 29 | 29,0 | 62,1 |
| SG-27, 3-4 cm | | | | | | | | 3 | 3 | | | | | | | | | | | | | | | 7 | | | | | 1 | | | 2 | | | | | | | | 1 | | | | | | | | | | | | | | | | 1 | 17 | 17,0 | 64,7 |
| SG-27, 4-5 cm | | | | | | | 2 | 2 | | | | 1 | 1 | | | | | | | | | | | 4 | | | | | | | | 1 | 3 | | | | | | | 3 | | | | | | | | | | | | | | | | 1 | 17 | 17,0 | 64,7 |
| SG-27, 5-6 cm | | | | | | | | 3 | | | | 1 | | | | | | | | | | | | 2 | | | | | | | | 7 | | | | | | | | | | | | | | | | | | | | | | | | 1 | 13 | 13,0 | 69,2 |

Table B2. *Continues.*

| Station | Taxa_S | Dominance_D | Simpson_1-D | Shannon_H | Evenness_e^H/S | Brillouin | Menhinick | Margalef | Equitability_J | Fisher_alpha | Berger-Parker | Chao-1 |
|---|---|---|---|---|---|---|---|---|---|---|---|---|
| SG-01 | 12 | 0,22 | 0,78 | 1,73 | 0,47 | 1,64 | 0,83 | 2,06 | 0,69 | 2,76 | 0,3 | 15 |
| SG-03 | 26 | 0,16 | 0,84 | 2,32 | 0,39 | 2,22 | 1,28 | 4,15 | 0,71 | 6,17 | 0,33 | 27,2 |
| SG-04 | 19 | 0,22 | 0,78 | 2,01 | 0,39 | 1,94 | 0,85 | 2,9 | 0,68 | 3,91 | 0,39 | 19,5 |
| SG-06 | 21 | 0,24 | 0,76 | 1,8 | 0,29 | 1,72 | 1,12 | 3,41 | 0,59 | 4,89 | 0,36 | 43,5 |
| SG-07 | 24 | 0,13 | 0,87 | 2,39 | 0,45 | 2,27 | 1,32 | 3,97 | 0,75 | 5,96 | 0,22 | 29,3 |
| SG-08 | 11 | 0,47 | 0,53 | 0,95 | 0,24 | 0,91 | 0,63 | 1,75 | 0,4 | 2,24 | 0,59 | 13,5 |
| SG-09 | 11 | 0,64 | 0,36 | 0,89 | 0,22 | 0,84 | 0,61 | 1,73 | 0,37 | 2,2 | 0,8 | 12,5 |
| SG-10 | 12 | 0,46 | 0,54 | 1,24 | 0,29 | 1,18 | 0,66 | 1,89 | 0,5 | 2,44 | 0,65 | 13,5 |
| SG-11 | 23 | 0,3 | 0,7 | 1,8 | 0,26 | 1,69 | 1,31 | 3,84 | 0,57 | 5,75 | 0,5 | 28,6 |
| SG-12 | 12 | 0,27 | 0,73 | 1,63 | 0,43 | 1,51 | 1 | 2,22 | 0,66 | 3,12 | 0,43 | 15 |
| SG-13 | 5 | 0,59 | 0,41 | 0,86 | 0,47 | 0,77 | 0,57 | 0,92 | 0,53 | 1,2 | 0,75 | 5 |
| SG-14 | 10 | 0,59 | 0,41 | 0,84 | 0,23 | 0,78 | 0,74 | 1,72 | 0,37 | 2,27 | 0,74 | 17,5 |
| SG-15 | 19 | 0,29 | 0,71 | 1,72 | 0,29 | 1,63 | 1,07 | 3,13 | 0,58 | 4,44 | 0,47 | 22,3 |
| SG-16 | 11 | 0,53 | 0,47 | 1,07 | 0,26 | 0,99 | 0,81 | 1,92 | 0,44 | 2,57 | 0,71 | 12 |
| SG-17 | 21 | 0,17 | 0,83 | 2,13 | 0,4 | 2 | 1,34 | 3,64 | 0,7 | 5,51 | 0,26 | 33 |
| SG-18 | 22 | 0,22 | 0,78 | 2,02 | 0,34 | 1,91 | 1,23 | 3,65 | 0,65 | 5,37 | 0,39 | 23 |
| SG-20A | 22 | 0,25 | 0,75 | 1,9 | 0,3 | 1,81 | 1,14 | 3,55 | 0,61 | 5,12 | 0,41 | 31,3 |
| SG-20B | 22 | 0,22 | 0,79 | 2,01 | 0,34 | 1,92 | 1,11 | 3,52 | 0,65 | 5,03 | 0,35 | 25,8 |
| SG-21 | 32 | 0,12 | 0,88 | 2,61 | 0,43 | 2,47 | 1,7 | 5,28 | 0,75 | 8,54 | 0,25 | 36,7 |
| SG-22 | 10 | 0,28 | 0,72 | 1,68 | 0,53 | 1,57 | 0,82 | 1,8 | 0,73 | 2,41 | 0,48 | 10 |
| SG-24 | 8 | 0,25 | 0,75 | 1,57 | 0,6 | 1,46 | 0,75 | 1,48 | 0,75 | 1,96 | 0,32 | 8 |
| SG-26A | 25 | 0,3 | 0,7 | 1,85 | 0,25 | 1,73 | 1,43 | 4,19 | 0,57 | 6,43 | 0,51 | 43,3 |
| SG-26B | 23 | 0,13 | 0,87 | 2,35 | 0,46 | 2,24 | 1,26 | 3,79 | 0,75 | 5,62 | 0,2 | 28,3 |
| SG-27 | 25 | 0,15 | 0,85 | 2,37 | 0,43 | 2,26 | 1,25 | 4 | 0,74 | 5,9 | 0,32 | 27 |
| SG-28 | 9 | 0,21 | 0,79 | 1,81 | 0,68 | 1,47 | 1,7 | 2,4 | 0,82 | 4,59 | 0,36 | 19 |
| SG-29 | 13 | 0,19 | 0,81 | 1,81 | 0,47 | 1,74 | 0,75 | 2,1 | 0,71 | 2,77 | 0,27 | 16 |
| SG-30 | 16 | 0,27 | 0,73 | 1,68 | 0,33 | 1,61 | 0,81 | 2,51 | 0,61 | 3,36 | 0,39 | 19,3 |
| SG-31 | 7 | 0,26 | 0,74 | 1,49 | 0,64 | 1,39 | 0,7 | 1,31 | 0,77 | 1,72 | 0,32 | 7 |
| SG-32 | 11 | 0,63 | 0,37 | 0,87 | 0,22 | 0,83 | 0,58 | 1,7 | 0,36 | 2,15 | 0,79 | 14 |

Table B3. Foraminiferal census data and diversity indices for the entire assemblage (>63 μm; i.e., the >125 plus 63–125 μm fractions).

| Station | P. fusca | Lagenammina sp. | L. tubulata | encrusting monothal. | H. hirudinea | P. decorata | V. gaussi | M. earlandi | C. jeffreysii | C. wiesneri | V. scitulus | A. rostratus | A. incertus | Glomospira sp. | P. wiesneri | E. bullata | C. alba | Pelosina sp. | Tex. earlandi | R. subfusiformis | A. glomeratum | E. minuta | P. antarctica | other agglutinated | C. aff. parkerianus | G. aff. rossensis | Cassidulina sp. | Trif. earlandi | B. aculeata | S. fusiformis | B. pseudopunctata | Buccella sp. | A. echolsi | P. subcarinata | N. grateloupii | N. iridea | Epistominella sp. | Cibicides sp. | R. globularis | other Rotaliida | P. felsinea | H. gracillimum | P. gracilis | P. fusiformis | other Lagenidae | G. fragilis | C. antarctica |
|---|---|---|---|---|---|---|---|---|---|---|---|---|---|---|---|---|---|---|---|---|---|---|---|---|---|---|---|---|---|---|---|---|---|---|---|---|---|---|---|---|---|---|---|---|---|---|---|
| SG-01 | | | | | | | 2,0 | 1,0 | 37,0 | | | 63,0 | | | 1,0 | | | | | 58,0 | | | | 5,0 | 1,0 | 1,0 | 5,0 | | | | | | 6,0 | 30,0 | | | | 1,0 | 1,0 | | | 1,0 | | | | | |
| SG-03 | 11,2 | | | | 13,1 | | | 132,8 | 31,5 | 3,2 | | 4,8 | 5,9 | | 27,7 | 1,6 | 1,6 | | 12,0 | 80,8 | | 6,7 | | 13,9 | 117,6 | 257,6 | 10,7 | 6,4 | 1,3 | 30,7 | 46,1 | | 83,7 | 62,9 | 20,0 | 10,9 | 1,3 | | | 5,3 | 24,3 | 6,1 | 5,3 | 22,7 | 6,9 | 46,7 | 7,5 |
| SG-04 | 6,0 | | | | 18,0 | | | 678,0 | 78,0 | 20,0 | | | 14,0 | | 52,0 | | 10,0 | | 26,0 | 34,0 | | | | | 754,0 | 560,0 | | | | 136,0 | | | 48,0 | 46,0 | | 16,0 | 16,0 | | | 8,0 | 186,0 | | | 122,0 | 22,0 | | |
| SG-06 | | 4,0 | | | 4,0 | | | 193,3 | 410,7 | | | 12,0 | 4,0 | | 90,7 | | | | | 242,7 | | | | 12,0 | 229,3 | 832,0 | 8,0 | 4,0 | 12,0 | | | | 38,7 | 4,0 | | | 21,3 | | | 4,0 | 10,7 | | | 66,7 | 21,3 | | 8,0 |
| SG-07 | | | | | | | | 22,0 | 21,0 | | 1,0 | 74,0 | 4,0 | | 8,0 | 1,0 | | | | 17,0 | | 5,0 | | 2,0 | 133,0 | 14,0 | 4,0 | | | 26,0 | 40,0 | | 70,0 | 21,0 | 17,0 | 45,0 | 1,0 | | | | 16,0 | 11,0 | 8,0 | 19,0 | 6,0 | | 1,0 |
| SG-08 | | | | | | | | 311,0 | 2,0 | | | | | | 12,0 | | | 4,0 | | 2,0 | | 1,0 | | 1,0 | 423,0 | 37,0 | 14,0 | | | 2,0 | 13,0 | | 17,0 | | | | 8,0 | | 1,0 | 1,0 | 17,0 | | | 11,0 | 3,0 | | 3,0 |
| SG-09 | | | | | | | | 38,0 | | | | | | | 23,0 | | | | | 4,0 | | | | 1,0 | 487,0 | 17,0 | 9,0 | | | 3,0 | | | 17,0 | | 1,0 | 7,0 | | 1,0 | | | | | | 18,0 | | | 2,0 |
| SG-10 | | | 14,2 | | | | | 13,3 | 3,6 | | | | | | 38,0 | | | | | 36,8 | | | | | 602,6 | 43,1 | 1,0 | | | 4,0 | | | 98,2 | | 2,0 | | | 1,0 | 1,0 | | 1,8 | 1,0 | | 8,9 | | | 2,0 |
| SG-11 | 2,0 | | | | 1,0 | | 1,0 | 4,0 | 11,0 | | | 15,0 | 4,0 | | 12,0 | 1,0 | | 4,0 | | 37,0 | | 3,0 | | 4,0 | 162,0 | 41,0 | | 1,0 | 6,0 | 3,0 | 8,0 | | 235,0 | 6,0 | 25,0 | 3,0 | | | 2,0 | 3,0 | 8,0 | 8,0 | 3,0 | | | | 4,0 |
| SG-12 | | | | | | | | 44,5 | | | | | | | 16,5 | | | | | | | 21,5 | | 0,5 | 17,5 | 3,0 | 19,0 | | | 1,0 | 4,0 | | | | | 2,5 | 1,5 | | | | | | | 4,5 | | 1,5 | 2,0 |
| SG-13 | 1,0 | | | | | | | 33,5 | | | | | | | | | | | | 0,5 | | | | | 1,0 | | | | | | | | | | 1,0 | 0,5 | | | 0,5 | | | | | 3,5 | | 4,0 | 0,5 |
| SG-14 | | | | | | | | 39,0 | | | | | | | 4,5 | | | | | 20,5 | 0,5 | | | | 105,5 | 8,5 | 3,0 | 0,5 | | 1,0 | 1,5 | | 0,5 | | 1,0 | | 2,5 | | | | | 6,0 | | | 1,0 | | 1,0 |
| SG-15 | | | | | 4,0 | | | 32,0 | 3,0 | | | 16,0 | | | 53,0 | | | | 8,0 | 19,0 | | 1,0 | 4,0 | 2,0 | 124,0 | 185,0 | 15,0 | 1,0 | | 25,0 | 2,0 | | 28,0 | 1,0 | 2,0 | | 10,0 | | 3,0 | | 40,0 | | | 9,0 | | | 1,0 |
| SG-16 | | | | | | | | 100,5 | 0,5 | | | | | | 0,5 | | | | 10,0 | | | | | 0,5 | 5,0 | | 2,0 | | | | | | 1,0 | | | | 0,5 | | | 1,5 | | | | 18,5 | | | 1,0 |
| SG-17 | | | | | 3,0 | | | 17,0 | 14,0 | | | 11,0 | 2,0 | | 50,0 | | | 4,0 | 12,0 | 91,0 | | 3,0 | | 22,0 | 117,0 | 78,0 | 13,0 | 1,0 | | 54,0 | 3,0 | | 2,0 | 5,0 | 1,0 | 2,0 | | | | | 20,0 | 2,0 | 1,0 | 14,0 | | | |
| SG-18 | 4,0 | | | | | | | 75,4 | 76,0 | | | 20,0 | 16,6 | | 176,6 | | 4,0 | | 30,9 | | 6,9 | 20,6 | | 18,3 | 775,4 | 349,7 | 22,3 | | | 50,3 | 94,9 | 24,0 | 8,0 | | | | 25,1 | | | | 78,9 | 8,0 | 4,0 | 56,6 | 6,9 | | 9,1 |
| SG-20A | 21,3 | | | | | | | 80,0 | 98,7 | | | | 36,0 | | 236,0 | 9,3 | | 10,7 | 37,3 | | | 8,0 | | 10,7 | 926,7 | 1216,0 | 48,0 | | | 9,3 | 24,0 | 126,7 | 5,3 | 97,3 | 102,7 | 25,3 | 28,0 | 56,0 | | | 16,0 | 4,0 | 4,0 | 13,3 | 9,3 | | 5,3 |
| SG-20B | 4,7 | | | | 2,3 | | | 27,8 | 51,5 | | | | 9,2 | | 264,2 | 2,3 | | 4,7 | 13,7 | 4,6 | | | | 9,1 | 583,3 | 321,3 | 13,7 | 53,8 | | 20,6 | 38,9 | 23,4 | 32,7 | 37,3 | | 11,4 | 38,9 | 2,3 | | 4,7 | 16,2 | 2,3 | | 41,6 | 4,6 | 7,0 | |
| SG-21 | 1,5 | 17,5 | | | | | | 11,6 | 23,9 | 17,0 | 88,7 | 19,7 | 9,8 | | 63,2 | 23,7 | | | 6,9 | 50,5 | 2,3 | 9,1 | 6,9 | 12,9 | 340,6 | 18,5 | 4,6 | 4,4 | 12,5 | 155,2 | 4,6 | 4,4 | 71,9 | 73,1 | 46,3 | 90,6 | 2,9 | | | 10,4 | | 3,7 | | 4,6 | 7,5 | | |
| SG-22 | 2,0 | | | | | | 9,0 | 11,0 | 13,0 | | | | | | 74,0 | | 4,0 | 4,0 | | | | | | | 28,0 | 5,0 | | | | | 4,0 | | | | 1,0 | | | | | | | | | 13,0 | | 5,0 | |
| SG-24 | 35,0 | | 25,0 | | | | | 61,0 | | | | | | | | | | | | | | | | | 6,0 | 2,0 | | | | | | | | | | | | | 1,0 | | | | | 13,0 | | 5,0 | |
| SG-26A | 1,9 | 3,2 | | | 1,9 | | | 9,6 | 22,0 | 3,2 | | | | 1,3 | 80,9 | 5,6 | | 1,9 | 3,2 | | | 8,0 | | 14,0 | 279,5 | 391,5 | 14,7 | 29,6 | | 4,0 | 8,5 | 22,6 | 3,8 | 40,9 | | 17,3 | 16,0 | | | 3,8 | 1,9 | | | 39,1 | 5,9 | | |
| SG-26B | 1,6 | 1,6 | | | | | | 1,9 | | | 1,6 | 11,5 | 3,3 | | 181,6 | 6,6 | | 1,9 | 11,5 | | | 1,9 | 3,8 | 10,8 | 202,0 | 164,5 | 24,5 | 19,7 | | 1,6 | 24,5 | 18,1 | 30,0 | 166,8 | 11,3 | 16,9 | 20,7 | | | 1,9 | 1,9 | | 1,9 | 59,6 | 5,4 | | 3,3 |
| SG-27 | 4,0 | | | | | | | 88,0 | 293,6 | | | 12,0 | 98,1 | 55,2 | 575,2 | 64,8 | | | 12,8 | 8,0 | | 12,8 | | 29,9 | 946,9 | 238,9 | | 20,0 | | 4,3 | 59,7 | 36,0 | 143,5 | 114,1 | | 12,8 | 68,3 | | | | 65,3 | | 41,9 | 113,9 | 8,5 | | 4,0 |
| SG-28 | | | | | | | | 1,5 | 0,5 | | | | 0,5 | | 1,5 | 0,5 | | | | | | 1,5 | | 0,5 | 3,5 | 6,0 | 10,5 | | | 2,5 | 32,0 | | | | | | 7,0 | | 3,0 | | | | | 5,0 | | | |
| SG-29 | 5,0 | | 1,0 | 58,0 | 57,0 | | | 6,0 | | | | | | | 28,0 | | | | | | | 1,0 | | 3,0 | 91,0 | 94,0 | 3,0 | | | | 21,0 | | | | | 2,0 | 27,0 | 4,0 | | | | | | | | | |
| SG-30 | 5,3 | 13,3 | | | 3,9 | 107,5 | | 64,0 | 38,2 | | | 2,7 | 2,5 | | 82,5 | | | | 2,7 | | | 1,2 | | 8,8 | 432,4 | 528,8 | 7,4 | | | | 30,8 | | | | | | 52,9 | | | 10,3 | 1,2 | | | 11,1 | 3,7 | 1,2 | |
| SG-31 | 21,0 | | 32,0 | | | | | 68,0 | | | | | | | | | | | | 1,0 | | | | | 3,0 | 4,0 | | | | | | | | | | | | | | | | | | 18,0 | | 2,0 | |
| SG-32 | 1,6 | | | | 1,0 | | | 47,0 | | | | | | | 4,6 | | | | 3,2 | 13,6 | 33,8 | | | 1,0 | 540,2 | 31,2 | 3,0 | | | 9,0 | 1,0 | | 92,8 | | | 3,0 | 4,0 | 2,0 | | | 2,0 | | | 14,8 | 1,0 | | 2,0 |

| Station | *L. calomorpha* | *Lenticulina sp.* | *T. hornibrooki* | *M. cf. antarctica* | *M. lutea* | *?Miliolinella sp.* | *P. patagonica* | *other Miliolida* | total picked forams | total foraminifera per sample | % calcareous forms | Taxa_S | Dominance_D | Simpson_1-D | Shannon_H | Evenness_e^H/S | Brillouin | Menhinick | Margalef | Equitability_J | Fisher_alpha | Berger-Parker | Chao-1 |
|---|---|---|---|---|---|---|---|---|---|---|---|---|---|---|---|---|---|---|---|---|---|---|---|
| SG-01 | | | | | | | | | 213 | 213,0 | 21,6 | 15 | 0,216 | 0,784 | 1,783 | 0,396 | 1,682 | 1,03 | 2,614 | 0,658 | 3,686 | 0,297 | 25,5 |
| SG-03 | | | 6,4 | 3,2 | 1,3 | | | 3,2 | 769 | 1134,9 | 69,45 | 38 | 0,097 | 0,903 | 2,805 | 0,435 | 2,619 | 1,128 | 5,273 | 0,771 | 7,576 | 0,227 | 38 |
| SG-04 | | | 730,0 | | 32,0 | | | 4,0 | 827 | 3616,0 | 74,12 | 24 | 0,15 | 0,85 | 2,224 | 0,385 | 2,207 | 0,399 | 2,807 | 0,7 | 3,45 | 0,209 | 24 |
| SG-06 | 4,0 | | 6,7 | | 4,0 | | | 4,0 | 668 | 2252,0 | 56,78 | 26 | 0,202 | 0,798 | 2,021 | 0,29 | 1,975 | 0,548 | 3,24 | 0,62 | 4,124 | 0,369 | 26 |
| SG-07 | 25,0 | | 59,0 | 27,0 | 1,0 | | | 4,0 | 703 | 703,0 | 77,95 | 31 | 0,082 | 0,918 | 2,849 | 0,557 | 2,761 | 1,169 | 4,576 | 0,83 | 6,635 | 0,189 | 36 |
| SG-08 | 4,0 | | 1,0 | | | | | | 566 | 870,0 | 61,84 | 20 | 0,367 | 0,633 | 1,378 | 0,198 | 1,339 | 0,678 | 2,807 | 0,46 | 3,651 | 0,486 | 21,5 |
| SG-09 | 1,0 | | | | 41,0 | | | | 670 | 670,0 | 90,15 | 16 | 0,539 | 0,461 | 1,182 | 0,204 | 1,141 | 0,618 | 2,305 | 0,426 | 2,946 | 0,727 | 19 |
| SG-10 | | | | 6,8 | 8,0 | | | 2,8 | 631 | 890,0 | 88,1 | 20 | 0,477 | 0,523 | 1,31 | 0,185 | 1,225 | 0,67 | 2,801 | 0,437 | 3,633 | 0,676 | 22 |
| SG-11 | 5,0 | 2,0 | 1,0 | | | | | 2,0 | 627 | 627,0 | 84,21 | 33 | 0,219 | 0,781 | 2,135 | 0,256 | 2,048 | 1,318 | 4,968 | 0,611 | 7,418 | 0,375 | 35 |
| SG-12 | | | | 1,0 | 0,5 | | | 3,5 | 289 | 144,5 | 42,56 | 17 | 0,166 | 0,834 | 2,127 | 0,493 | 1,773 | 1,414 | 3,242 | 0,751 | 5,004 | 0,305 | 17,5 |
| SG-13 | | | 1,0 | | | | 3,5 | | 101 | 50,5 | 30,69 | 12 | 0,458 | 0,542 | 1,337 | 0,317 | 0,839 | 1,689 | 2,857 | 0,538 | 4,977 | 0,654 | 18 |
| SG-14 | | | 2,0 | | | | 0,5 | 0,5 | 397 | 198,5 | 67,51 | 18 | 0,336 | 0,664 | 1,574 | 0,268 | 1,315 | 1,278 | 3,23 | 0,545 | 4,807 | 0,529 | 19,5 |
| SG-15 | 2,0 | 23,0 | 1,0 | 4,0 | 13,0 | | | 1,0 | 632 | 632,0 | 77,53 | 30 | 0,146 | 0,854 | 2,429 | 0,378 | 2,345 | 1,193 | 4,497 | 0,714 | 6,551 | 0,293 | 33 |
| SG-16 | | | 1,0 | 2,0 | 0,5 | 14,5 | | | 317 | 158,5 | 29,34 | 15 | 0,43 | 0,571 | 1,351 | 0,257 | 1,096 | 1,191 | 2,779 | 0,499 | 4,067 | 0,631 | 15,33 |
| SG-17 | | | 1,0 | 1,0 | 3,0 | | | 1,0 | 548 | 548,0 | 58,21 | 29 | 0,118 | 0,882 | 2,492 | 0,417 | 2,4 | 1,239 | 4,44 | 0,74 | 6,529 | 0,214 | 32 |
| SG-18 | 8,0 | | 52,6 | 8,0 | 20,6 | | | | 659 | 2051,4 | 78,11 | 29 | 0,189 | 0,811 | 2,297 | 0,343 | 2,224 | 0,64 | 3,674 | 0,682 | 4,782 | 0,378 | 29 |
| SG-20A | | | 69,3 | | 24,0 | | | | 716 | 3358,7 | 83,68 | 30 | 0,218 | 0,782 | 2,106 | 0,274 | 2,066 | 0,518 | 3,573 | 0,619 | 4,54 | 0,362 | 30 |
| SG-20B | | | 11,6 | 2,3 | | | | | 718 | 1662,2 | 76,28 | 31 | 0,192 | 0,809 | 2,211 | 0,294 | 2,104 | 0,76 | 4,05 | 0,644 | 5,409 | 0,351 | 31 |
| SG-21 | 6,0 | 2,3 | 9,8 | 24,9 | 6,0 | | 1,5 | 10,2 | 689 | 1281,5 | 71,51 | 40 | 0,109 | 0,891 | 2,793 | 0,408 | 2,612 | 1,117 | 5,462 | 0,757 | 7,839 | 0,265 | 40 |
| SG-22 | | | | | | | | | 155 | 155,0 | 24,52 | 11 | 0,279 | 0,721 | 1,706 | 0,5 | 1,593 | 0,884 | 1,983 | 0,711 | 2,706 | 0,477 | 11 |
| SG-24 | | | | | | | | | 131 | 148,0 | 18,24 | 8 | 0,265 | 0,735 | 1,557 | 0,593 | 1,47 | 0,658 | 1,401 | 0,749 | 1,812 | 0,412 | 8 |
| SG-26A | 5,6 | 1,3 | 2,7 | 1,9 | | | | 1,9 | 660 | 1049,1 | 85,06 | 33 | 0,222 | 0,778 | 2,1 | 0,247 | 1,928 | 1,019 | 4,611 | 0,601 | 6,479 | 0,373 | 33 |
| SG-26B | | 1,6 | 1,6 | | | | | 14,8 | 646 | 1136,1 | 69,77 | 35 | 0,114 | 0,886 | 2,536 | 0,361 | 2,346 | 1,038 | 4,846 | 0,713 | 6,837 | 0,178 | 35 |
| SG-27 | 16,5 | | 74,4 | 16,3 | 16,8 | 54,7 | | 16,0 | 805 | 3327,2 | 62,3 | 33 | 0,133 | 0,867 | 2,578 | 0,399 | 2,518 | 0,572 | 3,948 | 0,737 | 5,089 | 0,284 | 33 |
| SG-28 | | | | | 0,5 | | | | 153 | 76,5 | 91,5 | 16 | 0,219 | 0,781 | 2,01 | 0,466 | 1,454 | 1,829 | 3,519 | 0,725 | 6,163 | 0,418 | 16 |
| SG-29 | | | | | | | | 1,0 | 402 | 402,0 | 60,45 | 16 | 0,16 | 0,84 | 2,061 | 0,491 | 1,99 | 0,798 | 2,501 | 0,743 | 3,333 | 0,234 | 17,5 |
| SG-30 | | | 3,9 | | 1,2 | 24,0 | | 13,3 | 727 | 1454,8 | 77,14 | 26 | 0,234 | 0,766 | 1,928 | 0,264 | 1,833 | 0,682 | 3,437 | 0,592 | 4,496 | 0,363 | 26 |
| SG-31 | | | | | | | | | 149 | 149,0 | 18,12 | 8 | 0,29 | 0,71 | 1,487 | 0,553 | 1,403 | 0,655 | 1,399 | 0,715 | 1,809 | 0,456 | 8 |
| SG-32 | | | 22,6 | | 1,0 | | | | 620 | 835,4 | 87,34 | 23 | 0,438 | 0,562 | 1,437 | 0,183 | 1,348 | 0,796 | 3,273 | 0,458 | 4,375 | 0,646 | 25,5 |

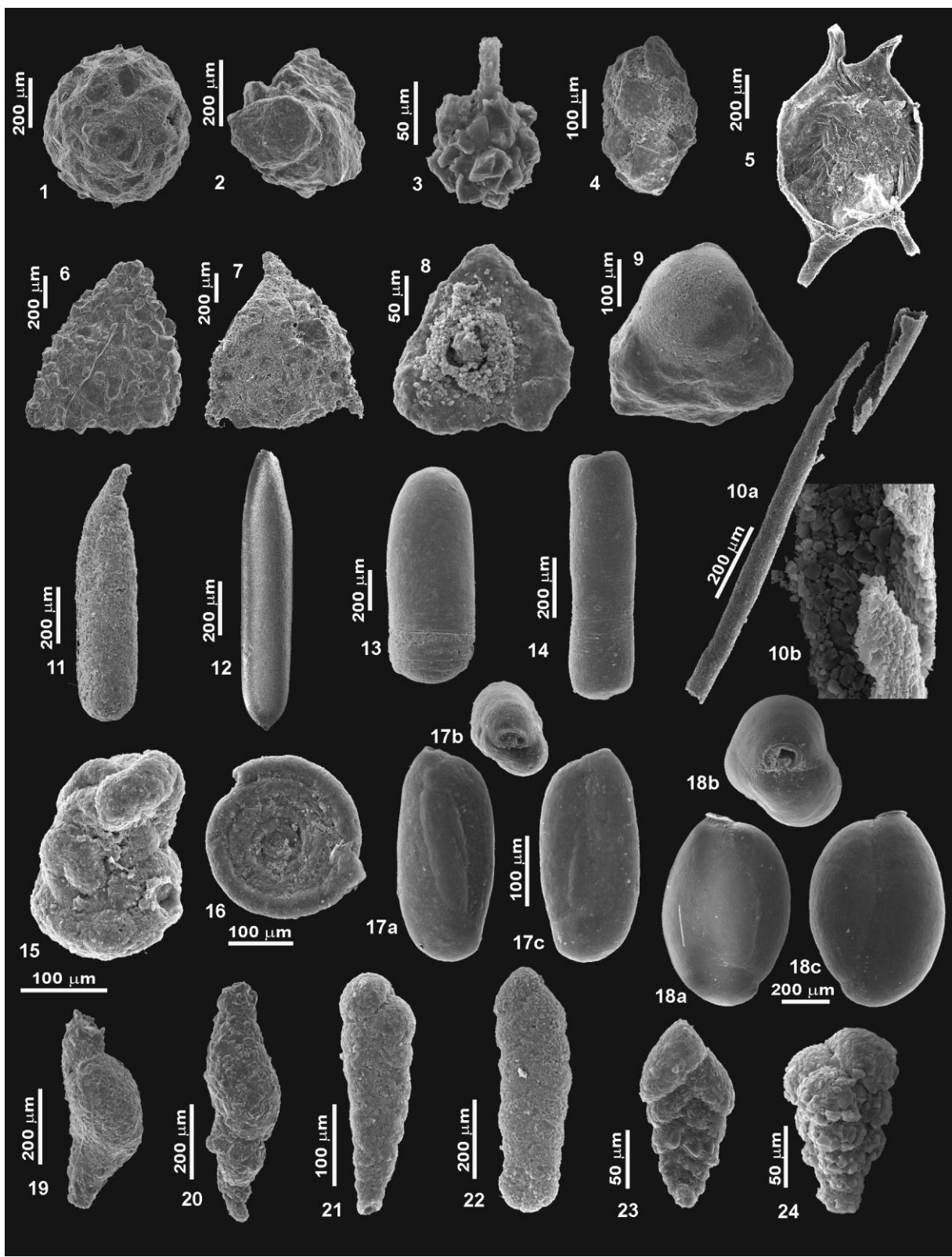

**Fig. C1. Agglutinated foraminifera: 1-2.** *Psammosphaera fusca* **Schulze, 1875, SG-24, SG-24; 3.** *Lagenammina* **cf.** *tubulata* **(Rhumbler, 1931), SG-21; 4.** *Proteonina decorata* **Earland, 1933, SG-30; 5.** *Vanhoeffenella gaussi* **Rhumbler, 1905, SG-27; 6-7.** *Astrorhiza trangularis* **Earland, 1933, SG-30, SG-22; 8-9. Encrusting monothalamiids, SG-29, SG-29; 10.** *Bathysiphon* **sp., SG-04; 11.** *Pelosina variabilis* **var.** *constricta* **Earland, 1933), SG-11; 12.** *Cribrothallammina alba* **(Heron-Allen and Earland, 1932), SG-21; 13-14.** *Hippocrepinella hirudinea* **Heron-Allen and Earland, 1932, SG-11,**

SG-24; 15. *Glomospira* sp., SG-27; 16. *Ammodiscus incertus* Cushman, 1917, SG-27; 17. *Miliammina earlandi* Loeblich and Tappan, 1955, SG-27, 18. *Miliammina lata* Heron-Allen and Earland, 1930, SG-21; 19-20. *Reophax subfusiformis* Earland, 1933, SG-21, SG-17; 21. *Textularia earlandi* Parker, 1952, SG-27; 22. *Spiroplectammina biformis* (Parmore roundedker and Jones, 1865), SG-22; 23. *Pseudobolivina antarctica* Wiesner, 1931, SG-27; 24. *Eggerella minuta* (Wiesner, 1931), SG-21.

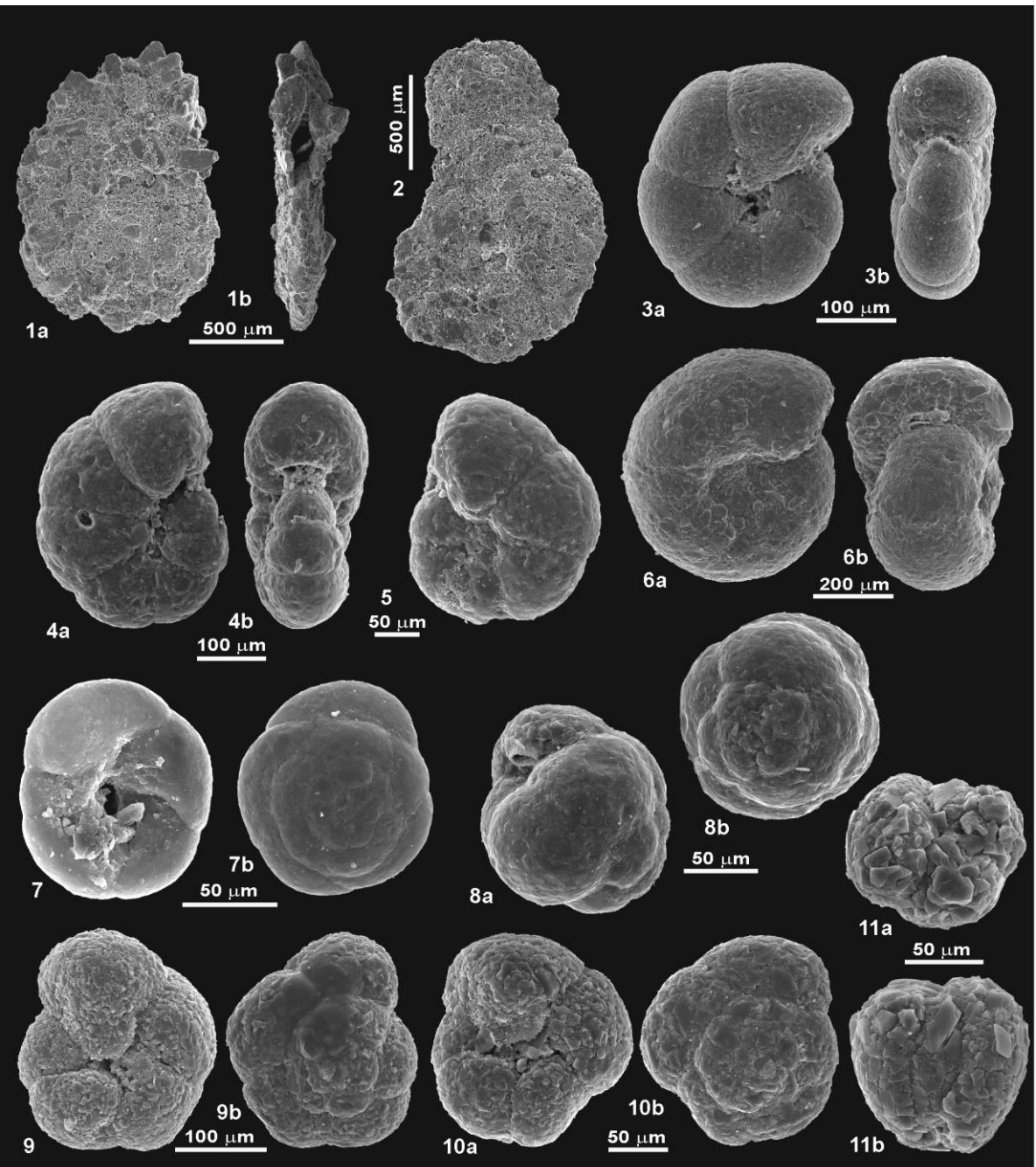

**Fig. C2. Agglutinated foraminifera: 1-2.** *Ammobaculites rostratus* **Heron-Allen and Earland, 1929, SG-26B, SG-18; 3.** *Cribrostomoides wiesneri* **(Parr, 1950), SG-21; 4-5.** *Cribrostomoides jeffreysii* **(Williamson, 1858), SG-27, SG-27; 6.** *Veleroninoides scitulus* **(Brady, 1881), SG-21; 7.** *Paratrochammina bartrami* **(Hedley, Hurdle et Burdett, 1967), SG-29; 8.** *Earlandammina bullata* **(Höglund, 1947), SG-27; 9-10.** *Portatrochammina antarctica wiesneri* **(Parr, 1950), SG-28, SG-27; 11.** *Adercotryma glomeratum* **(Brady, 1878), SG-21.**

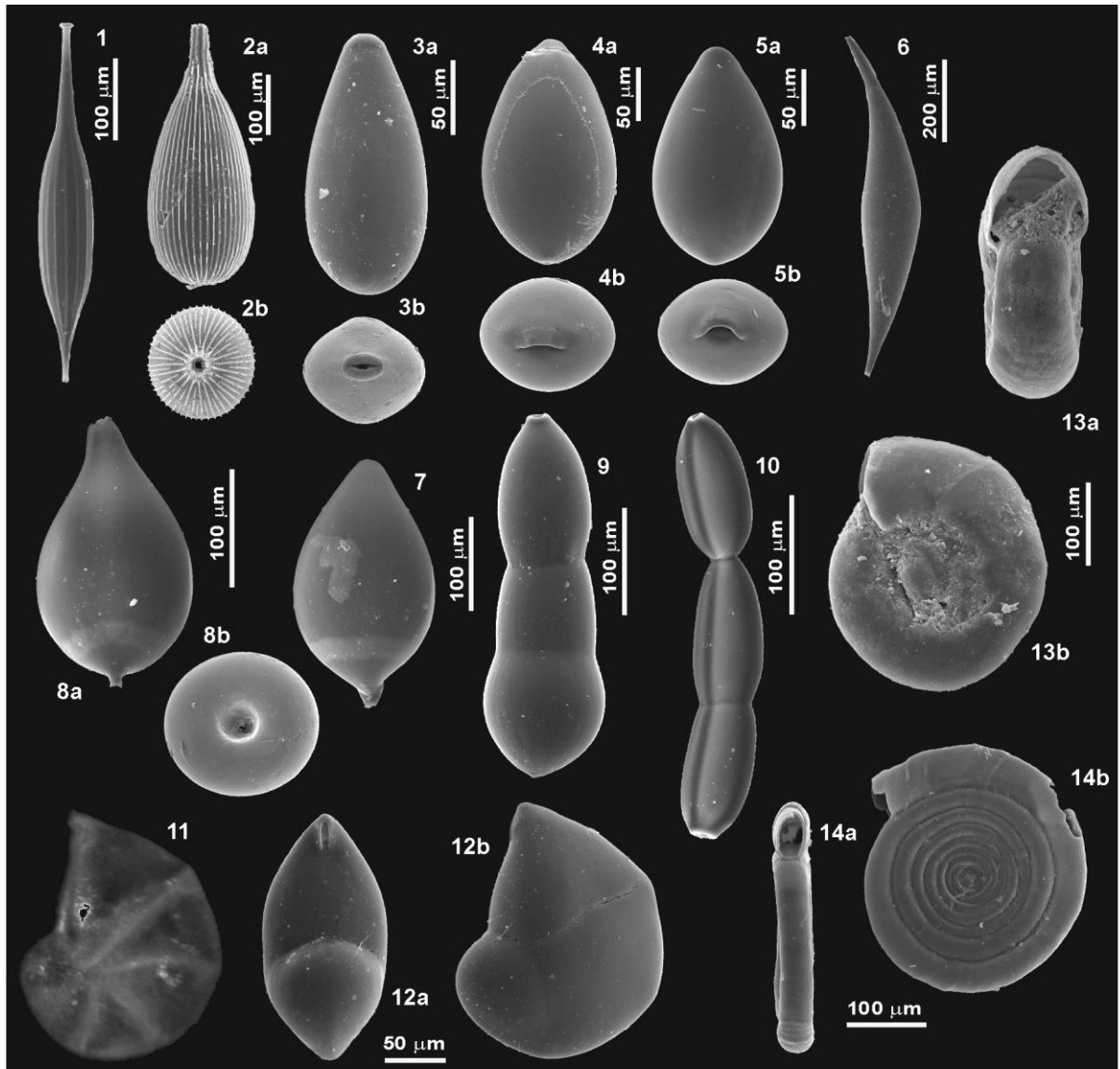

**Fig. C3. Unilocular and some miliolid calcareous foraminifera: 1.** *Procerolagena gracilis* **(Williamson, 1848), SG-27; 2.** *Lagena substriata* **Williamson, 1848, SG-22; 3.** *Fissurina* **sp., SG-27; 4-5.** *Parafissurina fusiformis* **(Wiesner, 1931), SG-27, SG-21; 6.** *Hyalinonetrion* **gracillimum (Seguenza, 1862), SG-21; 7-8.** *Parafissurina felsinea* **(Fornasini, 1894), SG-27, SG-21; 9.** *Nodosaria* **sp., SG-11; 10.** *Lotostomoides calomorpha* **(Reuss, 1866), SG-24; 11-12.** *Lenticulina* **sp., SG-15; 13.** *Gordiospira fragilis* **Heron-Allen and Earland, 1932, SG-24; 14.** *Cornuspira antarctica* **Rhumbler 1931, SG-21.**

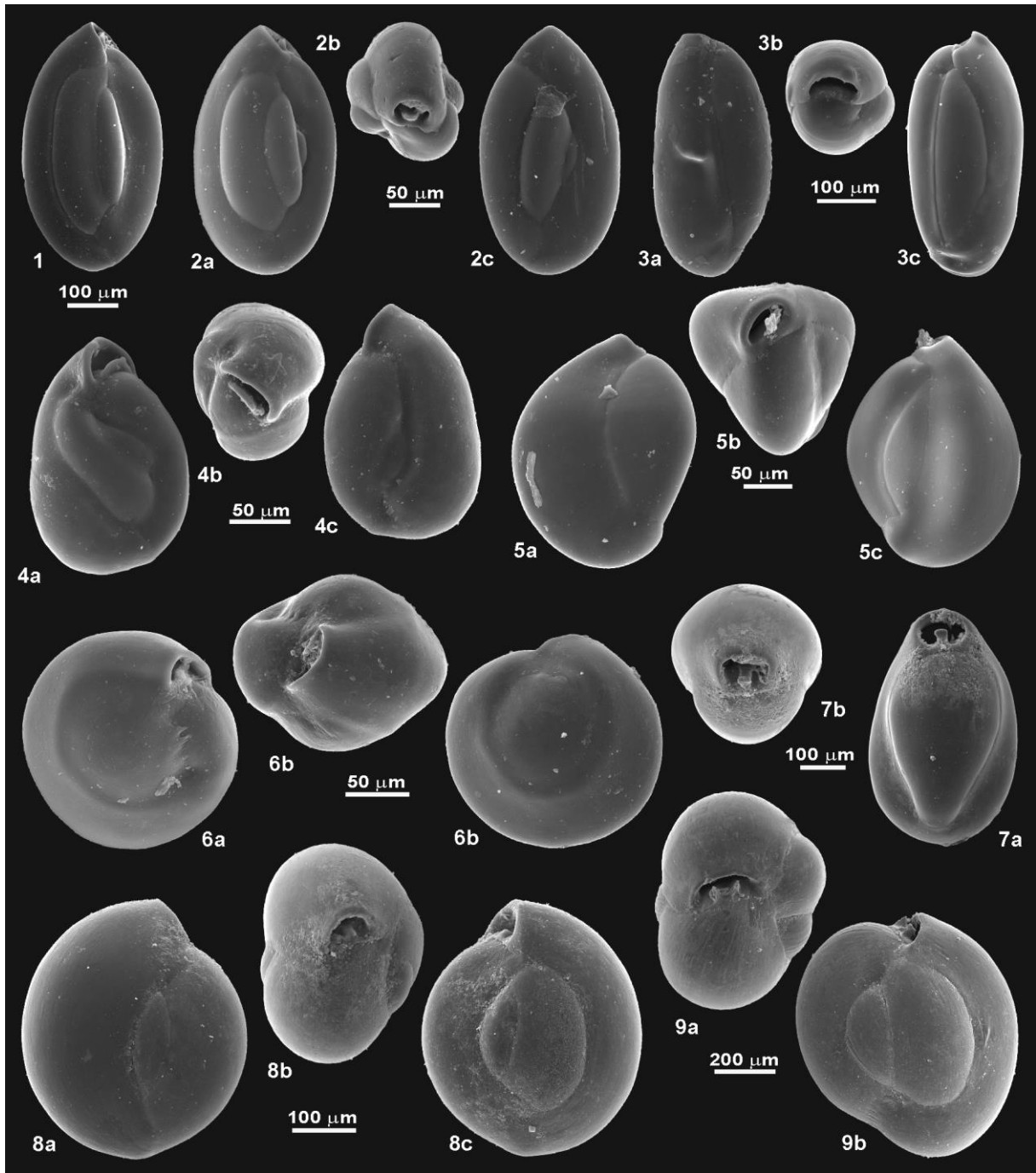

**Fig. C4. Miliolid foraminifera: 1-2.** *Triloculinella hornibrooki* **(Vella, 1957), SG-04, SG-27; 3.** *Miliolinella* **cf.** *antarctica* **(Kennett, 1967), SG-07; 4.** *Miliolinella lutea* **(d'Orbigny, 1939)** *sensu* **Figueroa et al. (2006), SG-27; 5.** *Quinqueloculina seminulum* **(Linnaeus, 1758), SG-27; 6. ?***Miliolinella* **sp., SG-27; 7.** *Pyrgo patagonica* **(d'Orbigny, 1839), SG-16; 8-9.** *Miliolinella subrotunda* **(Montagu, 1803)***,* **SG-07, SG-26B.**

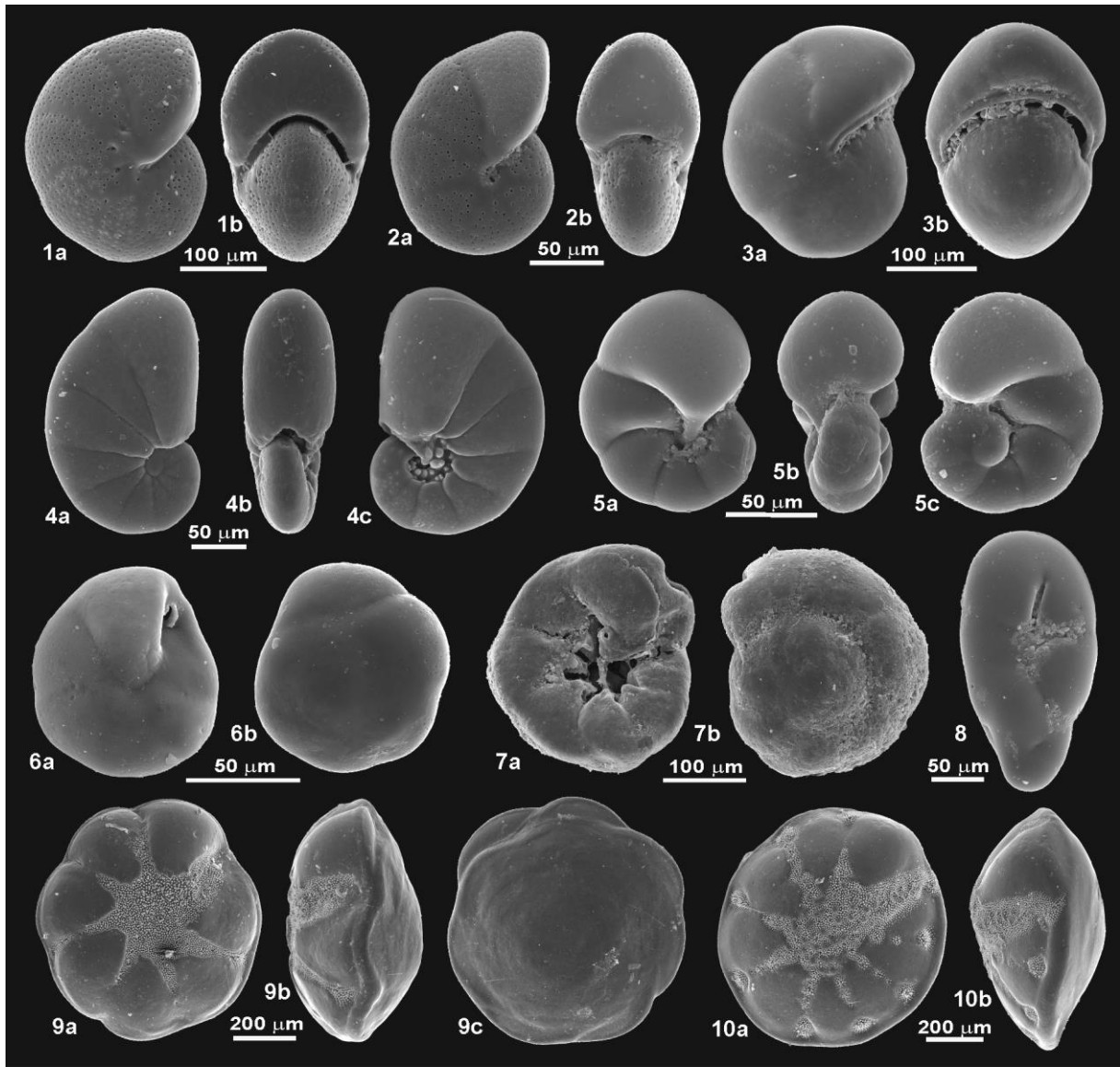

**Fig. C5. Some rotalid foraminifera and *Robertinoides*: 1-2.** *Astrononion echolsi* **Kennett, 1967, SG-27, SG-21; 3.** *Pullenia subcarinata* **(d'Orbigny, 1839), SG-27; 4.** *Nonionoides grateloupii* **(d'Orbigny, 1839), SG-21; 5.** *Nonionella iridea* **Herron-Allen and Earland, 1932, SG-21; 6.** *Epistominella* **sp., SG-26B; 7.** *Rosalina globularis* **d'Orbigny, 1826, SG-15; 8.** *Robertinoides* **sp., SG-21; 9-10.** *Buccella* **sp., SG-26B, SG-26B.**

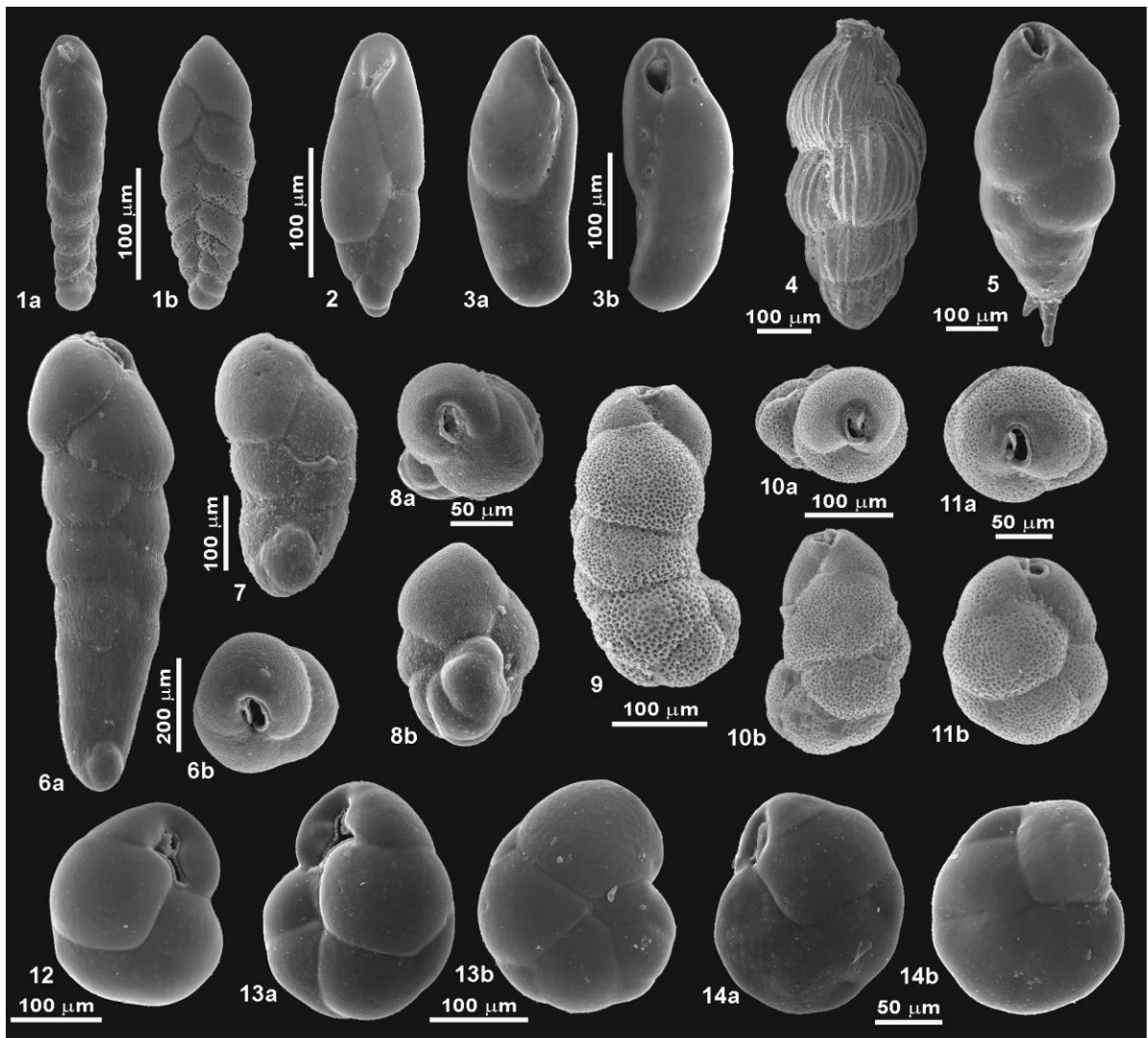

**Fig. C6. Rotalid foraminifera: 1.** *Bolivinellina pseudopunctata* **(Höglund, 1947), SG-27; 2.** *Stainforthia fusiformis* **(Williamson, 1858), SG-21; 3.** ?*Bulimina* **sp., SG-21; 4.** *Trifarina earlandi* **(Parr, 1950), SG-27; 5.** *Bulimina aculeata* **d'Orbigny, 1826, SG-21; 6-11.** *Cassidulinoides* **aff.** *parkerianus* **Brady, 1881, SG-09, SG-09, SG-09, SG-17, SG-27, SG-27; images 6 to 8 show the smoothly-walled conical morphotype morphotype assigned by Heron-Allen and Earland (1929) and Earland (1933) to** *Ehrenbergina crassa***; 12-13.** *Globocassidulina* **aff.** *rossensis* **Kennett, 1967, SG-27, SG-17; 14.** *Cassidulina* **sp., SG-12.**

Table D1. Results of the Q-mode principal component (PC) analysis for the >125 μm dataset. The PC scores >1 and <-1 (in bold) show significant contribution of the selected variables (foraminiferal species) for each foraminiferal assemblages, following Malmgren and Haq (1982). Nominative species are underlined.

| | PC1 | PC2 | PC3 | PC4 | Total |
|---|---|---|---|---|---|
| Total variance explained | 32.85 | 14.95 | 27.31 | 8.00 | 83.10 |
| *P. fusca* | -0.34 | **1.08** | -0.18 | -0.53 | |
| *Lagenammina* sp. | -0.18 | -0.29 | -0.31 | -0.46 | |
| *L. tubulata* | -0.12 | -0.31 | -0.36 | -0.37 | |
| *H. hirudinea* | -0.33 | **1.08** | -0.23 | -0.48 | |
| *P. decorata* | 0.06 | -0.39 | -0.01 | **-1.05** | |
| *M. earlandi* | 0.05 | **5.51** | 0.07 | 0.22 | |
| *C. jeffreysii* | -0.52 | -0.29 | **1.05** | 1.73 | |
| *C. wiesneri* | -0.13 | -0.29 | -0.33 | -0.40 | |
| *V. scitulus* | 0.13 | -0.38 | -0.52 | -0.03 | |
| *A. rostratus* | -0.38 | -0.18 | -0.47 | **3.12** | |
| *A. incertus* | -0.12 | -0.30 | -0.21 | -0.37 | |
| *P. wiesneri* | 0.58 | -0.04 | 0.19 | -1.11 | |
| *E. bullata* | -0.09 | -0.33 | -0.32 | -0.35 | |
| *Tex. earlandi* | -0.12 | -0.26 | -0.32 | -0.48 | |
| *R. subfusiformis* | -0.65 | -0.11 | 0.43 | **2.73** | |
| *C.* aff. *parkerianus* | **5.68** | -0.12 | 0.74 | 0.40 | |
| *G.* aff. *rossensis* | -0.82 | -0.23 | **5.51** | -0.70 | |
| *Cassidulina* sp. | -0.15 | -0.19 | -0.34 | -0.46 | |
| *Trif. earlandi* | -0.22 | -0.32 | -0.07 | -0.26 | |
| *B. aculeata* | -0.15 | -0.29 | -0.34 | -0.34 | |
| *S. fusiformis* | -0.15 | -0.29 | -0.34 | -0.41 | |
| *B. pseudopunctata* | -0.16 | -0.27 | -0.28 | -0.44 | |
| *Buccella* sp. | -0.15 | -0.34 | -0.15 | -0.42 | |
| *A. echolsi* | 0.21 | -0.33 | -0.28 | **2.46** | |
| *P. subcarinata* | -0.29 | -0.31 | 0.25 | **1.36** | |
| *R. globularis* | -0.17 | -0.26 | -0.30 | -0.48 | |
| *P. felsinea* | -0.09 | -0.30 | -0.19 | -0.27 | |
| *P. fusiformis* | -0.21 | 0.01 | -0.09 | -0.27 | |
| *G. fragilis* | -0.21 | 0.10 | -0.32 | -0.43 | |
| *C. antarctica* | -0.17 | -0.22 | -0.31 | -0.41 | |
| *Lenticulina* sp. | -0.18 | -0.29 | -0.31 | -0.44 | |
| *T. hornibrooki* | -0.02 | -0.25 | -0.28 | 0.12 | |
| *M. cf. antarctica* | -0.07 | -0.25 | -0.39 | 0.08 | |
| *M. lutea* | -0.14 | -0.27 | -0.34 | -0.41 | |
| ?*Miliolinella* sp. | -0.16 | -0.29 | -0.30 | -0.47 | |
| *P. patagonica* | -0.23 | 0.19 | -0.34 | -0.36 | |

Table D2. Results of the Q-mode principal component (PC) analysis for the >63 μm dataset. The PC scores >1 and <-1 (in bold) show significant contribution of the selected variables (foraminiferal species) for each foraminiferal assemblages, following Malmgren and Haq (1982). Nominative species are underlined.

| | PC1 | PC2 | PC3 | PC4 | Total |
|---|---|---|---|---|---|
| Total variance explained | 31.00 | 17.47 | 28.12 | 5.39 | 81.97 |
| *P. fusca* | -0.40 | 0.72 | -0.33 | 0.20 | |
| *L. tubulata* | -0.22 | -0.27 | -0.39 | 0.25 | |
| *H. hirudinea* | -0.40 | 0.71 | -0.33 | 0.10 | |
| *P. decorata* | -0.38 | -0.35 | 0.04 | 0.87 | |
| ***M. earlandi*** | -0.10 | **6.16** | 0.10 | -0.21 | |
| ***C. jeffreysii*** | -0.19 | -0.33 | 0.78 | **-1.47** | |
| *C. wiesneri* | -0.22 | -0.25 | -0.38 | 0.23 | |
| *V. scitulus* | -0.02 | -0.30 | -0.49 | 0.16 | |
| ***A. rostratus*** | 0.35 | -0.30 | -0.26 | **-3.02** | |
| *A. incertus* | -0.19 | -0.27 | -0.26 | 0.27 | |
| *Glomospira* sp. | -0.24 | -0.26 | -0.38 | 0.30 | |
| ***P. wiesneri*** | 0.41 | -0.41 | 1.00 | **2.53** | |
| *E. bullata* | -0.17 | -0.29 | -0.38 | 0.29 | |
| *Tex. earlandi* | -0.08 | -0.03 | -0.34 | 0.32 | |
| ***R. subfusiformis*** | 0.03 | -0.22 | 0.64 | **-3.41** | |
| *E. minuta* | -0.24 | -0.27 | -0.36 | 0.31 | |
| *P. antarctica* | -0.17 | 0.10 | -0.32 | 0.44 | |
| ***C. aff. parkerianus*** | **5.98** | 0.11 | **1.33** | 0.84 | |
| ***G. aff. rossensis*** | **-1.57** | -0.23 | **5.86** | -0.07 | |
| *Cassidulina* sp. | -0.29 | 0.08 | -0.11 | 0.86 | |
| *Trif. earlandi* | -0.24 | -0.29 | -0.22 | 0.12 | |
| *S. fusiformis* | 0.34 | -0.33 | -0.32 | -0.40 | |
| ***B. pseudopunctata*** | -0.32 | -0.21 | 0.37 | **1.25** | |
| *Buccella* sp. | -0.23 | -0.29 | -0.29 | 0.34 | |
| ***A. echolsi*** | **1.15** | -0.35 | -0.29 | **-2.16** | |
| ***P. subcarinata*** | 0.01 | -0.38 | 0.27 | **-1.23** | |
| *N. grateloupii* | -0.06 | -0.25 | -0.37 | -0.03 | |
| *N. iridea* | 0.27 | -0.33 | -0.47 | -0.39 | |
| *Epistominella* sp. | -0.33 | -0.29 | 0.06 | 0.82 | |
| *R. globularis* | -0.28 | -0.20 | -0.32 | 0.38 | |
| *P. felsinea* | -0.14 | -0.17 | -0.03 | -0.02 | |
| *H. gracillimum* | -0.17 | -0.26 | -0.38 | 0.05 | |
| *P. gracilis* | -0.20 | -0.26 | -0.37 | 0.18 | |
| *P. fusiformis* | -0.18 | 0.71 | 0.00 | 0.20 | |
| *G. fragilis* | -0.37 | 0.10 | -0.23 | 0.07 | |
| *C. antarctica* | -0.26 | -0.19 | -0.34 | 0.26 | |
| *L. calomorpha* | -0.09 | -0.26 | -0.42 | 0.01 | |
| *Lenticulina* sp. | -0.29 | -0.26 | -0.28 | 0.27 | |
| *T. hornibrooki* | 0.21 | 0.04 | -0.27 | -0.36 | |

| | | | | |
|---|---|---|---|---|
| *M.* cf. *antarctica* | -0.04 | -0.19 | -0.44 | -0.01 |
| *M. lutea* | -0.16 | -0.24 | -0.34 | 0.31 |
| ?*Miliolinella* sp. | -0.25 | -0.26 | -0.35 | 0.31 |
| *P. patagonica* | -0.29 | 0.07 | -0.41 | 0.21 |