# Peer review of "Unique benthic foraminiferal communities (stained) in diverse environments of sub-Antarctic fjords, South Georgia"

_Biogeosciences, 2022_

## Author Comment (AC2)

[revised manuscript text omitted]
 | C1 | C2 | C3 | C4 | C5 | C6 | C7 | C8 | C9 | C10 | C11 | C12 | C13 | C14 | C15 | C16 | C17 | C18 | C19 | C20 | C21 | C22 | C23 | C24 | C25 | C26 | C27 | C28 |
|---|---|---|---|---|---|---|---|---|---|---|---|---|---|---|---|---|---|---|---|---|---|---|---|---|---|---|---|---|
| SG-21, 0-1 cm | 2 |  | 3 | 8 | 4 | 4 | 3 | 2 | 4 | 7 | 1 | 20 |  | 1 | 36 | 7 | 1 | 3 | 3 | 2 |  | 25 | 13 | 2 | 1 | 3 |  | 1 |
| SG-21, 1-2 cm | 1 |  | 2 | 6 | 6 | 3 |  |  | 3 | 1 |  |  |  |  | 14 | 1 | 1 |  | 4 | 1 |  | 14 | 8 | 4 | 1 | 4 | 2 | 3 |
| SG-21, 2-3 cm |  |  |  | 1 | 1 |  |  |  |  |  |  |  |  |  | 1 |  |  |  |  |  |  | 4 | 2 | 4 |  |  |  | 1 |
| SG-21, 3-4 cm |  |  |  |  |  |  |  |  |  |  |  |  |  |  | 3 |  |  |  |  |  |  | 3 | 2 | 1 |  |  |  |  |
| SG-21, 4-5 cm |  |  |  | 1 |  | 1 |  |  |  |  |  |  | 1 |  | 3 |  |  |  |  |  |  | 5 | 4 |  |  |  |  |  |
| SG-21, 5-6 cm |  |  |  |  |  |  |  |  | 2 |  |  |  |  |  | 2 |  |  |  |  |  |  | 2 | 1 |  |  |  |  |  |
|  |  |  |  |  |  |  |  |  |  |  |  |  |  |  |  |  |  |  |  |  |  |  |  |  |  |  |  |  |
| SG-27, 0-1 cm | 1 | 1 | 12 | 60 |  |  | 13 |  | 48 | 9 |  | 1 |  |  | 157 | 69 | 6 |  |  |  | 3 | 22 | 7 |  |  |  |  | 9 |
| SG-27, 1-2 cm |  |  | 3 | 11 | 2 |  | 5 |  |  | 1 | 3 |  |  | 1 | 111 | 8 | 4 |  |  | 8 |  | 38 | 14 |  |  |  |  | 11 |
| SG-27, 2-3 cm |  |  | 2 | 7 |  | 1 | 1 |  |  |  |  |  |  |  | 10 | 3 |  |  |  | 1 |  | 2 |  |  |  |  |  |  |
| SG-27, 3-4 cm |  |  | 3 | 3 |  |  |  |  |  |  |  |  |  |  | 7 |  |  |  |  | 1 |  |  | 2 |  |  |  |  | 1 |
| SG-27, 4-5 cm |  | 2 | 2 |  |  | 1 | 1 |  |  |  |  |  |  |  | 4 |  |  |  |  |  |  | 1 | 3 |  |  |  |  | 3 |
| SG-27, 5-6 cm |  |  |  | 3 |  |  | 1 |  |  |  |  |  |  |  | 2 |  |  |  |  |  |  |  | 7 |  |  |  |  |  |

[revised manuscript text omitted]

---

## Author Response (AR1)

Dear Editor,

Together with the co-authors we prepared revision of the manuscript. We would like to thank the Reviewers for their comments and suggestions. Thank you very much for all these!

Following the remark from Katrine Husum, the target number of specimens for picking in the 63–125 mm and >125 mm fractions was increased to reach 300 each. If it was not reached, there was simply no more material to be analyzed. We recalculated all statistics, updated figures, adjusted the text and corrected Tables B1 to B3, C1 and C2. The only significant difference in the final outcome of this change was that in one assemblage the nominative species changed from *Reophax subfusiformis* to *Ammobaculites rostratus*. It was due to a change in relative importance of these two species for defining that assemblage. Both species were important in the old and new versions of the manuscript.

The replies to the detailed comments of the final reviewer are listed below.

With the kindest regards,

Wojtek Majewski and co-authors

**Lines 44-46: references needed.**

Sentence "Moreover, many calcareous foraminifera secrete their tests in equilibrium with ambient sea water, which makes them useful for reconstructing past water temperatures and salinities using proxies based on analyzing elemental ratios."

Is changed into

"Moreover, elemental and isotopic composition of tests of calcareous foraminiferal may be calibrated to reflect composition of ambient sea water, which makes them useful for reconstructing past environmental conditions, including water temperatures and salinities (de Nooijer et al., 2014).

de Nooijer, L.J., Spero, H.J., Erez, J. Bijma, J., Reichart, G.J. Biomineralization in perforate foraminifera, Earth-Science Reviews, 135, 48–58, https://doi.org/10.1016/j.earscirev.2014.03.013, 2014.

**Line 116: from Table 1, it looks like CTD measurements are available from 19 stations (not 20).**

Table 1 is corrected. Thank you for finding this. They are 20 stations.

**Line 143: why not all van Veen Grab samples were analyzed for grain size?**

Indeed, four samples from four samples were not taken for grain size analysis due to limited material in the grab sampler. In the case of 20B and 26B, where only single grab samples were taken (due to heavy drift of the vessel), only samples from 20A and 26A were analyzed. In the case of station 22 the sediment was composed of a mixture of gravel, algae and finer

sediments. The finer sediments were sampled for foraminifera, and the remaining material was not representative of the bulk sample. Sample 30 was originally taken as an extra sample for foraminifera analysis only, and no subsample for grain size was collected during the survey.

We added a short notice on it in line 133.

**Lines 161-162: "For stations SG-12, SG-13, SG-14, SG-16, and SG-28, specimens from both replicates were picked and further analyzed." Were replicates treated in the same way as the other samples? At line 134 the authors say: "Replicates from the remaining stations were archived.", so it might be worth clearly stating how the replicates were treated, to avoid confusion.**

Sentences "If samples yielded suspiciously few stained individuals, replicates were checked for consistency. For stations SG-12, SG-13, SG-14, SG-16, and SG-28, specimens from both replicates were picked and further analyzed."

are changed into:

"If samples yielded <300 stained individuals, specimens from replicates were also picked in the same way as the regular samples. Consequently, for stations SG-12, SG-13, SG-14, SG-16, and SG-28, specimens from both replicates were further analyzed."

I hope it is clear now.

**Line 200: Station SG-02 is not reported in Figures 1 and 2 or in Table 1.**

Yes, it should be SG-03.

**Lines 283-287: this information belong to the Discussion section.**

Its more an interpretation and it is important. I am afraid that if we place it in the discussion, in chapter "4.2.1 Inner parts of Cumberland Bay and shallow-water coves: strong glacial influence and sediment anoxia" this information will being somehow "hidden"

Thus, it is moved to the figure caption of Fig. 5, which now reads:

"Fig. 5. Total organic carbon to total sulfur ratio (TOC/TS) (a), carbon stable isotopes of bulk organic matter in the sediments (b) in relation to distance from fjord mouth, and relation between TOC and the $\delta^{13}C$ values (c). Note different trends/mixing lines in (a): towards increasing TOC/TS ratios in Stromness Bay and towards lower ratios in Cumberland Bay, and in (b): towards less negative $\delta^{13}C$ coves for Stromness Bay and coves and towards more negative values for the main basins of Cumberland Bay affected by tidewater glaciers. Dashed TOC/TS lines in (a) are after Berner (1983). They demarcate TOC/TS ratios suggested for anoxic (<1.5), periodically anoxic (1.5–5), and oxic conditions (ratios >5). Trends in (b) may represent the progressive mixing of different types of bulk organic matter, namely (1) material typical for open-marine conditions ($\delta^{13}C$ ~–24‰), (2) likely petrogenic organic carbon supplied by glaciers ($\delta^{13}C$ ~–26‰), and (3) organic matter derived from fresh terrestrial and marine sources ($\delta^{13}C$ ~–23‰)."

**Section 3.2.1 The authors should refer to the dataset they compiled and that they are planning to submit to Pangaea after the paper has been accepted for publication (although I recommend starting the process early rather than later because it might take a while for the dataset to be published in the repository).**

I see that BG actually allows Appendices, therefore we turned the Supplements into Appendices. It will solve the problem.

**Lines 310-311. I recommend adding a reference to the photos of these species in Supp. Figure S1-S6.**

References added.

**Line 369. Can the authors provide an estimate/opinion about how much the reduced staining time could have biased their results? They already provide a nice discussion on how they think that the modification they made to the FOBIMO protocol did not affect their results, but something similar is missing with respect to the substantial decrease in incubation time with Rose Bengal.**

A sentence was added:

"We believe it was a reasonable precaution, especially since 24–48 hours of staining can already provide satisfactory results (Bernhard et al. 2006).

Bernhard, J.M., Ostermann, D.R. Williams, D.S., and Blanks J.K. Comparison of two methods to identify live benthic foraminifera: A test between Rose Bengal and CellTracker Green with implications for stable isotope paleoreconstructions, Paleoceanography, 21, PA4210, doi:10.1029/2006PA001290, 2006.

**I would be nice if the authors were to add a (small) section regarding the applicability of their results, possibly to fossil data, as well as suggestions for future research directions.**

This section may be added at the beginning of the final chapter 5. Concluding remarks:

"As already indicated, benthic foraminifera can serve as valuable proxies for marine environmental conditions recorded in the geological record. It is therefore somewhat surprisingly that no attempts have been made to use them in order to reconstruct coastal environments around sub-Antarctic islands during past climatic oscillations, such as those associated with Quaternary glacial/interglacial cycles. Our results demonstrate that these microfossils have considerable potential in this regard. They seem especially suitable for studying paleoenvironmental changes in the most restricted settings proximal to tidewater glacial fronts in shallow-water settings that are strongly affected by processes taking place on land. These may be rich in organic matter, which is believed to exert a strong control on foraminiferal assemblage composition and diversity.
        The sensitivity of foraminifera to environmental changes linked to current and likely future climatic changes are of more immediate interest…'

Addition future direction of studies, i.e., monitoring of faunal changes in reaction to rapid warming of the Drake Passage sector of the Southern Ocean is stated at the end of the final chapter.

**Table 1 – header 'Date' – please specify the year. Station SG-13, please revise date format. Also, can the authors add additional columns to specify which samples were analyzed for grain size, TC, TOC, TS, and d¹³C of bulk organic matter?**

Corrected, additional column "Sedimentary indices" added.

**Legend Figures 3, 4, 5, 6, and 9. "outer Cunberland Bay" – it should be Cumberland.**

Done. Thank you.

**Figure 2. I wonder if this figure would be more helpful if the data were grouped by sampling position with respect to the fjord (i.e., near shore, mid-fjord, outer fjord) rather than sampling area (i.e., Antarctic, Fortuna, Cumberland, Stromness bays). In case, Section 3.1.1 should be revised accordingly.**

We found easier to group and describe the CTD results as they differ between different fjords, that is why they are shown this way in Fig. 2. The difference between long, glaciated fjords of Cumberland Bay and short, unglaciated Stromness is very large and inner sites at these two locations are very different. The profiles from outer fjord setting are marked in yellow in all cases, on the same scales, so comparison between those on the present figure is also possible.

**Figure 5b. Can the authors include TOC data on this figure to better show the relationship between TOC and organic matter d13C?**

The additional graph (c) is added to Fig. 5. Revised caption goes as the answer for the comment for Lines 283-287…

**Figure 6. I am having a bit of a hard time deciphering the maps (?) on the y-axis of the plots. If the authors think that these are important, then I suggest adding them to the site of the plots, making them bigger, and explaining their meaning in the figure caption.**

If I understand correctly, these are not maps, but schematic images of glacier fronts. The sentences "Schematic profiles of the Nordenskjöld and Neumayer glaciers marked in bright and dark grey, respectively." Is added to the captions of Figs. 6 and 9.

**Caption Figure 8. FA should be defined.**

Done. It now goes "Important species for each foraminiferal assemblage (FA), encircled, defined by the Q-mode PC analysis (Table D1) are in bold, the dominant species are underlined

**Table S3 is not cited anywhere in the text.**

Added in chapter "3.2.4 Relation between FAs and environmental and sediment properties" and "3.1.1 Water temperature and salinity" consequently it became Table A and the remaining tables (Appendices) are now re-numbered.

---

## Author Response (AR2)

Dear Editor,

We have now corrected the manuscript according to the Technical corrections suggested.

- Please remove acronyms where possible. I had trouble following your story because I could not memorize the large number of acronyms used in your manuscript. As a rule of thumb use acronyms only if all of the following applies: (1) the term is used more than 5 times and (2) the term consists of more than two words. Some examples where sentences become almost unreadable due to too many abbreviations:

507-508: According to our CCA, This FA is strongly correlated with elevated salinity (Fig. 10), pointing to a possible association with WW and UCDW.

515-516: By contrast with the apparent association of A. rostratus FA with WW and UCDW, the distribution of the G. aff. rossensis FA (Fig. 9) may coincide with the upper and warmer AASW...

>> DONE. We have removed acronyms for total carbon (TC) and all the water masses (WW, UCDW, AASW, LW).

- Figures: Please do not use red and green symbols together in one graph unless there is another characteristic to distinguish the datapoints.

>> All datapoints marked in dark orange changed into grey.

- Supplementary Figures and Tables are usually labeled Fig. S1, S2; Table S1, S2 etc.

>> No change here, as the additional files are Appendices not Supplements. They are numbered according to the www guidelines :

1. "**Appendices**: all material required to understand the essential aspects of the paper such as experimental methods, data, and interpretation should preferably be included in the main text. Additional figures, tables, as well as technical and theoretical developments which are not critical to support the conclusion of the paper, but which provide extra detail and/or support useful for experts in the field and whose inclusion in the main text would disrupt the flow of descriptions or demonstrations may be presented as appendices. These should be labelled with capital letters: Appendix A, Appendix B etc. Equations, figures and tables should be numbered as (A1), Fig. B5 or Table C6, respectively. Please keep in mind that appendices are part of the manuscript whereas supplements (see below) are published along with the manuscript."

- I don't understand how the second page of Table B2 and B3 fit with their respective first page. Please clarify.

>> Corrected.

With the kindest regards,

Wojciech Majewski and co-authors